# Instance-Level Composed Image Retrieval

**Bill Psomas**[1*]   **George Retsinas**[2*]   **Nikos Efthymiadis**[1]   **Panagiotis Filntisis**[2,4]
**Yannis Avrithis**[5]   **Petros Maragos**[2,3,4]   **Ondrej Chum**[1]   **Giorgos Tolias**[1]

[1]VRG, FEE, Czech Technical University in Prague   [2]Robotics Institute, Athena Research Center
[3]National Technical University of Athens   [4]Hellenic Robotics Center of Excellence   [5]IARAI

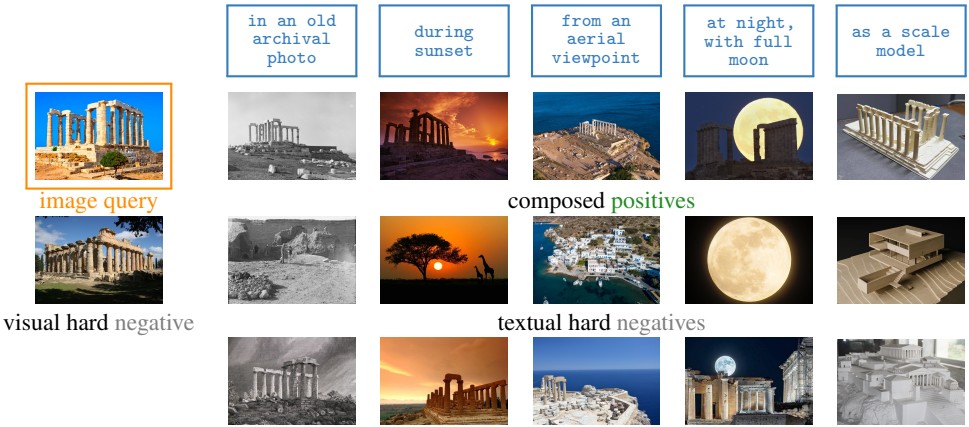

Figure 1: We introduce *i*-nstance-level Composed Image Retrieval (*i*-CIR) evaluation dataset. Given an image query depicting a specific instance (*e.g.*, Temple of Poseidon) along with a modifying text query, the task is to retrieve images showing the same instance altered according to the text (composed positives). Unlike existing datasets [54, 30, 2], *i*-CIR explicitly ensures the presence of challenging negative examples across three distinct dimensions: visual, textual, and composed.

## Abstract

The progress of *composed image retrieval* (CIR), a popular research direction in image retrieval, where a combined visual and textual query is used, is held back by the absence of high-quality training and evaluation data. We introduce a new evaluation dataset, *i*-CIR, which, unlike existing datasets, focuses on an instance-level class definition. The goal is to retrieve images that contain the same particular object as the visual query, presented under a variety of modifications defined by textual queries. Its design and curation process keep the dataset compact to facilitate future research, while maintaining its challenge—comparable to retrieval among more than 40M random distractors—through a semi-automated selection of hard negatives. To overcome the challenge of obtaining clean, diverse, and suitable training data, we leverage pre-trained vision-and-language models (VLMs) in a training-free approach called BASIC. The method separately estimates query-image-to-image and query-text-to-image similarities, performing late fusion to upweight images that satisfy both queries, while downweighting those that exhibit high similarity with only one of the two. Each individual similarity is further improved by a set of components that are simple and intuitive. BASIC sets a new state of the art on *i*-CIR but also on existing CIR datasets that follow a semantic-level class definition. Project page: https://vrg.fel.cvut.cz/icir/.

---

[*]Equal contribution

39th Conference on Neural Information Processing Systems (NeurIPS 2025).

# 1 Introduction

*Composed image retrieval* (CIR) combines image-to-image retrieval and text-to-image retrieval. CIR uses a composed query, *i.e.* an image and text, to retrieve images whose content matches both the visual and textual parts of the query. Vision and language models (VLMs) [41, 25, 26, 20, 58] provide the foundation for developing CIR methods, either through further training [30, 11, 1, 31] or in a training-free manner [39, 9, 23]. The use of VLMs, owing to their large-scale pre-training, enables CIR to operate in an open-world setting and compare any kind of visual or textual content. This capability paves the way for novel applications and advanced methods to explore and browse large image collections. However, the main limitation of CIR lies in the lack of appropriate data for both evaluating progress and training models. This work addresses these challenges.

Existing CIR datasets [30, 54, 2] often suffer from poor quality [2, 18] due to their construction process, *i.e.* two similar images are selected automatically and their difference is textually described. This approach incorrectly assumes that such a difference always forms a meaningful text query for retrieval, regardless of the image pair. In contrast, given one image, we first specify a textual modification such that both together form a meaningful composed query. We then identify positive and a large set of hard negative images to construct *i*-CIR, a compact yet challenging dataset. The goal is to retrieve images that depict the same object instance as the image query, under the modification described by the text query. Such an instance-level object class definition is missing from existing datasets and is identified as a limitation by prior work [44]. By integrating diverse object types and modification types, *i*-CIR accurately reflects a wide range of real-world use cases.

CIR methods that rely on further training on top of VLMs require large amounts of training triplets in the form of (query image, query text, positive image), which are challenging to obtain at scale. As a result, training is typically performed on small sets of manually labeled triplets [51, 30, 1, 5], or on automatically generated triplets obtained through crawling [60, 21, 29] or synthetic data generation [13]. However, these automated methods significantly compromise triplet quality, and in all cases, the diversity of visual object types and textual modifications remains limited compared to the variety present in VLM pre-training, thereby restricting generalization ability. Instead, we develop a training-free Baseline Approach for SurprIsingly strong Composition, BASIC, which fully leverages existing VLM capabilities and remains adaptable to future advances.

BASIC separately computes the similarity with respect to each query component and performs fusion inspired by the classical Harris corner principle [15]; both responses must be jointly high, rather than just one of the two. The image-to-image dot product similarity is enhanced through a projection computed analytically not in the image space, but in the text representation space. This is facilitated by a large language model (LLM) that provides common object names and typical textual modifications. The aim is to increase distinctiveness regarding object variations, *i.e.* to better represent image objects beyond other visual cues, while achieving invariance to textually described modifications, so that the same object is retrieved despite variations. Interestingly, this projection can also be equivalently applied solely on the query side, enabling user-specific or application-specific customization. The text-to-image dot product similarity is refined via a query-time contextualization process designed to bridge the distribution gap between the text inputs seen during VLM pre-training and the text queries used at inference. Our contributions are summarized as follows:

- We introduce *i*-CIR, a new evaluation dataset for CIR, meant to retrieve images containing the same particular object as the visual query under modifications defined by the text query.

- We introduce BASIC, a training-free approach leveraging pre-trained VLMs for class-level or instance-level CIR that is based on image-to-image and text-to-image similarities, without the need to update the database embeddings.

- BASIC sets a new state of the art on *i*-CIR and across existing class-level CIR datasets.

# 2 Related work

**Methods.** While early methods like TIRG [51], CIRPLANT [30] and CLIP4CIR [1] rely on supervised training with annotated triplets, recent efforts in zero-shot CIR (ZS-CIR) avoid triplet supervision and fall into three main categories. *Textual-inversion* methods (*e.g.* Pic2Word [44], SEARLE [2], ISA [8], LinCIR [14]) map the reference image to a pseudo-text token, which is then

composed with the modification text in the language domain and processed by a vision-language model. *Pseudo-triplet* approaches (*e.g.* TransAgg [29], HyCIR [21], CompoDiff [13], CoVR-2 [50], MCL [27]) generate synthetic training data using LLMs [28] and image generative models [42], either from caption-editing strategies or from natural web-based image pairs. *Training-free methods* (*e.g.* WeiCom [39], FreeDom [9], CIReVL [23], GRB [47], WeiMoCIR [55]) leverage off-the-shelf VLMs [41, 25, 26] and LLMs [49] to perform CIR without any additional training by either recasting it as text-based retrieval or fusing visual and textual embeddings directly via weighted sums or geometric interpolations.

**Datasets.** Key benchmarks include FashionIQ [54] (77k images, 30k triplets across three fashion sub-tasks) and CIRR [30] (22k images, 37k triplets), both criticized for label ambiguity, high false-negative rates, and text-only shortcuts [2, 18], and recently refined by [18]. CIRCO [2] (1k queries over 120k COCO-unlabeled distractors, 4.53 targets/query) extends this paradigm with more diverse negatives. Four additional domain-conversion datasets-ImageNet-R [16] (30k stylized images of 200 classes in four style domains), MiniDomainNet [62] (140k images of 126 classes in four domains), NICO++ [61] (89k images of 60 categories in six contexts), and LTLL [10] (500 images of 25 locations)-explore class-level retrieval under style or context shifts. Concurrent to our work, ConCon-Chi [43] introduces an image–caption benchmark for personalized concept–context understanding, designed for text-to-image generation, retrieval, and editing. Their *concepts* correspond to our *instances*, while their *contexts* parallel the modifications expressed by our *text queries*.

## 3  *i*-CIR dataset

### 3.1  Overview and structure

We introduce the *i*-nstance-level Composed Image Retrieval (*i*-CIR) evaluation dataset. Following the instance-level class definition [40, 56], we group all visually indistinguishable objects, *i.e.* the same particular object, into a single class. For example, a class may correspond to (i) a concrete physical entity, such as the Temple of Poseidon, or (ii) a fictional yet visually distinctive character or object, such as Batman. In practice, if a human observer can confidently state that multiple visual manifestations represent the *same object*, they belong to the same instance-level class.

Given a *composed query* $(q^v, q^t)$ consisting of a visual query $q^v$ depicting a particular object, also referred to as an object instance or simply *instance*, and a text query $q^t$ describing a modification, the goal is to rank a database of images such that those depicting the same instance under the requested modification appear at the top. We refer to these images as *composed positives* or simply *positives*. For each composed query, we consider the following types of hard negative images: (i) *visual hard negative*: depicts an identical or visually similar object as $q^v$ but does not match the textual modification $q^t$, (ii) *textual hard negative*: matches the semantics of $q^t$ but depicts a different instance, typically from the same semantic category, (iii) *composed hard negative*: nearly matches both query parts, while one of the two may be identically matched, *i.e.* depicts an object similar/identical to $q^v$ with semantics similar to $q^t$, or an object similar to $q^v$ with semantics identical/similar to $q^t$. Examples are shown in Figure 1 and Figure 3.

All types of negatives, including non-hard ones, are treated equally during evaluation. However, we include a significant number of hard negatives in our dataset to create a challenging yet manageable benchmark that supports future research. There are $n^v$ image queries for the same instance combined with $n^t$ text queries that are combined to construct $n^v n^t$ composed queries (values of $n^v$ and $n^t$ vary per instance). Unlike typical retrieval benchmarks that use a single common database for all queries, we employ the same image database for all $n^v n^t$ composed queries of an instance, but a different database for queries of other instances. This design ensures scalable and error-free labeling, avoiding the impracticality of verifying each database image as positive or negative for every query.

### 3.2  Collection and curation

The dataset construction process combines human input with automated image retrieval[2]. Our aim is to curate, for each instance, composed queries, sets of corresponding positives, and a well-structured set of challenging hard negatives, with all *i*-CIR images sourced from the LAION [46] dataset.

---

[2]We perform *image-to-image* and *text-to-image* retrieval using dot product search based on image and text representations obtained from OpenAI CLIP [41].

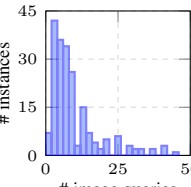 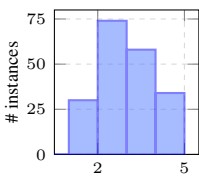 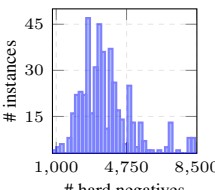 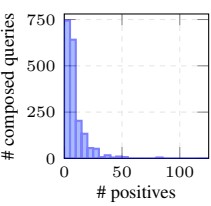

Figure 2: i-*CIR statistics*. From left to right: Number of (a) image queries, (b) text queries, and (c) hard negatives per instance; (d) positives per composed query.

The process for each instance begins by defining the object instance, *e.g.* Temple of Poseidon, and selecting semantically meaningful modifications, *e.g.* ''at sunset'', while avoiding implausible ones, *e.g.* ''with snow''. We then create *seed images* and *seed sentences* to serve as queries for retrieving neighbors from LAION, which collectively form a *candidate image pool* that includes potential queries, positives, and hard negatives.

**Seed images:** 2 to 5 high-quality images depicting (i) the object instance, *e.g.* the Temple of Poseidon, or (ii) a composed positive, *e.g.* the Temple of Poseidon at sunset. These images are gathered from web searches in Creative Commons repositories or personal photo collections. The neighbors retrieved from LAION are categorized as visual and composed hard negatives for cases (i) and (ii), respectively.

**Seed sentences:** Textual descriptions of (i) the instance (*e.g.* ''Temple of Poseidon''), (ii) another object of the same category (*e.g.* ''Ancient Greek Temple''), (iii) rephrased versions of defined modifications (*e.g.* ''a photo of dusk''), (iv) the instance under the modifications (*e.g.* ''Temple of Poseidon at sunset''), (v) an object of the same semantic category under the modification (*e.g.* ''an Ancient Greek Temple at sunset''). The neighbors retrieved from LAION are classified as visual, textual, and composed hard negatives for cases (i & ii), (iii), and (iv & v), respectively.

After building the candidate image pool, automated filtering removes low-resolution, watermarked, or duplicate content using perceptual hashing and resolution checks. Annotators then manually inspect the remaining images to identify composed *positives* per composed query. Unmarked images are considered *negatives*. *Visual hard negatives* are associated with all composed queries of an instance, while *textual* and *composed hard negatives* are associated only with the specific composed query from which (or from whose text query) they were derived. Finally, annotators manually select images within the visual hard negatives to serve as *image queries*. All images that were neither filtered out from the candidate image pool nor selected as queries form the database for this instance. Positives and hard negatives associated with a composed query are negatives for another composed query.

To avoid bias towards/against CLIP-based methods, seed images are discarded and not included in *i*-CIR, while seed sentences do not include the exact phrasing of a text query.

### 3.3   Statistics and visualisations

Figure 2 summarizes per-instance and per-query statistics in *i*-CIR. We include 202 object instances and 750 K images in total. Each instance has 1–46 image queries (195 with >1, median: 6) and 1–5 text modifications (median: 2), yielding 1,883 composed queries overall. Queries can be categorized either by their visual part (the object instance) or by their textual part (the modification). Each composed query has 1–127 positives (median: 5) and each instance's database contains 951–10,045 hard negatives (median: 3,420), creating a challenging retrieval benchmark. Figure 3 illustrates a set of randomly chosen pairings from the categorization: for each of eight visual–textual category combinations, we show the image query, composed positive, visual hard negative, textual hard negative, and composed hard negative. These visualisations highlight the rich diversity of *i*-CIR, both in terms of the wide array of visual categories (*e.g.* landmarks, products, fictional characters, tech gadgets) and the broad spectrum of textual modifications (*e.g.* viewpoints, attributes, contexts, additions), setting *i*-CIR apart from existing CIR datasets. More info in the Appendix subsection C.1.

### 3.4   Shortcomings of existing benchmarks

Commonly used CIR datasets include CIRR [30], FashionIQ [54], CIRCO [2], and ImageNet-R [16]. CIRR, FashionIQ, and CIRCO share a common limitation: their construction relies on an automated process to select two similar images, guided by either textual or visual similarity. These images form the image query and the positive pair. Due to the nature of the image sources, either there is no obvious relation of such selected pairs or the relationship is typically at semantic level only,

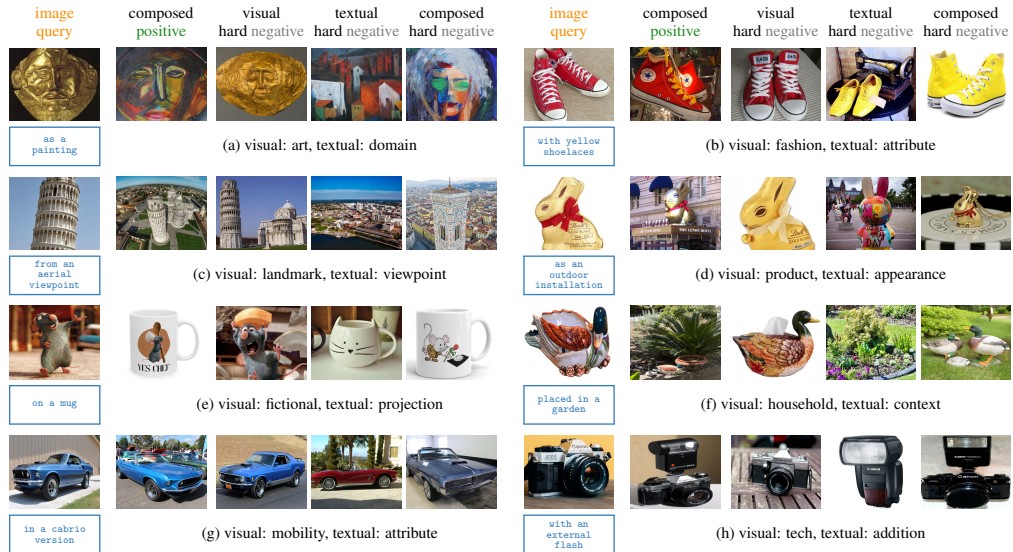

Figure 3: *Visualization of visual and textual category examples from* i-*CIR. For each of the eight randomly chosen category pairings (a–h), we display:* the image query, the text query, the composed positive, the visual hard negative, the textual hard negative, and the composed hard negative.

rather than at instance level. Subsequently, a language-based description is generated to capture the difference between the two images. However, the lack of concrete differences between the images, often coupled with their low relevance, results in descriptions that are either poor representations of meaningful text queries or inadequate components of a composed query. In many cases, the text query alone suffices to describe the positive image (Figure 6), making the image query redundant. Moreover, these datasets exhibit a paucity of challenging negatives and a substantial rate of false negatives in their ground-truth annotations [2], inflating reported performance. We present such cases in the Appendix subsection C.2.

Domain-conversion benchmarks such as ImageNet-R [16], NICO++ [61], and MiniDomainNet (MiniDN) [62] extend CIR to style or context shifts (*e.g.* ''photo''→''cartoon'') but define positives by semantic class membership rather than object identity, lacking instance-level granularity. LTLL [10] is the sole existing instance-level domain-conversion dataset, but it is extremely limited in scale (500 images of 25 locations, two domains) and provides only binary ''archive'' *vs.* ''today'' modifications. Furthermore, these benchmarks offer very narrow categorical variation—FashionIQ is confined to fashion items, LTLL to a two-way temporal shift, and the domain sets to domain changes only. These semantic-level definitions, small scale, minimal textual variation, and weak negative mining in prior benchmarks motivate the creation of *i*-CIR.

## 4 A surprisingly strong baseline

In the task of *composed image retrieval* (CIR), we are given an image query $q^v \in \mathcal{X}^v$ and a text query $q^t \in \mathcal{X}^t$, where $\mathcal{X}^v$ is the image input space and $\mathcal{X}^t$ is the text input space. The goal is to retrieve images $x^v$ from a database $X = \{x_1^v, \dots, x_n^v\} \subset \mathcal{X}^v$ that are visually relevant to the image query and reflect the modifications specified by the text query. Features are extracted using a pre-trained visual encoder $\phi^v : \mathcal{X}^v \to \mathbb{R}^d$ and text encoder $\phi^t : \mathcal{X}^t \to \mathbb{R}^d$, *e.g.* CLIP [41], which map image and text queries to a shared embedding space of dimension $d$. Image-to-image and image-to-text similarities are computed via dot product of the corresponding features.

The proposed training-free method, called BASIC, is based on the assumption that both modalities in the composed retrieval query encode complementary information that jointly contribute to the retrieval objective. This makes the composed retrieval task analogous to performing a logical conjunction over the two modalities: we seek results that are simultaneously relevant to both the image and the text [3].

---

[3]While this assumption holds well for standard composite tasks (*e.g.* this image and the concept "winter"), it may not apply in tasks where one modality dominates (*e.g.* purely textual transformations) or where the text query is highly entangled with image content (*e.g.* CIRR-like datasets). In such cases, the benefits of the proposed mechanisms may diminish.

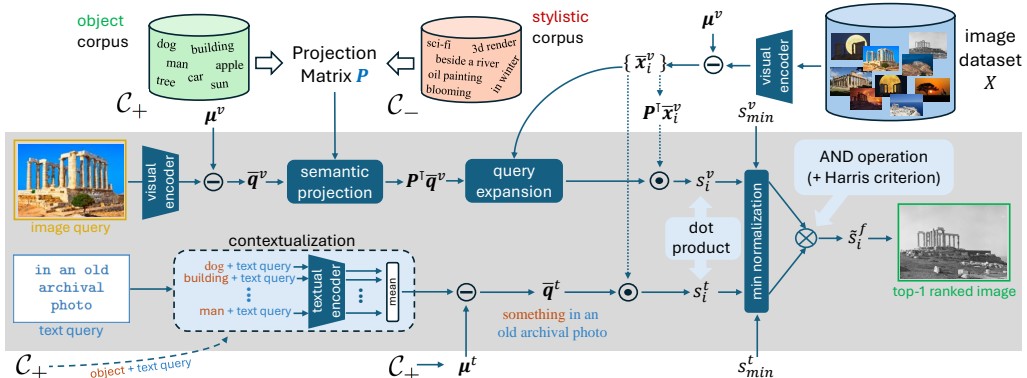

Figure 4: *Overview of our training-free composed image retrieval method* BASIC. Given a query image and text, we apply centering and semantic projection, guided by corpora $C+$ and $C-$, to suppress irrelevant dimensions. The text is contextualized using caption-like prompts. Both modalities are scored against the database with min-based normalization and fused via a multiplicative "AND" operation regularized by a Harris-like criterion to retrieve jointly relevant results.

Following the aforementioned assumption, we compute the similarity per modality and then combine them. We improve the representation per modality by removing modality-specific noise and spurious correlations that can interfere with their effective combination. In practice, visual features may be entangled with background clutter, image composition, or dataset-specific styles, whereas textual features can reflect lexical biases or corpus-level drift. Figure 4 presents a high-level overview of BASIC, which consists of a sequence of conceptually simple yet effective steps that progressively filter out the aforementioned noise and modality-specific artifacts from the image and text features.

**Centering for bias removal.** We remove modality-specific biases by subtracting mean features. These means typically capture low-level regularities unique to each modality: the average image feature reflects general visual patterns, while the average text feature captures common linguistic patterns. Subtracting them helps isolate semantic content from distributional bias. In particular, after extracting the image features $\mathbf{q}^v = \phi^v(q^v) \in \mathbb{R}^d$ and text features $\mathbf{q}^t = \phi^t(q^t) \in \mathbb{R}^d$, also from the database images, we subtract a precomputed image feature mean $\boldsymbol{\mu}^v \in \mathbb{R}^d$, and a text feature mean $\boldsymbol{\mu}^t \in \mathbb{R}^d$, respectively. The centered features are

$$\bar{\mathbf{q}}^v = \mathbf{q}^v - \boldsymbol{\mu}^v = \phi^v(q^v) - \boldsymbol{\mu}^v \quad \text{and} \quad \bar{\mathbf{q}}^t = \mathbf{q}^t - \boldsymbol{\mu}^t = \phi^t(q^t) - \boldsymbol{\mu}^t. \tag{1}$$

To ensure scalability and generalization, we compute $\boldsymbol{\mu}^v$ using a large external dataset $X^v$, *e.g.* LAION [46]. Similarly, we calculate the text mean $\boldsymbol{\mu}^t$ on a predefined textual corpus which contains content-relevant concepts (see next step).

**Projection onto semantic subspace.** We aim to transform the image features to retain information related to the main objects, while suppressing information related to image styles, object domains, or background setting, *i.e.* that correspond to text query modifications of common use cases. This is achieved by projecting into a lower-dimensional subspace derived from text CLIP features. To construct this projection, we use two textual corpora: $C_+$, a *object corpus* containing object-oriented terms (*e.g.* "building", or "dog"), and $C_-$, a *stylistic corpus* containing terms related to style, viewing conditions or contextual setting (*e.g.* "cartoon", "aerial view", or "in a cloudy day"). Inspired by [37], we compute a weighted contrastive covariance matrix as follows :

$$\mathbf{C} = (1 - \alpha)\mathbf{C}_+ - \alpha\mathbf{C}_-, \quad \text{where} \quad \mathbf{C}_\pm = \frac{1}{|C_\pm|} \sum_{x \in C_\pm} (\phi^t(x) - \boldsymbol{\mu}^t)(\phi^t(x) - \boldsymbol{\mu}^t)^\top \tag{2}$$

and $\alpha$ is an empirically determined hyperparameter. We extract the top-$k$ eigenvectors of $\mathbf{C} \in \mathbb{R}^{d \times d}$ to form a projection matrix $\mathbf{P} \in \mathbb{R}^{d \times k}$. The eigenvectors capture directions with high variance in $\mathbf{C}_+$ and small in $\mathbf{C}_-$, emphasizing object-specific cues while suppressing style-related variation. We then project the centered image features $\bar{\mathbf{x}}^v = \phi(x^v) - \boldsymbol{\mu}^v \in \mathbb{R}^d$, either query or database, as $\mathbf{P}^\top \bar{\mathbf{x}}^v \in \mathbb{R}^k$. Note that the corpora $C_+$ and $C_-$ need not match the domain of the retrieval database. Even generic corpora for $C_+$, *e.g.* class names from ImageNet-1K [6], yield performance improvements, as the captured directions are semantically rich and broadly transferable.

**Image query expansion.** In the literature, image retrieval performance, recall in particular, has been shown to be significantly improved by query expansion [3, 12]. The proposed method benefits from

applying the *optional* step of query expansion using the image query. Following [12], the original feature of the image query is enhanced by a weighted combination of the top-ranked database features that it retrieves. The weights are an increasing function of the corresponding similarities.

**Contextualization of text queries.** CLIP is trained primarily on natural language captions and full sentences. As a result, using single-word text queries (*e.g.* ``sculpture'') or sentence parts (*e.g.* ``during sunset'') constitute out-of-distribution input and may produce text features that are not well-aligned with image features. To address this, we introduce a *contextualization* step that enriches such textual queries with additional terms. Let $q^t$ be a raw text query (*e.g.* ``sculpture''). We generate multiple caption-like queries by combining $q^t$ with elements from the subject corpus $C_+$. We add a random set of terms before (*e.g.* ``dog during the sunset'') and after (*e.g.* ``sculpture dog'') the text query. These composed phrases are embedded using CLIP's text encoder, centered, and averaged. This operation yields a more robust textual feature that better reflects how CLIP interprets concepts in natural language (*e.g.* ``[something] during the sunset'').

**Score normalization and fusion.** The final step is to combine similarities from the two modalities to rank the database items. Given the centered image query embedding $\bar{\mathbf{q}}^v \in \mathbb{R}^d$ and the contextualized centered text query (either original, or expanded) embedding $\bar{\mathbf{q}}^t \in \mathbb{R}^d$, we compute similarities to the centered embedding $\bar{\mathbf{x}}^v \in \mathbb{R}^d$ of a database image $x^v \in X$ as:

$$s^v = \left\langle \mathbf{P}^\top \bar{\mathbf{x}}^v, \mathbf{P}^\top \bar{\mathbf{q}}^v \right\rangle \quad \text{and} \quad s^t = \left\langle \bar{\mathbf{x}}^v, \bar{\mathbf{q}}^t \right\rangle. \tag{3}$$

To reflect the complementary nature of the modalities, we fuse the two scores by multiplication: $s = s^v s^t$. However, due to modality imbalance and differences in representation ranges, one modality can disproportionately dominate the final score.

*Min-based normalization.* To mitigate range imbalances, an affine re-scaling of the similarities $s$ in each modality is performed. The empirical minimum $s_{\min} < 0$ of the dot product in (3) is used, so that $s_{\min}$ is mapped to 0 and 0 is mapped to 1:

$$\tilde{s} = (s - s_{\min})/|s_{\min}|.$$

We apply this to both $s^v$ and $s^t$ using predefined statistics for $s^v_{\min}$ and $s^t_{\min}$, estimated on an external dataset.

*Fused similarity with Harris criterion.* Finally, we fuse the normalized scores using multiplication and a regularizer inspired by the Harris corner detector. The final score is:

$$\tilde{s}^f = \tilde{s}^v \tilde{s}^t - \lambda(\tilde{s}^v + \tilde{s}^t)^2.$$

The first term rewards items that are jointly relevant to both modalities. The second term penalizes items where only one modality is highly activated, thereby suppressing false positives from unbalanced queries. The scalar $\lambda$ controls the trade-off and is fixed across all experiments.

**Computational complexity.** The strengths of our approach stem from its simplicity and efficiency. The entire pipeline is (deep-network) training-free and is composed of operations that scale linearly or sub-linearly with the dataset size, since similarity computation over the dataset items is a simple inner product and can be efficiently handled by existing libraries, *e.g.* FAISS [22]. The proposed similarity computation efficiently operates over a stored database of original CLIP representations. The similarity $s^v$ is efficiently computed as follows

$$s^v = \left\langle \mathbf{P}^\top(\mathbf{x}^v - \boldsymbol{\mu}^v), \mathbf{P}^\top(\mathbf{q}^v - \boldsymbol{\mu}^v) \right\rangle = \left\langle \mathbf{x}^v, \mathbf{P}\mathbf{P}^\top(\mathbf{q}^v - \boldsymbol{\mu}^v) \right\rangle - \left\langle \boldsymbol{\mu}^v, \mathbf{P}\mathbf{P}^\top(\mathbf{q}^v - \boldsymbol{\mu}^v) \right\rangle,$$

where the first term is a dot-product computed over the unaltered database features and the second term is a query dependent constant. Thus, all computation related to centering and projection can be computed on-the-fly on the query side. This is valuable, since the mean and the projection matrix can be alternated (*e.g.* with specific knowledge of the domain) without touching the stored index. This makes our method particularly well-suited for large-scale deployments, requiring no adaptation, no fine-tuning, and no backpropagation. See the Appendix D for more BASIC-related technical details.

## 5 Experiments

### 5.1 Experimental setup

**Datasets and evaluation protocol.** We evaluate BASIC on our proposed *i*-CIR as well as four composed image retrieval benchmarks: ImageNet-R, MiniDN, NICO++, and LTLL. Retrieval

performance is measured using the standard mean Average Precision (mAP) metric. Average Precision (AP) is computed per query by averaging the precision values at the ranks of all relevant items in the retrieval list. The mean Average Precision (mAP) is then obtained by averaging AP over all queries, providing a global measure of retrieval effectiveness that accounts for the order of relevant results. For *i*-CIR, we report the *macro-mAP* over instances, defined by first computing mAP per instance and then taking the mean of these per-instance mAPs across all instances.

**Baselines and competitors.** We include four simple baselines. "Text" scores each database image $x^v \in X$ by $\langle \phi^t(q^t), \phi^v(x^v) \rangle$; "Image" scores by $\langle \phi^v(q^v), \phi^v(x^v) \rangle$; "Text + Image" combines the similarities by summation; "Text × Image" by product. We also benchmark BASIC against state-of-the-art zero-shot composed image retrieval methods: WeiCom [39], Pic2Word [44], CompoDiff [13], CIReVL [23], SEARLE [2], MCL [27], MagicLens [60], CoVR-2 [50], and FreeDom [9]. All methods use CLIP with ViT-L/14 [7], whereas CompoDiff employs the larger CLIP ViT-G/14.

**BASIC.** For fair comparison, we also use CLIP [41] ViT-L/14 [7]. We set $k = 250$ components for PCA, $\lambda = 0.1$ for the Harris criterion and $\alpha = 0.2$. These values were fixed once on a small privately owned development set, named *i*-CIR $_{\text{dev}}$. The corpora $C_+$ and $C_-$ were automatically generated using ChatGPT [17]. The statistics $s_{\min}^v$ and $s_{\min}^t$ were computed over a synthetically generated dataset constructed using Stable Diffusion [42] with automatically created prompts. More details are included in the Appendix D and Appendix subsection E.1.

## 5.2 Experimental results

**Per-category performance.** In Figure 5 we report the per-category performance of selected baselines and competitors on *i*-CIR split by the a) primary visual and (b) textual categories of the queries.

In Figure 5(a), BASIC ranks first in six of the eight visual categories, delivering particularly large margins on fictional (47.8% *vs*. 31.1% for SEARLE), mobility (45.8% *vs*. 29.3% for MagicLens), and technology (30.6% *vs*. 23.0% for Text × Image). It also leads on product (33.7% *vs*. 26.7% for MagicLens), landmark (39.3% *vs*. 35.0% for MagicLens), and art (38.0% *vs*. 35.0% for MagicLens). The only exceptions are fashion, where MagicLens edges out at 25.6% *vs*. 22.0% for BASIC, and household, where MagicLens peaks at 29.1%; BASIC is second at 22.4%. In contrast, the other methods show uneven strengths.

Figure 5(b) further confirms the consistency of BASIC. BASIC dominates projection (53.1% *vs*. 31.1% for MagicLens), appearance (48.8% *vs*. 36.8% for SEARLE), and domain (39.3% *vs*. 31.1% for MagicLens). It also leads on vewpoint (47.8% *vs*. 40.1% for MagicLens) and attribute (26.3% *vs*. 24.1% for MagicLens). BASIC is second on context (35.6% *vs*. 36.4% for MagicLens) and addition (24.0% *vs*. 28.2% for MagicLens).

**Comparison with SOTA.** We further evaluate the performance of BASIC against all considered baselines and state-of-the-art CIR methods across five datasets, including *i*-CIR. Results are shown in Table 1. As observed, BASIC consistently outperforms all competing methods across the board. Runtime comparisons are provided in the Appendix subsection E.6.

Table 1: Average mAP (%) comparison across datasets. $^\dagger$: without query expansion.

| Method | ImageNet-R | NICO++ | MiniDN | LTLL | *i*-CIR |
|---|---|---|---|---|---|
| Text | 0.74 | 1.09 | 0.57 | 5.72 | 3.01 |
| Image | 3.84 | 6.32 | 6.66 | 16.49 | 3.04 |
| Text + Image | 6.21 | 9.30 | 9.33 | 17.86 | 8.20 |
| Text × Image | 7.83 | 9.79 | 9.86 | 23.16 | 17.48 |
| WeiCom | 10.47 | 10.54 | 8.52 | 26.60 | 18.03 |
| Pic2Word | 7.88 | 9.76 | 12.00 | 21.27 | 19.36 |
| CompoDiff | 12.88 | 10.32 | 22.95 | 21.61 | 9.63 |
| CIReVL | 18.11 | 17.80 | 26.20 | 32.60 | 18.66 |
| SEARLE | 14.04 | 15.13 | 21.78 | 25.46 | 19.90 |
| MCL | 8.13 | 19.09 | 18.41 | 16.67 | 19.89 |
| MagicLens | 9.13 | 19.66 | 20.06 | 24.21 | 27.35 |
| CoVR-2 | 11.52 | 24.93 | 27.76 | 24.68 | 28.50 |
| FreeDom | 29.91 | 26.10 | 37.27 | 33.24 | 17.24 |
| FreeDom $^\dagger$ | 25.81 | 23.24 | 32.14 | 30.82 | 15.76 |
| BASIC | **32.13** | **31.65** | **39.58** | **41.38** | 31.64 |
| BASIC $^\dagger$ | 27.54 | 28.90 | 35.75 | 38.22 | **34.35** |

**Note.** For *i*-CIR we report *macro-mAP*

## 5.3 Ablation studies

**BASIC components.** Table 2 presents a detailed ablation study on the contribution of each component of BASIC across all evaluated datasets. Starting with a simple Text×Image baseline, we progressively add the components of BASIC, highlighting the cumulative benefits of each module.

*Centering* provides a notable boost across most datasets (17.48% → 28.33% on *i*-CIR) with the exception of LTLL, likely due to its narrow focus on landmarks. *Normalization and Harris fusion* further enhance retrieval, as demonstrated by their removal, with min normalization being especially

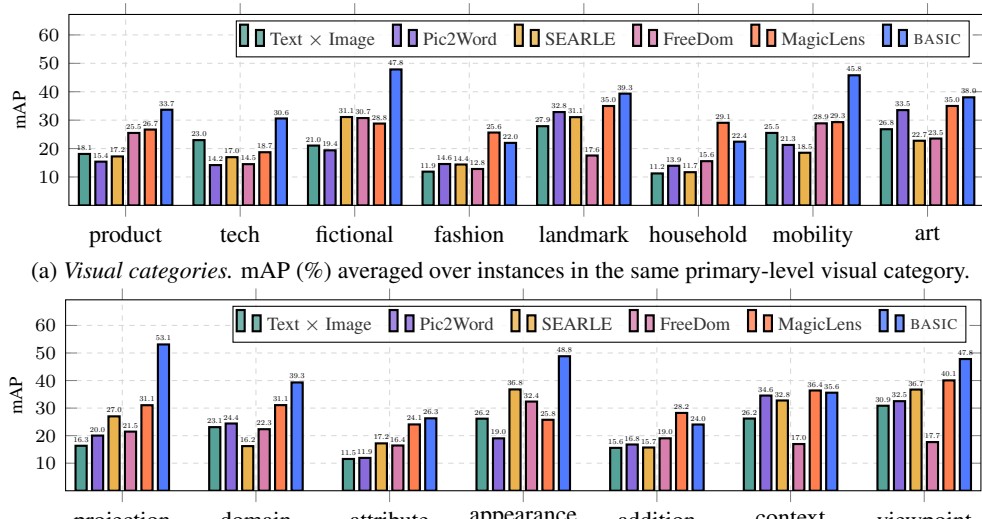

(a) *Visual categories.* mAP (%) averaged over instances in the same primary-level visual category.

(b) *Textual categories.* mAP (%) averaged over instances in the same primary-level textual category.

Figure 5: *Performance comparison on* i-*CIR per primary category of queries.* (a) Visual, (b) Textual.

Table 2: Ablation study reporting average mAP (%) across datasets. Each row progressively adds or removes components of the proposed method: mean centering (*Centering*), min-based normalization (*Min Norm.*), Harris criterion (*Harris*), text contextualization (*Context.*), semantic projection (*Proj.*), and query expansion (*Q. Exp.*). The first row (no component applied) corresponds to Text × Image.

| Centering | Min Norm. | Harris | Context. | Proj. | Q. Exp. | ImageNet-R | NICO++ | MiniDN | LTLL | i-CIR |
|---|---|---|---|---|---|---|---|---|---|---|
| ✗ | ✗ | ✗ | ✗ | ✗ | ✗ | 7.66 | 9.26 | 9.48 | 19.78 | 17.48 |
| ✓ | ✗ | ✗ | ✗ | ✗ | ✗ | 12.16 | 9.95 | 12.16 | 16.93 | 28.33 |
| ✓ | ✓ | ✗ | ✗ | ✗ | ✗ | 12.06 | 17.20 | 17.72 | 22.20 | 27.30 |
| ✓ | ✓ | ✓ | ✗ | ✗ | ✗ | 16.21 | 15.06 | 17.79 | 29.70 | 28.42 |
| ✓ | ✓ | ✓ | ✓ | ✗ | ✗ | 18.61 | 15.34 | 21.01 | 33.74 | 33.48 |
| ✓ | ✓ | ✓ | ✓ | ✓ | ✗ | 27.54 | 28.90 | 35.75 | 38.22 | **34.35** |
| ✓ | ✓ | ✓ | ✓ | ✗ | ✓ | 17.31 | 13.96 | 21.22 | 22.42 | 31.78 |
| ✓ | ✓ | ✓ | ✗ | ✓ | ✓ | 26.18 | 30.61 | 33.64 | 34.50 | 25.85 |
| ✓ | ✓ | ✗ | ✓ | ✓ | ✓ | 30.75 | 29.82 | 38.85 | 40.65 | 31.61 |
| ✓ | ✗ | ✗ | ✓ | ✓ | ✓ | 24.50 | 22.74 | 29.65 | 19.36 | 30.75 |
| ✓ | ✓ | ✓ | ✓ | ✓ | ✓ | **32.13** | **31.65** | **39.58** | **41.38** | 31.64 |

critical, since its absence causes a significant drop. Harris consistently contributes moderate gains. *Text contextualization* is also important. Its removal results in a substantial performance decline, particularly on datasets requiring nuanced language understanding (31.64% → 25.85% on *i*-CIR). On the image side, *semantic projection* accounts for the majority of the performance gain in many cases, serving as a key enhancement. *Query expansion* offers additional improvements, particularly on category-level datasets, though it leads to performance decrease in *i*-CIR. Note that some components depend on the presence of others to be effective (*e.g.* projection assumes centered features, Harris step requires min-normalized scores).

Table 3: mAP(%) on each dataset using different negative corpora. The first column lists the evaluation datasets.

| Eval. Dataset | Negative Corpora Source | | | | | |
|---|---|---|---|---|---|---|
| | none | generic | Imagenet-R | NICO++ | MiniDN | LTLL |
| Imagenet-R | 30.15 | 32.13 | **33.22** | 30.74 | 31.17 | 30.91 |
| NICO++ | 30.67 | **31.65** | 30.84 | 31.17 | 30.67 | 31.20 |
| MiniDN | 38.64 | 39.38 | 39.34 | 38.75 | **39.58** | 38.68 |
| LTLL | 41.80 | 41.24 | 41.33 | 43.39 | 42.09 | **43.98** |
| i-CIR | 31.51 | **31.64** | 31.61 | 31.20 | 31.32 | 31.06 |

**Controlling semantic projection.** Table 3 shows the effect of omitting $C_-$, using a generic negative corpus, or using a dataset-specific corpus (generated via ChatGPT) designed to reflect the domain variability of ImageNet-R, NICO++, MiniDN, and LTLL. Results indicate that leveraging application-related knowledge can improve performance, particularly compared to omitting $C_-$. This idea is further discussed in the Appendix subsection E.5.

*i*-**CIR: Compact but hard.** We use randomly selected images from LAION as negatives to assess how challenging *i*-CIR is in comparison to a large-scale database that is commonly shared across all queries and lacks explicit hard negatives. Using the performance of Text × Image baseline as a reference (17.48%), we find that more than 40M distractor images are required for this baseline to

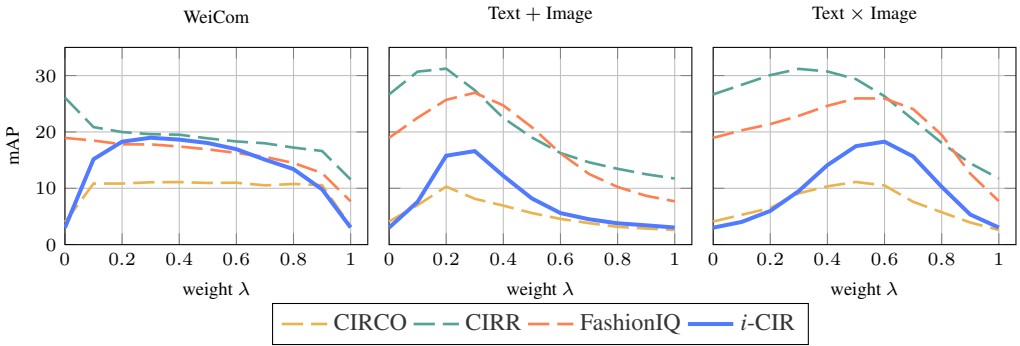

Figure 6: *Modality bias via weight sweeps.* mAP(%) *vs.* fusion weight $\lambda$ for three simple methods, where $\lambda=0$ is text-only and $\lambda=1$ is image-only. Compositional datasets should peak at interior $\lambda$ and exceed both endpoints. *i*-CIR shows strong interior optima and large gains over the best uni-modal baseline; CIRR and FashionIQ peak at $\lambda=0$, indicating text dominance.

reach a similarly low performance. Note that the performance measured using unlabeled LAION images as negatives is only a lower bound because of the inevitable presence of false negatives. This is four orders of magnitude larger than the 3.7K database images per query, on average, that *i*-CIR uses, or 1.5 orders of magnitude larger than the 750K database images used among all queries; the latter defines the experimental processing cost. More analysis is provided in the Appendix subsection E.7.

***i*-CIR: Truly compositional.** A dataset that requires *composition* should reward combining text and image, not either modality alone. To diagnose this, we sweep a mixing weight $\lambda \in [0, 1]$ between text-only ($\lambda=0$) and image-only ($\lambda=1$) similarity for three simple fusion methods (WeiCom, Text $+$ Image, Text $\times$ Image), and plot mAP as a function of $\lambda$ (Figure 6). For each method we compute the *composition gain* $\Delta$: the difference between the peak value of the curve and the best uni-modal endpoint, and then average $\Delta$ across the three methods. On *i*-CIR, the average composition gain is large: $+14.9$ mAP ($+490\%$ relative to the best uni-modal baseline), with peaks occurring at interior $\lambda$—clear evidence that both modalities must work together. By contrast, it shrinks to $+3.0$ mAP ($+11\%$) on CIRR, $+5.0$ mAP ($+26\%$) on FashionIQ, and $+6.8$ mAP ($+167\%$) on CIRCO. Moreover, the highest uni-modal performance of CIRR and FashionIQ is always when ($\lambda = 0$), *i.e.* text-only. Thus legacy datasets reward composition only marginally, whereas *i*-CIR demands genuine cross-modal synergy; BASIC is designed for the latter scenario.

## 6    Conclusions

We introduced *i*-CIR, an instance-level benchmark for composed image retrieval with *explicit hard negatives* (visual, textual, and composed). It fills a long-standing gap by providing an ambiguity-free evaluation suite that rewards composition rather than single-modality shortcuts. We also presented BASIC, a simple, efficient, *training-free* method that compares favorably to both training-based and training-free baselines across benchmarks. BASIC is built from a few transparent components, whose combination delivers strong accuracy, transfers well, and exhibits broad hyperparameter plateaus. We hope *i*-CIR becomes a reliable target for assessing genuinely compositional retrieval, and that the simplicity of BASIC catalyzes adoption and further advances.

**Acknowledgments.**    This work was supported by the Junior Star GACR GM 21-28830M; the Czech National Recovery Plan—CEDMO 2.0 NPO (MPO 60273/24/21300/21000) provided by the Ministry of Industry and Trade; the EU Horizon Europe programme MSCA PF RAVIOLI (No. 101205297) and HERON-Hellenic Robotics Center of Excellence (No. 101136568); the National Recovery and Resilience Plan "Greece 2.0"/NextGenerationEU project "Applied Research for Autonomous Robotic Systems" (MIS5200632); and Czech Technical University in Prague (SGS23/173/OHK3/3T/13 and institutional Future Fund). We acknowledge VSB – Technical University of Ostrava, IT4Innovations National Supercomputing Center, Czech Republic, for awarding this project (OPEN-33-67) access to the LUMI supercomputer, owned by the EuroHPC Joint Undertaking, hosted by CSC (Finland) and the LUMI consortium, through the Ministry of Education, Youth and Sports of the Czech Republic via the e-INFRA CZ project (ID: 90254). We also acknowledge the OP VVV project "Research Center for Informatics" (CZ.02.1.01/0.0/0.0/16 019/0000765).

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

# NeurIPS Paper Checklist

1. **Claims**

   Question: Do the main claims made in the abstract and introduction accurately reflect the paper's contributions and scope?

   Answer: [Yes]

   Justification: The abstract and introduction accurately summarize our key contributions: the creation of an instance-level composed image retrieval dataset (*i*-CIR) with challenging hard negatives, the proposal of a training-free CIR method (BASIC) that leverages existing VLMs, and the demonstration of state-of-the-art performance on *i*-CIR and multiple benchmarks. All claims are supported by detailed methodology and experiments in Sections 3–5, as also in supplementray material.

   Guidelines:
   - The answer NA means that the abstract and introduction do not include the claims made in the paper.
   - The abstract and/or introduction should clearly state the claims made, including the contributions made in the paper and important assumptions and limitations. A No or NA answer to this question will not be perceived well by the reviewers.
   - The claims made should match theoretical and experimental results, and reflect how much the results can be expected to generalize to other settings.
   - It is fine to include aspirational goals as motivation as long as it is clear that these goals are not attained by the paper.

2. **Limitations**

   Question: Does the paper discuss the limitations of the work performed by the authors?

   Answer: [Yes]

   Justification: We explicitly acknowledge in Sec. 3 (footnote) that our logical-AND assumption may break down when one modality dominates (e.g., pure style edits or highly entangled image–text pairs like CIRCO). We point readers to additional method and dataset limitations in the supplementary material.

   Guidelines:

   - The answer NA means that the paper has no limitation while the answer No means that the paper has limitations, but those are not discussed in the paper.
   - The authors are encouraged to create a separate "Limitations" section in their paper.
   - The paper should point out any strong assumptions and how robust the results are to violations of these assumptions (e.g., independence assumptions, noiseless settings, model well-specification, asymptotic approximations only holding locally). The authors should reflect on how these assumptions might be violated in practice and what the implications would be.
   - The authors should reflect on the scope of the claims made, e.g., if the approach was only tested on a few datasets or with a few runs. In general, empirical results often depend on implicit assumptions, which should be articulated.
   - The authors should reflect on the factors that influence the performance of the approach. For example, a facial recognition algorithm may perform poorly when image resolution is low or images are taken in low lighting. Or a speech-to-text system might not be used reliably to provide closed captions for online lectures because it fails to handle technical jargon.
   - The authors should discuss the computational efficiency of the proposed algorithms and how they scale with dataset size.
   - If applicable, the authors should discuss possible limitations of their approach to address problems of privacy and fairness.
   - While the authors might fear that complete honesty about limitations might be used by reviewers as grounds for rejection, a worse outcome might be that reviewers discover limitations that aren't acknowledged in the paper. The authors should use their best judgment and recognize that individual actions in favor of transparency play an important role in developing norms that preserve the integrity of the community. Reviewers will be specifically instructed to not penalize honesty concerning limitations.

3. **Theory assumptions and proofs**

   Question: For each theoretical result, does the paper provide the full set of assumptions and a complete (and correct) proof?

   Answer: [NA]

   Justification: The paper contains no formal theorems or theoretical results requiring proof; it is entirely empirical and algorithmic.

   Guidelines:

   - The answer NA means that the paper does not include theoretical results.
   - All the theorems, formulas, and proofs in the paper should be numbered and cross-referenced.
   - All assumptions should be clearly stated or referenced in the statement of any theorems.
   - The proofs can either appear in the main paper or the supplemental material, but if they appear in the supplemental material, the authors are encouraged to provide a short proof sketch to provide intuition.
   - Inversely, any informal proof provided in the core of the paper should be complemented by formal proofs provided in appendix or supplemental material.
   - Theorems and Lemmas that the proof relies upon should be properly referenced.

4. **Experimental result reproducibility**

Question: Does the paper fully disclose all the information needed to reproduce the main experimental results of the paper to the extent that it affects the main claims and/or conclusions of the paper (regardless of whether the code and data are provided or not)?

Answer: [Yes]

Justification: We give exhaustive details of our dataset construction (including LAION search and retrieve, filtering and annotation protocols) and our method (all pre-processing steps, corpora generation, hyper-parameters $k, \lambda, \alpha$, normalization statistics), and we commit to releasing both the dataset and code with scripts to reproduce every result.

Guidelines:

- The answer NA means that the paper does not include experiments.
- If the paper includes experiments, a No answer to this question will not be perceived well by the reviewers: Making the paper reproducible is important, regardless of whether the code and data are provided or not.
- If the contribution is a dataset and/or model, the authors should describe the steps taken to make their results reproducible or verifiable.
- Depending on the contribution, reproducibility can be accomplished in various ways. For example, if the contribution is a novel architecture, describing the architecture fully might suffice, or if the contribution is a specific model and empirical evaluation, it may be necessary to either make it possible for others to replicate the model with the same dataset, or provide access to the model. In general. releasing code and data is often one good way to accomplish this, but reproducibility can also be provided via detailed instructions for how to replicate the results, access to a hosted model (e.g., in the case of a large language model), releasing of a model checkpoint, or other means that are appropriate to the research performed.
- While NeurIPS does not require releasing code, the conference does require all submissions to provide some reasonable avenue for reproducibility, which may depend on the nature of the contribution. For example
  (a) If the contribution is primarily a new algorithm, the paper should make it clear how to reproduce that algorithm.
  (b) If the contribution is primarily a new model architecture, the paper should describe the architecture clearly and fully.
  (c) If the contribution is a new model (e.g., a large language model), then there should either be a way to access this model for reproducing the results or a way to reproduce the model (e.g., with an open-source dataset or instructions for how to construct the dataset).
  (d) We recognize that reproducibility may be tricky in some cases, in which case authors are welcome to describe the particular way they provide for reproducibility. In the case of closed-source models, it may be that access to the model is limited in some way (e.g., to registered users), but it should be possible for other researchers to have some path to reproducing or verifying the results.

5. **Open access to data and code**

Question: Does the paper provide open access to the data and code, with sufficient instructions to faithfully reproduce the main experimental results, as described in supplemental material?

Answer: [No]

Justification: Both *i*-CIR dataset and code will be made publicly available through our project page https://vrg.fel.cvut.cz/icir/.

Guidelines:

- The answer NA means that paper does not include experiments requiring code.
- Please see the NeurIPS code and data submission guidelines (https://nips.cc/public/guides/CodeSubmissionPolicy) for more details.
- While we encourage the release of code and data, we understand that this might not be possible, so "No" is an acceptable answer. Papers cannot be rejected simply for not including code, unless this is central to the contribution (e.g., for a new open-source benchmark).

- The instructions should contain the exact command and environment needed to run to reproduce the results. See the NeurIPS code and data submission guidelines (`https://nips.cc/public/guides/CodeSubmissionPolicy`) for more details.
- The authors should provide instructions on data access and preparation, including how to access the raw data, preprocessed data, intermediate data, and generated data, etc.
- The authors should provide scripts to reproduce all experimental results for the new proposed method and baselines. If only a subset of experiments are reproducible, they should state which ones are omitted from the script and why.
- At submission time, to preserve anonymity, the authors should release anonymized versions (if applicable).
- Providing as much information as possible in supplemental material (appended to the paper) is recommended, but including URLs to data and code is permitted.

6. **Experimental setting/details**

   Question: Does the paper specify all the training and test details (e.g., data splits, hyper-parameters, how they were chosen, type of optimizer, etc.) necessary to understand the results?

   Answer: [Yes]

   Justification: The experimental section of both the main paper and supplementary material is exhaustive and detailed. It specifies all the details needed to understand the results.

   Guidelines:
   - The answer NA means that the paper does not include experiments.
   - The experimental setting should be presented in the core of the paper to a level of detail that is necessary to appreciate the results and make sense of them.
   - The full details can be provided either with the code, in appendix, or as supplemental material.

7. **Experiment statistical significance**

   Question: Does the paper report error bars suitably and correctly defined or other appropriate information about the statistical significance of the experiments?

   Answer: [No]

   Justification: Our method is training-free, so it does not involve sources of randomness such as weight initialization, optimization, or data shuffling.

   Guidelines:
   - The answer NA means that the paper does not include experiments.
   - The authors should answer "Yes" if the results are accompanied by error bars, confidence intervals, or statistical significance tests, at least for the experiments that support the main claims of the paper.
   - The factors of variability that the error bars are capturing should be clearly stated (for example, train/test split, initialization, random drawing of some parameter, or overall run with given experimental conditions).
   - The method for calculating the error bars should be explained (closed form formula, call to a library function, bootstrap, etc.)
   - The assumptions made should be given (e.g., Normally distributed errors).
   - It should be clear whether the error bar is the standard deviation or the standard error of the mean.
   - It is OK to report 1-sigma error bars, but one should state it. The authors should preferably report a 2-sigma error bar than state that they have a 96% CI, if the hypothesis of Normality of errors is not verified.
   - For asymmetric distributions, the authors should be careful not to show in tables or figures symmetric error bars that would yield results that are out of range (e.g. negative error rates).
   - If error bars are reported in tables or plots, The authors should explain in the text how they were calculated and reference the corresponding figures or tables in the text.

8. **Experiments compute resources**

   Question: For each experiment, does the paper provide sufficient information on the computer resources (type of compute workers, memory, time of execution) needed to reproduce the experiments?

   Answer: [Yes]

   Justification: Technical and implementation details are included in the supplementary material.

   Guidelines:
   - The answer NA means that the paper does not include experiments.
   - The paper should indicate the type of compute workers CPU or GPU, internal cluster, or cloud provider, including relevant memory and storage.
   - The paper should provide the amount of compute required for each of the individual experimental runs as well as estimate the total compute.
   - The paper should disclose whether the full research project required more compute than the experiments reported in the paper (e.g., preliminary or failed experiments that didn't make it into the paper).

9. **Code of ethics**

   Question: Does the research conducted in the paper conform, in every respect, with the NeurIPS Code of Ethics https://neurips.cc/public/EthicsGuidelines?

   Answer: [Yes]

   Justification: We use only publicly available, license-annotated sources (*e.g.*, LAION-derived URLs and rights-cleared repositories), and we respect licenses and site Terms of Service. All images in *i*-CIR were reviewed by trained annotators; inappropriate, copyrighted, or privacy-sensitive content was removed. In categories where people are inherently present (*e.g.*, apparel), we automatically pixelate faces and perform spot checks; no raw PII is released. *i*-CIR is an evaluation-only benchmark, distributed under CC-BY-NC-SA with an explicit prohibition on surveillance/biometric or other privacy-invasive uses. We publish a misuse policy, provide a "Report misuse/PII" channel, honor takedown requests, and reserve the right to revoke access for violations. Our method is training-free and does not scrape private data, minimizing environmental and privacy risks. No human-subjects research was conducted (IRB not applicable). Overall, collection, curation, release, and documentation follow the NeurIPS Code of Ethics.

   Guidelines:
   - The answer NA means that the authors have not reviewed the NeurIPS Code of Ethics.
   - If the authors answer No, they should explain the special circumstances that require a deviation from the Code of Ethics.
   - The authors should make sure to preserve anonymity (e.g., if there is a special consideration due to laws or regulations in their jurisdiction).

10. **Broader impacts**

    Question: Does the paper discuss both potential positive societal impacts and negative societal impacts of the work performed?

    Answer: [Yes]

    Justification: The paper explicitly discusses benefits and risks. On the positive side, instance-level composed retrieval can support cultural-heritage search (GLAM), assistive-vision use cases, product provenance, and reproducible evaluation of compositional models. On the negative side, we analyze dual-use pathways—including surveillance/profiling, indirect "object-of-interest" tracking (*e.g.*, via distinctive belongings), fine-tuning our techniques on face/plate corpora, and misuse of crawl scripts—and we describe harms from both correct and incorrect system behavior. However, we exhaustively outline concrete mitigations too.

    Guidelines:
    - The answer NA means that there is no societal impact of the work performed.
    - If the authors answer NA or No, they should explain why their work has no societal impact or why the paper does not address societal impact.

- Examples of negative societal impacts include potential malicious or unintended uses (e.g., disinformation, generating fake profiles, surveillance), fairness considerations (e.g., deployment of technologies that could make decisions that unfairly impact specific groups), privacy considerations, and security considerations.
- The conference expects that many papers will be foundational research and not tied to particular applications, let alone deployments. However, if there is a direct path to any negative applications, the authors should point it out. For example, it is legitimate to point out that an improvement in the quality of generative models could be used to generate deepfakes for disinformation. On the other hand, it is not needed to point out that a generic algorithm for optimizing neural networks could enable people to train models that generate Deepfakes faster.
- The authors should consider possible harms that could arise when the technology is being used as intended and functioning correctly, harms that could arise when the technology is being used as intended but gives incorrect results, and harms following from (intentional or unintentional) misuse of the technology.
- If there are negative societal impacts, the authors could also discuss possible mitigation strategies (e.g., gated release of models, providing defenses in addition to attacks, mechanisms for monitoring misuse, mechanisms to monitor how a system learns from feedback over time, improving the efficiency and accessibility of ML).

11. **Safeguards**

Question: Does the paper describe safeguards that have been put in place for responsible release of data or models that have a high risk for misuse (e.g., pretrained language models, image generators, or scraped datasets)?

Answer: [Yes]

Justification: We release *i*-CIR as an *evaluation-only* benchmark under CC-BY-NC-SA with an explicit ban on surveillance/biometric and other privacy-invasive uses; access is governed by a misuse policy with revocation. Images are curated with a privacy-first process: annotators preferentially exclude PII, and in categories where people are intrinsic (*e.g.*, apparel) we retain images but *exhaustively pixelate* visible faces before release; we also filter watermarks, near-duplicates, and low-quality items, followed by manual spot checks. Any search/crawl scripts are released under the same restrictive license with hard-coded keyword blocks and documentation discouraging sensitive-content collection (IRB review recommended for modifications). The project page provides a prominent "Report misuse / PII" channel; we commit to prompt review, content takedown, and access revocation when warranted. We will periodically red-team object-level re-identification risks and update the release if failure modes are found. No identification models or person-level embeddings are released; our method is training-free and not tailored to biometric tasks.

Guidelines:

- The answer NA means that the paper poses no such risks.
- Released models that have a high risk for misuse or dual-use should be released with necessary safeguards to allow for controlled use of the model, for example by requiring that users adhere to usage guidelines or restrictions to access the model or implementing safety filters.
- Datasets that have been scraped from the Internet could pose safety risks. The authors should describe how they avoided releasing unsafe images.
- We recognize that providing effective safeguards is challenging, and many papers do not require this, but we encourage authors to take this into account and make a best faith effort.

12. **Licenses for existing assets**

Question: Are the creators or original owners of assets (e.g., code, data, models), used in the paper, properly credited and are the license and terms of use explicitly mentioned and properly respected?

Answer: [Yes]

Justification: We build on publicly available assets and cite them in the paper and project page. For data, we rely on LAION *metadata* (CC-BY 4.0) and keep each image under

its original source license. For models/code, we use public VLMs and toolchains under their original licenses and cite them: CLIP (MIT), OpenCLIP (Apache-2.0), and, for anonymization, InsightFace (MIT). Our own release is evaluation-only under CC-BY-NC-SA and does not alter upstream terms; LICENSE files and attributions are included in our repo.

Guidelines:

- The answer NA means that the paper does not use existing assets.
- The authors should cite the original paper that produced the code package or dataset.
- The authors should state which version of the asset is used and, if possible, include a URL.
- The name of the license (e.g., CC-BY 4.0) should be included for each asset.
- For scraped data from a particular source (e.g., website), the copyright and terms of service of that source should be provided.
- If assets are released, the license, copyright information, and terms of use in the package should be provided. For popular datasets, paperswithcode.com/datasets has curated licenses for some datasets. Their licensing guide can help determine the license of a dataset.
- For existing datasets that are re-packaged, both the original license and the license of the derived asset (if it has changed) should be provided.
- If this information is not available online, the authors are encouraged to reach out to the asset's creators.

13. **New assets**

    Question: Are new assets introduced in the paper well documented and is the documentation provided alongside the assets?

    Answer: [Yes]

    Justification: We release *i*-CIR with a comprehensive datasheet detailing collection protocols, licensing, annotation guidelines, dataset statistics, and limitations; our code repository includes installation instructions, example scripts, configuration files for all experiments, and explicit CC-BY-NC-SA license information.

    Guidelines:

    - The answer NA means that the paper does not release new assets.
    - Researchers should communicate the details of the dataset/code/model as part of their submissions via structured templates. This includes details about training, license, limitations, etc.
    - The paper should discuss whether and how consent was obtained from people whose asset is used.
    - At submission time, remember to anonymize your assets (if applicable). You can either create an anonymized URL or include an anonymized zip file.

14. **Crowdsourcing and research with human subjects**

    Question: For crowdsourcing experiments and research with human subjects, does the paper include the full text of instructions given to participants and screenshots, if applicable, as well as details about compensation (if any)?

    Answer: [Yes]

    Justification: We did not use crowdsourcing; all annotations were performed by salaried institutional staff. The supplemental material includes annotation guidelines/instructions. Annotators received domain and ethics training, and we ran weekly QA spot-checks and inter-annotator-agreement audits. Compensation exceeds the legal minimum ($>80\%$ of a first-year PhD stipend with full social-security coverage). No participants were recruited, no demographic attributes were collected, and PII was removed/redacted; thus this work does not constitute human-subjects research, but we nevertheless followed our institution's ethics policies throughout.

    Guidelines:

- The answer NA means that the paper does not involve crowdsourcing nor research with human subjects.
- Including this information in the supplemental material is fine, but if the main contribution of the paper involves human subjects, then as much detail as possible should be included in the main paper.
- According to the NeurIPS Code of Ethics, workers involved in data collection, curation, or other labor should be paid at least the minimum wage in the country of the data collector.

15. **Institutional review board (IRB) approvals or equivalent for research with human subjects**

Question: Does the paper describe potential risks incurred by study participants, whether such risks were disclosed to the subjects, and whether Institutional Review Board (IRB) approvals (or an equivalent approval/review based on the requirements of your country or institution) were obtained?

Answer: [NA]

Justification: This work does not constitute human-subjects research: no participants were recruited or interacted with; annotators were salaried staff performing routine labeling; no demographic or behavioral data were collected; and all images were sourced from public datasets with PII removed/pixelated prior to release. Under these conditions, IRB (or equivalent) review is not required per common definitions and institutional policies.

Guidelines:

- The answer NA means that the paper does not involve crowdsourcing nor research with human subjects.
- Depending on the country in which research is conducted, IRB approval (or equivalent) may be required for any human subjects research. If you obtained IRB approval, you should clearly state this in the paper.
- We recognize that the procedures for this may vary significantly between institutions and locations, and we expect authors to adhere to the NeurIPS Code of Ethics and the guidelines for their institution.
- For initial submissions, do not include any information that would break anonymity (if applicable), such as the institution conducting the review.

16. **Declaration of LLM usage**

Question: Does the paper describe the usage of LLMs if it is an important, original, or non-standard component of the core methods in this research? Note that if the LLM is used only for writing, editing, or formatting purposes and does not impact the core methodology, scientific rigorousness, or originality of the research, declaration is not required.

Answer: [NA]

Justification: The core method development in this research does not involve LLMs.

Guidelines:

- The answer NA means that the core method development in this research does not involve LLMs as any important, original, or non-standard components.
- Please refer to our LLM policy (https://neurips.cc/Conferences/2025/LLM) for what should or should not be described.

# Appendices

The Appendices includes the following information and results:

- An ethics statement is presented in Appendix A.
- Additional related work is presented in Appendix B.
- The *i*-CIR structure, statistics and a sample of image and text queries are presented in subsection C.1.
- The shortcomings of some existing datasets (FashionIQ, CIRR, CIRCO) are discussed and supported by examples in subsection C.2.
- Additional technical and implementation details of BASIC are discussed in Appendix D.
- Additional results are presented in Appendix E. In particular:
  - The impact of hyper-parameter values in subsection E.1.
  - In subsection E.2 we show that the proposed components are effective beyond the scope of composed image retrieval, *i.e*. for image-to-image and text-to-image retrieval too.
  - Detailed performance analysis per domain and category is presented in subsection E.3.
  - The performance comparison on instruction-like datasets is presented in subsection E.4.
  - The impact of corpora, and how they control the semantic projection, is further explored in subsection E.5.
  - Time comparison is presented in subsection E.6.
  - In subsection E.7 we present an experiment showing that *i*-CIR, despite being compact, is as challenging as including more than 40M distractor images.
  - In subsection E.8 we present an experiment to demonstrate the *i*-CIR is not biased towards the model used in the data collection process.
  - In subsection E.9 we assess robustness to text-query wording by rewriting text queries, showing how phrasing alone can markedly shift retrieval performance.
  - In subsection E.10 we present retrieval results in *i*-CIR, visually comparing different components of BASIC.

## A  Ethics statement

### A.1  Broader-impact & dual-use discussion

Instance-level composed image retrieval offers clear societal benefits: it can power fine-grained search across museum and GLAM archives, helping curators and the general public surface rare artifacts and provenance links that would be impractical to discover manually; it can also bolster assistive-vision tools that describe surroundings to blind or low-vision users by retrieving the precise objects referenced in natural-language queries. At the same time, the very same instance-level matching capabilities entail dual-use risks: in the wrong hands they could enable large-scale surveillance, doxxing, or targeted advertising by tracing a specific building, vehicle, or logo across web-scale image corpora.

Precisely because of these concerns, our dataset (*i*-CIR) was intentionally designed to minimize such potential misuse:

- *Privacy-first curation.* Annotators were trained to preferentially exclude images containing faces, license plates, private premises, or other personally identifiable information (PII) whenever doing so did not harm the task—*e.g*. for landmark queries where abundant alternatives without PII existed. For categories where people are intrinsic to the depiction (*e.g*. apparel modeled by humans), we retained the images but subsequently applied automatic, exhaustive anonymization: visible faces were pixelated across all retained images to protect identity while preserving the visual evidence needed for instance-level retrieval. Examples of one image query per instance and unique text queries are presented in Figure C2 and Figure E9 respectively.

- *Evaluation-only release.* *i*-CIR is published solely as an evaluation benchmark, not a training corpus. A model evaluated on *i*-CIR, containing no human-centric identifiers, cannot perform surveillance tasks without additional fine-tuning on sensitive data. While such re-purposing is possible in principle, it requires access to an external, privacy-violating dataset; our license explicitly forbids this.

- *Domain specificity.* Instance-level retrieval models are highly specialized to the visual domain on which they are trained. The gap between *i*-CIR (landmarks, products, fiction, fashion, tech gadgets) and human-centric surveillance domains further reduces direct transferability. Moreover, the categories of instances represented in *i*-CIR —landmarks, fictional characters, consumer products, fashion items, and technology gadgets (see Figure C1)—are already common in benchmarks such as Google Landmarks (GLD) [53], INSTRE [52], Stanford Online Products [36], and In-Shop [33]; therefore, *i*-CIR does not introduce novel categories of high-risk.

## A.2 Residual dual-use vectors & mitigations

- *Fine-tuning pathway – Risk*: Techniques (*e.g.* architectures, loss functions) validated on *i*-CIR could later be fine-tuned on a face-centric or plate-centric corpus to build a surveillance model. *Mitigation*: Our CC-BY-NC-SA license forbids biometric or privacy-invasive applications, and downstream works must inherit these restrictions. We additionally provide a misuse policy and reserve the right to revoke access for violators. We recognize, however, that a determined adversary could ignore license terms. Our goal is to follow community best practice and exert every reasonable control that dataset creators can apply, while keeping the research benefits intact.

- *Dataset-construction pathway – Risk*: Our LAION-based search-and-retrieve scripts, if misused, could harvest a new dataset rich in faces or license plates. *Mitigation*: We release the scripts under the same restrictive license and with hard-coded filters that block sensitive keywords. The accompanying documentation explicitly instructs researchers not to re-purpose the pipeline for sensitive-content collection and encourages institutional review board (IRB) review for any modifications. Importantly, similar LAION-based search-and-retrieve scripts have previously been employed [4]; our code does not introduce a novel capability but re-implements such workflow with stricter safety defaults.

- *Object-of-interest tracking — Risk*: Even without identifying people directly, one could try to track an individual via their belongings (*e.g.* a rare handbag or customized laptop sticker). *Mitigation*: We explicitly *prohibit* using *i*-CIR to identify, profile, or infer the movements or associations of any person—directly or indirectly via personal effects. Retrieval results from *i*-CIR must not be construed as evidence of co-location or identity. We will conduct periodic red-team evaluations focused on object-level re-identification risks and update the release if failure modes are found. The project page will prominently include a "Report misuse of the dataset" channel (web form/email) so the community can flag suspected abuse or PII leakage; reports will be reviewed promptly and may result in content takedown or revocation of access, consistent with our license.

> **Report misuse.** *i*-CIR must not be used to identify, profile, or infer movements/associations of any person. If you believe the dataset or code is being used in such a way or you discover content that reveals PII—please submit a report via the "Report misuse of the dataset" form or email us. We acknowledge reports promptly and may take actions including content removal or access revocation under the license.

## A.3 Transparency & governance commitments

- *License & API safeguards.* CC-BY-NC-SA with explicit surveillance prohibition; research-only API gated behind user agreements; automated checks to block bulk reverse-image searches of sensitive facilities.

- *Community reporting.* We provide clear channels for reporting misuse or ethical concerns (see "Report misuse" above). Substantiated reports will receive prompt acknowledgement and a public response timeline, and may result in content removal or access revocation consistent with the license.

## A.4 Human-annotation protocol (labor conditions & oversight)

- *Employment & Compensation.* All annotators are salaried employees of our institution (not crowd-workers). Their contracts include full social-security coverage and wages that exceed 80% of a first-year PhD stipend, comfortably above the legally mandated minimum in our region.

- *Training & Oversight.*
  - Domain training: A dedicated one-week training covered instance-level composed-image retrieval guidelines, annotation software, and quality-control protocols.
  - Ethics training: The same program included refresher modules on copyright compliance, privacy protection, and sensitive-content redaction.
  - Ongoing monitoring: Senior staff (including authors) conducted random spot-checks and weekly inter-annotator-agreement audits; annotators receive feedback whenever discrepancies arise, and, if needed, re-labeling was performed.

# B    More related work

**Datasets.**    Beyond composed image retrieval, instance-level retrieval has long served as a core benchmark for measuring fine-grained visual discrimination. Early datasets such as UKB [34] and Holidays [19] introduced clean, small-scale benchmarks of object and landmark instances, while later landmark datasets like Oxford [40], Paris [40], and Google Landmarks [35, 53] expanded toward large-scale evaluation but at the cost of noisy or incomplete annotations. Domain-specific datasets followed, covering products [36, 59, 38], fashion [32], and multi-domain or mixed setups such as INSTRE [52], GPR1200 [45], and UnED [57]. ILIAS [24] introduces a large-scale, domain diverse, and error-free testbed with 1,000 object instances and manually verified query/positive images, evaluated against 100M YFCC100M [48] distractors; by restricting instances to objects emerging after 2014, it mitigates false negatives without extra labeling and offers a challenging, unsaturated benchmark for both foundation models and classical retrieval.

# C    More on *i*-CIR and existing CIR datasets

## C.1    Structure, statistics, visualisations, and limitations of *i*-CIR

**Taxonomies.**    To better understand the diversity and structure of *i*-CIR, we organize all object instances into a 3-level hierarchy of visual categories and all textual modifications into a 1-level taxonomy of textual categories.

The visual taxonomy captures the type and nature of the object instance depicted in the image query. It begins with broad classes such as landmark, fictional, product, tech, and mobility, which are further refined into subcategories and specific object types. For example, "Asterix" is categorized as fictional → character → comic, while "Temple of Poseidon" falls under landmark → architecture → temple.

In parallel, each text query is annotated with one of seven high-level textual categories based on the nature of the transformation it describes:

- *Addition*: One or more external elements are introduced into the scene alongside the instance — such as people, objects, or animals. Examples: ``with a man proposing'', ``next to coffee beans''.

- *Appearance*: A full transformation of the instance's physical form or structure. The object is still the main focus but appears in a completely different embodiment. Examples: ``as a figurine'', ``as a sculpture'', ``as a scale model''.

- *Attribute*: A partial modification of the instance itself — such as color, minor structural differences, or subtle material changes. The identity remains unchanged. Examples: ``in purple color'', ``with yellow shoelaces''.

- *Context*: A modification in the surrounding environment or scene, lighting conditions, or time of day. These affect the setting without altering the object instance itself. Examples: ``at night, during sunset, outdoors on grass``.

- *Domain*: The instance is rendered in a different representational domain or medium, such as sketches, paintings, 3D renders, magazine ads, or comics. Examples: ``as a painting'', ``in a manga panel''.
- *Projection*: The object instance is placed onto another object or surface, often as decoration or branding — such as clothing, packaging, or household items. Examples: ``on a t-shirt'', ``printed on a pillow''.
- *Viewpoint*: A change in camera/viewer perspective, such as aerial shots or top-down views. The instance and environment remain the same. Examples: ``from an aerial viewpoint'', ``from a top-down viewpoint''.

These taxonomies, illustrated in Figure C1, provides insight into the types of visual and textual variation covered in *i*-CIR and supports performance breakdowns by category.

**The 202 instances.** Figure C2 provides a visual overview of the 202 object instances included in *i*-CIR. We intentionally set the total to 202—200 general instances plus 2 pet/animal instances (dogs)—included as a symbolic nod to the pets of our team while keeping the benchmark broad. Each image corresponds to a distinct instance, ranging from iconic landmarks and branded products to fictional characters and vehicles. Roughly half of the instances are *nameable* (with a canonical, widely recognized name; *e.g.* "Eiffel Tower"), while the remaining half are everyday objects that are best described compositionally (*e.g.* a "white-and-pink dolphin plushie"). All main-paper results are reported on the full *i*-CIR; in ablations below we also report on two subsets: *i*-CIR $_{named}$ (only nameable instances) and *i*-CIR $_{descriptive}$ (only compositionally described instances).

**Unique text queries.** Figure E9 presents the complete list of unique textual queries used in *i*-CIR. Each query describes a specific modification applied to an object instance, ranging from appearance and material to setting, domain, or interaction. These compositional cues form the basis of our retrieval queries and illustrate the rich semantic diversity captured in the benchmark.

**Personally identifiable information redaction (face pixelation).** To protect privacy while preserving task utility, we automatically *pixelated* visible human faces across *i*-CIR. We used off-the-shelf detectors to localize faces and required overlap with a person detector before redaction, which reduces false positives from posters, figurines, or mannequins; detected boxes were conservatively expanded before applying a mosaic filter. For categories where people are incidental (*e.g.* landmarks), such images were preferentially filtered by annotators during curation; where people are intrinsic to the content (*e.g.* apparel), we retained the images but anonymized all faces exhaustively. Thresholds were set for high recall and audited via stratified spot checks; license plates and other PII were similarly removed or obfuscated when encountered.

**Limitations of *i*-CIR** Our semi-automatic pipeline trades some speed for fidelity: every composed query is manually vetted (*e.g.* for hard negatives, PII issues, etc.), so building *i*-CIR is slower and costlier than fully automatic datasets, but yields an ambiguity-free benchmark the field currently lacks. Moreover, the design mostly assumes a single salient object per image query (see Figure C2) and uses English-only phrasing with public-domain imagery. Despite filtering and automated redaction, some residual PII may persist; based on stratified manual audits of random samples across categories, we estimate the prevalence of unredacted faces or other PII to be $< 2\%$. Our project page will include a "Report misuse / PII" channel, and we commit to prompt review and content updates or takedown upon verified reports.

## C.2 Shortcomings of existing benchmarks

**CIRR.** To substantiate our claims regarding the limitations of existing CIR benchmarks, we present qualitative retrieval results from the CIRR dataset using Text $\times$ Image in Figure E10. In ``fewer paper towels per pack'', a semantically relevant match appears as the third result, yet it is incorrectly annotated as a negative, showcasing a clear false negative. In ``the target is a Pepsi bottle'', the query text alone suffices to retrieve relevant images containing Pepsi bottles; the image query adds minimal value beyond indicating the Pepsi brand. Notably, the fifth and eighth retrieved images include visible Pepsi bottles, but are labeled as negatives—likely due to the co-occurrence of other non-Pepsi items, further exposing annotation inconsistencies. In ``two

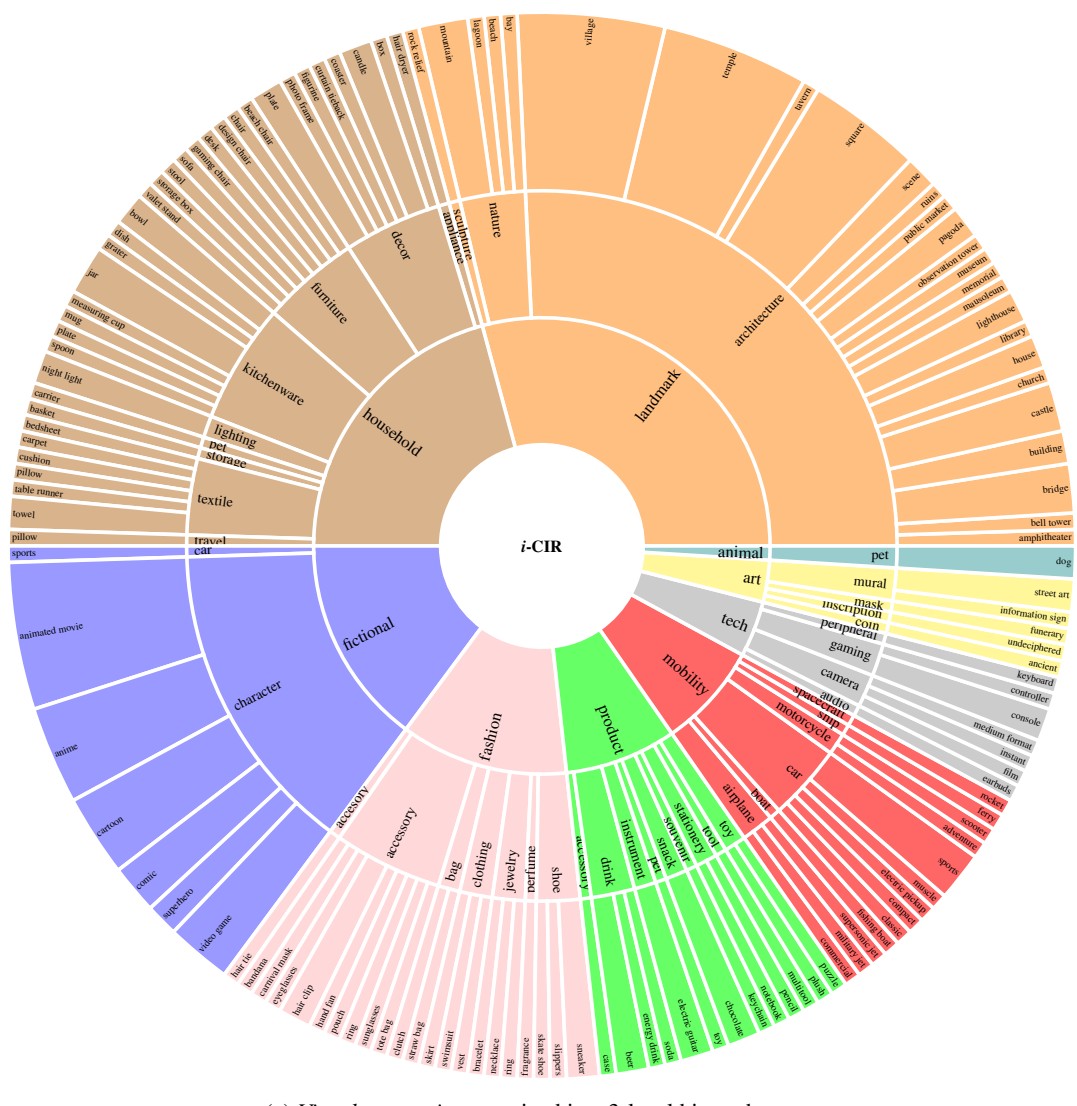

(a) *Visual categories* organized in a 3-level hierarchy.

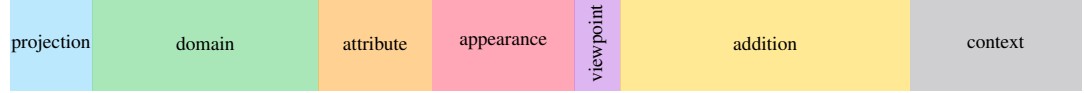

(b) *Textual modification categories* labeled at a single level.

Figure C1: *The i-CIR taxonomies* showing (a) the 3-level hierarchy of visual categories and (b) the single-level taxonomy of textual modification categories. This taxonomy captures the diversity and distribution of object instances and textual queries, and enables performance reporting per category.

antelopes standing one in front of the other'', again, the textual query alone retrieves the intended concept, rendering the image query unnecessary for composition. Yet, the seventh result clearly satisfies the query but is misclassified as a negative. Lastly, in ``widen the rows of TVs'', the supposed ground-truth positive contains four rows of TVs—just as in the image query—while the sixth retrieved image includes five rows, arguably an even better match. Nevertheless, it is annotated as a negative, underlining the prevalence of false negatives and the lack of fine-grained visual reasoning in the benchmark.

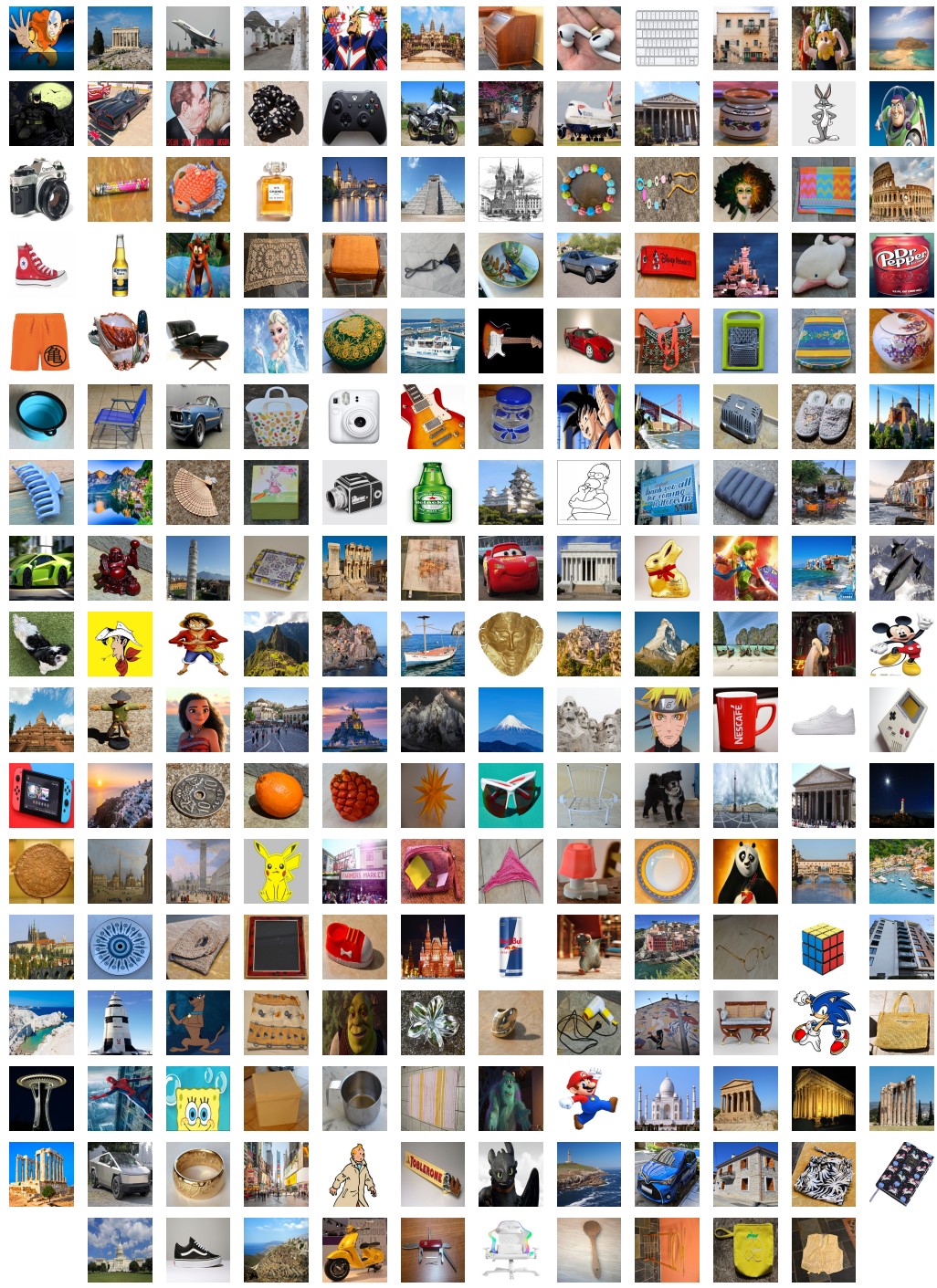

Figure C2: The 202 instances of *i*-CIR dataset. We randomly sample a query image per object instance.

**FashionIQ.** Figure E11 visualizes retrieval results on FashionIQ using Text × Image, revealing several inconsistencies. In many cases, such as ``is solid black, is long'' or ``is green with a four leaf clover, has no text'', correct results appear within the top-10 but are annotated as negatives, showing false negative noise. In other cases, the query image plays a minimal or redundant role; for instance, ``is the same'' easily retrieves the target based on visual matching alone, reducing the task to a simpler image-only similarity search. These observations support our

claim that FashionIQ, like other CIR benchmarks, suffers from supervision that stems from retrofitting relations between pre-existing images. As a result, the composed queries do not always require both modalities to be resolved.

**CIRCO.** Figure E12 presents retrieval examples from CIRCO using Text × Image, which appears to have less flaws than CIRR and FashionIQ. While CIRCO exhibits fewer problematic cases, it is not entirely free of issues. Due to the class-level nature of the dataset, the distinction between positive and negative samples can sometimes be ambiguous—even for human annotators. For instance, in the case of the query ``shows only one person'', visually plausible matches like the third retrieved image clearly align with the query yet are labeled as negatives. Similarly, for the query ``has more of them and is shot in greyscale'', multiple retrieved images fulfill both criteria but remain unannotated as positives. These examples highlight the challenge of assigning unambiguous relevance labels at the class-level, even in well-curated datasets.

# D    More on BASIC

**On projection matrix P.**    As described in the main manuscript (Sec. 4), we form the projection matrix $\mathbf{P} \in \mathbb{R}^{d \times k}$ by using the top-$k$ eigenvectors of $\mathbf{C}$:

$$\mathbf{C} = (1 - \alpha)\,\mathbf{C}_+ - \alpha\,\mathbf{C}_-\,, \quad \text{where} \quad \mathbf{C}_\pm = \frac{1}{|C_\pm|} \sum_{x \in C_\pm} \left(\phi^t(x) - \boldsymbol{\mu}^t\right)\left(\phi^t(x) - \boldsymbol{\mu}^t\right)^\top.$$

These eigenvectors capture directions of high variance in $\mathbf{C}_+$ and low variance in $\mathbf{C}_-$ when their corresponding eigenvalues are positive. Conversely, eigenvectors associated with negative eigenvalues indicate directions of high variance in $\mathbf{C}_-$ and low variance in $\mathbf{C}_+$, which we do not wish to retain.

To address this, we first count the number of "useful" components,

$$k_+ \;=\; \#\{\lambda_i > 0\},$$

*i.e.* the number of eigenvalues of $\mathbf{C}$ that are positive. We then set the final number of components as

$$k \;\leftarrow\; \min\!\big(k,\, k_+\big).$$

In practice, for moderate values of the mixing parameter $\alpha$, the initial choice of $k$ (*i.e.* $k = 250$) typically already excludes negative eigenvalues. However, for large $\alpha$ (*e.g.* $\alpha = 0.8$) negative eigenvalues can appear, necessitating this refinement.

**Query expansion details.**    Given the initial projected query image embedding $\mathbf{P}^\top \bar{\mathbf{q}}^v \in \mathbb{R}^d$, we compute its top-$k$ nearest neighbors $\{z_1^v, \ldots, z_k^v\} \subset X$ in the database $X$. This set is augmented with the query image itself, to avoid deviating from the original query. The centered features $\bar{\mathbf{z}}_i^v = \phi^v(z_i^v) - \boldsymbol{\mu}^v \in \mathbb{R}^d$ are then aggregated into an expanded query $\tilde{\mathbf{q}}^v \in \mathbb{R}^d$ via a soft attention-weighted mean:

$$\tilde{\mathbf{q}}^v = \sum_{i=1}^{k+1} w_i \bar{\mathbf{z}}_i^v, \quad \text{where} \quad w_i = \frac{\exp\left(\beta \left\langle \mathbf{P}^\top \bar{\mathbf{z}}_i^v, \mathbf{P}^\top \bar{\mathbf{q}}^v \right\rangle\right)}{\sum_{j=1}^{k+1} \exp\left(\beta \left\langle \mathbf{P}^\top \bar{\mathbf{z}}_j^v, \mathbf{P}^\top \bar{\mathbf{q}}^v \right\rangle\right)}. \tag{D1}$$

The "temperature" hyper-parameter $\beta$ is fixed to 0.1. Again, following the computational complexity analysis of Sec. 4, we can avoid operating directly on the dataset features. In particular, the top-$k$ retrieval can be accelerated (*e.g.* using FAISS [22]) by the formulation in the main manuscript, where sorting can be done using only the term $\left\langle \mathbf{x}^v, \mathbf{P}\mathbf{P}^\top(\mathbf{q}^v - \boldsymbol{\mu}^v) \right\rangle$.

**Contextualization insights.**    Contextualization provides a notable boost across all datasets (Table 2, main manuscript). However, the extent of this improvement varied by dataset. For example, contextualization does not produce the same gains for NICO++ as does for others. This dataset includes domains like ``ock'', ``water'', and ``outdoors'', where combining them with objects (*e.g.* "rock dog" or "dog rock") does not result in a particularly meaningful representation. In contrast, datasets like *i*-CIR benefit more clearly from context, *e.g.* ``dog during sunset''.

In some cases, adding the right level of textual detail, such as ``a dog in a rocky environment'', is important for forming a well-defined text query. This can be crucial for further unlocking the

potential of BASIC. However, generating such enriched text requires the use of LLMs, which can introduce prohibitive overhead when applied on-the-fly for each query. Alternatively, a lightweight (shallow) network trained specifically to contextualize abstract queries could be used directly on their embedding space. While this is a promising direction, it is not explored in this work.

**Corpora creation.** To create the various corpora used in this work, we used ChatGPT. We provided a carefully designed prompt with few examples and generated batches of entries (*e.g.* 100 corpus entries per prompt). Typically, the created corpus needs a post-processing of removing duplicates and erroneous outputs. For example, there were cases that after a number of entries, ChatGPT returned noisy data, such as `Description#87, Description#88` etc.

In particular, the prompt used to create the generic object corpus was:

---

**Generic object corpus prompt**

```
I would like you to generate a vocabulary of roughly 2,000 visual
classes that are similar in style and granularity to the categories
found in ImageNet-1K. These should include:
• Objects (e.g. ''espresso maker'', ''screwdriver'')
• Animals (e.g. ''Siberian husky'', ''hummingbird'')
• Scenes or natural elements (e.g. ''iceberg'', ''volcano'')
• Everyday items (e.g. ''shopping cart'', ''paper clip'')
• Tools, instruments, clothing, vehicles, and so on.

The classes should cover a broad range of everyday and recognizable
visual categories, suitable for training or evaluating vision-language
or classification models.  Some class names may be single-word (e.g.
''zebra'') while others may be multi-word expressions (e.g. ''motor
scooter'', ''garden hose'').

Please do not include brand names or highly specialized terms.  Favor
categories that have visually distinguishable appearances.

Output the vocabulary in chunks of 100 class names at a time,
formatted as a plain list (one per line).  Wait for me to say ''next''
before giving the next chunk.
```

---

In total, the generic corpora used consist of 1,800 entries in the object corpus and 1,000 stylistic elements in the negative corpus. To create the negative word corpus, which captures stylistic and contextual variations, we used the following prompt:

---

**Negative word corpus prompt**

```
We are working on a composed image retrieval task, where the input
consists of an image and a text.  The goal is to retrieve the most
relevant matching image from a database.  Typically, the image
contains a main object in a particular ''state''.  This state can refer
to a style (e.g. retro, handwritten, futuristic) or a contextual
setting (e.g. next to the sea, on the beach, beside a table).  For
example:  a retro mug, a cat next to a table, a handwritten notebook.
We would like you to generate a .txt file containing 100 possible
states in which an object can be found.  Each line in the file should
contain a single state expressed in natural language.  Wait for me to
say ''next'' before giving the next batch of states.
```

---

The first outputs of the above prompt are presented in Figure D3.

**Min normalization.** To normalize and balance the two query modalities, we proposed min-based normalization. To perform this step, the corresponding statistical values $s_{\min}^{v}$ and $s_{\min}^{t}$ must be

- on a wooden shelf
- floating in water
- wrapped in plastic
- painted with graffiti
- next to a fireplace
- sitting on grass
- surrounded by candles
- hanging from a tree
- under a spotlight

Figure D3: *Examples from the generic negative corpus.* Entries are natural-language style and contextual setting modifiers.

precomputed. In other words, we empirically estimate lower bounds on image-to-image and text-to-image similarities after the centralization step. These values can be extracted from a large existing dataset (*e.g.* LAION [46]). Instead, we opted to use a small synthetic dataset, generated via Stable Diffusion [42], where an image is created directly from its text description.

There are two main reasons for this choice: 1) we obtain an accurate correspondence between each text caption and its generated image, and 2) we compute all pairwise similarity scores (which scale quadratically with dataset size), allowing us to efficiently recompute statistics under different settings (*e.g.* with or without applying the projection step). Performing this over millions of embeddings would be prohibitive, whereas the required statistics are conceptually simple values.

Specifically, we first used ChatGPT to generate 100 simple and diverse textual prompts (*e.g.* ``a dog'', ``a calm sea'') and 100 more complex ones (*e.g.* ``a hidden waterfall in a lush, green jungle''). Each prompt was then used to generate 4 corresponding images using Stable Diffusion [42], encouraging intra-prompt diversity. All image-text and image-image pairs were then processed through our pipeline using the same CLIP backbone, and their similarity scores were computed. These scores were used to derive the normalization statistics $s^v_{\min}$ and $s^t_{\min}$, which are calculated after centralization.

For normalization, the minimum similarity values should always be negative. This is expected, as the pipeline is built upon cosine similarity, and given a sufficiently diverse set, centralization alone cannot shift all similarities into the positive range. This behavior is empirically validated in our experiments. Indicative values for the CLIP model are: $s^v_{\min} = -0.077$ and $s^t_{\min} = -0.117$.

**Limitations of BASIC.** BASIC assumes that both image and text components of a query provide semantically distinct and complementary, equally important, information. This allows us to interpret similarity multiplication as a logical conjunction operator. However, in tasks where one modality dominates (*e.g.* text-dominated edits as in FashionIQ [54] and CIRR [30]; examples shown in Figure E11 and Figure E10 respectively) or where the text query is highly entangled with image content (*e.g.* CIRCO [2] text queries like ``has a dog of a different breed and shows a jolly roger''), this assumption may not hold. In such cases, the benefits of our projection and fusion mechanism may diminish, or even have a negative impact, compared to using only the textual modality.

## E   More experiments

### E.1   Hyperparameter search of BASIC

During the development of BASIC, we set aside a small portion of the *i*-CIR crawl as a development set; none of these images appears in the final test benchmark. *i*-CIR $_{\text{dev}}$ consists of 15 object instances, 92 composed queries, and 45K images in total. All BASIC hyperparameters: contrastive scaling $\alpha$, number of PCA components k, and Harris regularization $\lambda$, were tuned once on this split and then frozen. To test the robustness of these choices, Table E1 runs a *leave-one-dataset-out* check: we treat each dataset in turn as the development set (re-tuning $\alpha, k, \lambda$ on that dataset only) and then evaluate the resulting configuration on all datasets. As expected, the best score per row typically occurs on the diagonal (in-domain), but the off-diagonal performance is very close, indicating that BASIC is not overly sensitive to where the hyperparameters are tuned. Notably, the configuration tuned on *i*-CIR $_{\text{dev}}$ performs on par with the cross-dataset alternatives; *i*-CIR $_{\text{dev}}$ is a small convenience split used during development and is not part of the public release.

To further assess the sensitivity of BASIC to its hyperparameters, Figure E4 presents one-factor-at-a-time sweeps over the contrastive scaling $\alpha$, the number of PCA components $k$, and the Harris regularization weight $\lambda$, holding the other two fixed. Across all datasets, the curves exhibit a broad

Table E1: *Cross-dataset hyperparameter stability for* BASIC. Each row lists the hyperparameters selected when tuning on the "Dev" dataset; each column reports the resulting score when evaluating that configuration on the "Test" dataset (higher is better). Bold indicates in-domain evaluation.

| Dev → Test | $i$-CIR | ImageNet-R | NICO++ | MiniDN | LTLL | Avg. |
|---|---|---|---|---|---|---|
| $i$-CIR ($\alpha = 0.2$, $k = 300$, $\lambda = 0.05$) | **32.95** | 30.60 | 30.21 | 38.50 | 41.02 | 34.66 |
| ImageNet-R ($\alpha = 0.4$, $k = 200$, $\lambda = 0.1$) | 30.67 | **32.63** | 31.39 | 38.49 | 40.41 | 34.72 |
| NICO++ ($\alpha = 0.2$, $k = 200$, $\lambda = 0.1$) | 30.95 | 32.55 | **31.85** | 39.81 | 41.32 | 35.30 |
| MiniDN ($\alpha = 0.2$, $k = 200$, $\lambda = 0.1$) | 30.95 | 32.55 | 31.85 | **39.81** | 41.32 | 35.30 |
| LTLL ($\alpha = 0.0$, $k = 300$, $\lambda = 0.1$) | 32.42 | 28.96 | 30.16 | 37.65 | **42.44** | 34.33 |
| $i$-CIR $_{dev}$ ($\alpha = 0.2$, $k = 250$, $\lambda = 0.1$) | 31.64 | 32.13 | 31.65 | 39.58 | 41.38 | 35.28 |

plateau around the optimum: while the exact maximizer may shift slightly within the (coarse) grid, it remains in the same neighborhood. We observe no sharp instabilities—moderate deviations from the tuned values incur only gradual changes in mAP— and the qualitative trends are consistent across datasets. Taken together, these results indicate that BASIC is robust to hyperparameter selection and that a single configuration transfers well across benchmarks.

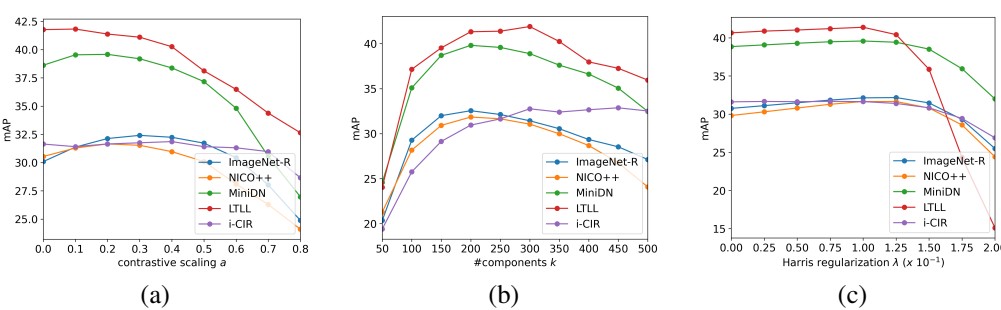

(a)  (b)  (c)

Figure E4: *Sensitivity of* BASIC *to hyperparameters.* mAP (%) on each dataset when varying one hyperparameter and holding the other two fixed: (a) contrastive scaling $\alpha$, (b) number of PCA components $k$, and (c) Harris regularization weight $\lambda$. Lines correspond to datasets in the legend (higher is better).

## E.2  Unimodal evaluation of BASIC components

To better understand the contribution of each component in BASIC, we conduct a unimodal evaluation, by analyzing the system's performance when querying with only one modality at a time. As shown in Table 2 of the main manuscript, the components of BASIC work in unison to produce the reported overall improvement. However, isolating the effects of each component in a unimodal setting can offer valuable insights into their individual contributions.

In this experiment, we consider two separate retrieval tasks: retrieving the most relevant images given a *text* query, and retrieving the most relevant images given an *image* query. We choose ImageNet-R for this analysis, as it allows for a clean decomposition of the two modalities. Table E2 presents the results, including a per-domain breakdown.

The following *key observations* can be made, supporting the analysis of the main manuscript:

- *Centering* consistently improves performance for both text and image queries.

- *Contextualization* significantly boosts performance in the text modality, with an improvement of over $6\%$ mAP, highlighting its importance in enriching sparse or abstract textual queries.

- The *semantic projection step* offers the most substantial gain, increasing mAP by over $17\%$, underscoring the importance of content-only preservation in the image modality.

- *Query expansion* further enhances the effect of projection, adding an additional $3\%$ improvement in mAP, acting as a complementary refinement mechanism. Notably, in the ORIGAMI case, this step reduces performance.

Table E2: Unimodal performance comparison in terms of mAP (%) using CLIP [41] for ImageNet-R. For each source domain (columns), we report the average mAP across all corresponding target domains. Each row progressively adds components in each modality, starting from the text-only or image-only baselines, respectively.

| Modality | Method | CAR | ORI | PHO | SCU | TOY | AVG |
|---|---|---|---|---|---|---|---|
| textual | Text | 67.02 | 67.00 | 74.94 | 71.77 | 77.31 | 71.61 |
| | + centr. | 72.12 | 72.52 | 79.62 | 76.10 | 79.53 | 75.98 |
| | + context. | 79.73 | 79.57 | 81.42 | 85.21 | 84.76 | 82.14 |
| visual | Image | 28.62 | 19.77 | 47.46 | 29.40 | 33.80 | 31.81 |
| | + centr. | 35.04 | 21.64 | 50.87 | 32.72 | 38.24 | 35.70 |
| | + proj. | 55.70 | 37.23 | 64.93 | 52.37 | 55.02 | 53.05 |
| | + q. exp. | 59.17 | 36.43 | 67.89 | 56.73 | 59.90 | 56.02 |

## E.3 Detailed comparisons across domains and categories

To further understand how BASIC performs across different domains in the evaluated datasets, as well as across the visual and textual categories of *i*-CIR, we provide detailed mAP results in Table E3 and Table E4, using two different vision-language models: CLIP [41] and SigLIP [58].

**CLIP.** Table E3 presents detailed results across domains and categories using CLIP [41] as backbone. Several key observations emerge:

- BASIC consistently outperforms all other approaches in the majority of domains and categories.
- FreeDom is the only method that achieves better performance than BASIC in any domain. Notably, this occurs only in the source domain PHOTO, and only within two datasets: ImageNet-R and MiniDN.
- In ImageNet-R, the source domain ORIGAMI is particularly challenging for all methods. For BASIC, ablating the negative-corpus term ($\alpha=0$) drops mAP from 21.11% to 16.65%, whereas emphasizing it ($\alpha=0.4$) raises mAP to 23.20%. This highlights how certain stylistic elements (*e.g.* ''origami'') can become entangled with the underlying semantic content, and how the contrastive formulation of the corpora helps disentangle them.
- On domain–conversion datasets, the ranking is stable: BASIC is consistently first and FreeDom a reliable second. On *i*-CIR, the runner-up is category-dependent: MagicLens is most often second and, in the *household* and *fashion* visual categories, it even ranks first—likely reflecting its training distribution (consumer/e-commerce imagery). Despite these fluctuations, BASIC remains the top method overall and the best performer across the majority of visual and textual categories.

**SigLIP.** Table E4 presents detailed results using SigLIP [58] as backbone. Not all methods in our evaluation are compatible with the SigLIP backbone. Approaches such as MagicLens and Pic2Word rely on training lightweight models (*e.g.* MLPs or small Transformers) directly on top of CLIP features, making them inherently tied to CLIP's representation space. Since SigLIP does not offer compatible pre-trained heads for these methods and retraining them on SigLIP features falls outside our scope, we restrict our comparison to FreeDom —a training-free method that directly operates on image and text features from the backbone. The following observations can be made:

- SigLIP improves overall retrieval performance across most datasets, with the only exception being NICO++, where a drop in performance was reported.
- BASIC significantly outperforms FreeDom on ImageNet-R, LTLL, and *i*-CIR. However, it reports lower mAP on both MiniDN and NICO++. Interestingly, in MiniDN, the two methods show complementary behavior—each outperforming the other on roughly half of the categories. This suggests that combining the strengths of both methods may further improve performance.
- In *i*-CIR, there are categories where the *Text* × *Image* baseline is on par or surpasses FreeDom. This highlights FreeDom's limitations in adapting to fine-grained, instance-level retrieval.

Table E3: Performance comparison in terms of mAP (%) using CLIP [41] across five datasets. (a-d): four domain conversion, where for each source domain (columns), we report the average mAP across all corresponding target domains; (e-f): *i*-CIR, where AP performance is averaged over queries grouped by their respective visual (e) or textual (f) category. AVG: average mAP over all source-target domain combinations. **Bold**: best; purple: second best.

| Method | CAR | ORI | PHO | SCU | TOY | AVG |
|---|---|---|---|---|---|---|
| Text | 0.75 | 0.66 | 0.60 | 0.80 | 0.75 | 0.71 |
| Image | 3.89 | 3.53 | 1.02 | 6.03 | 5.82 | 4.06 |
| Text + Image | 5.89 | 5.00 | 2.66 | 9.23 | 9.18 | 6.39 |
| Text × Image | 7.22 | 5.72 | 7.49 | 8.65 | 9.19 | 7.66 |
| Pic2Word | 7.60 | 5.53 | 7.64 | 9.39 | 9.27 | 7.88 |
| CompoDiff | 13.71 | 10.61 | 8.76 | 15.17 | 16.17 | 12.88 |
| WeiCom | 10.07 | 7.61 | 10.06 | 11.26 | 13.38 | 10.47 |
| SEARLE | 18.11 | 9.02 | 9.94 | 17.26 | 15.83 | 14.04 |
| MagicLens | 7.79 | 6.33 | 11.02 | 9.94 | 10.57 | 9.13 |
| FreeDom | 35.97 | 11.80 | **27.97** | 36.58 | 37.21 | 29.91 |
| BASIC | **36.49** | **21.11** | 26.07 | **39.30** | **37.68** | **32.13** |

(a) ImageNet-R

| Method | CLIP | PAINT | PHO | SKE | AVG |
|---|---|---|---|---|---|
| Text | 0.63 | 0.49 | 0.62 | 0.50 | 0.56 |
| Image | 8.19 | 7.42 | 5.12 | 7.52 | 7.06 |
| Text + Image | 11.08 | 9.74 | 10.71 | 8.20 | 9.94 |
| Text × Image | 9.72 | 7.21 | 15.55 | 5.44 | 9.48 |
| Pic2Word | 13.39 | 8.63 | 17.96 | 8.03 | 12.00 |
| CompoDiff | 19.06 | 24.27 | 23.41 | 25.05 | 22.95 |
| WeiCom | 7.52 | 7.04 | 15.13 | 4.40 | 8.52 |
| SEARLE | 25.04 | 18.72 | 23.75 | 19.61 | 21.78 |
| MagicLens | 24.40 | 17.54 | 28.59 | 9.71 | 20.06 |
| FreeDom | 41.96 | 31.65 | **41.12** | 34.36 | 37.27 |
| BASIC | **46.44** | **34.84** | 38.70 | **38.34** | **39.58** |

(b) MiniDomainNet

| Method | AUT | DIM | GRA | OUT | ROC | WAT | AVG |
|---|---|---|---|---|---|---|---|
| Text | 1.05 | 1.03 | 1.14 | 1.25 | 1.13 | 1.05 | 1.11 |
| Image | 6.82 | 5.00 | 6.02 | 8.02 | 8.06 | 5.91 | 6.64 |
| Text + Image | 8.99 | 6.89 | 9.66 | 12.34 | 11.91 | 8.61 | 9.74 |
| Text × Image | 7.99 | 6.32 | 11.40 | 11.64 | 10.09 | 8.11 | 9.26 |
| Pic2Word | 9.79 | 8.09 | 11.24 | 11.27 | 11.01 | 7.16 | 9.76 |
| CompoDiff | 10.07 | 7.83 | 10.53 | 11.41 | 11.93 | 10.15 | 10.32 |
| WeiCom | 8.58 | 7.39 | 13.04 | 13.17 | 11.32 | 9.73 | 10.54 |
| SEARLE | 13.49 | 13.73 | 17.91 | 17.99 | 15.79 | 11.84 | 15.13 |
| MagicLens | 18.76 | 15.17 | 22.14 | 23.61 | 21.99 | 16.30 | 19.66 |
| FreeDom | 24.35 | 24.41 | 30.06 | 30.51 | 26.92 | 20.37 | 26.10 |
| BASIC | **31.06** | **31.07** | **34.62** | **36.69** | **32.18** | **24.24** | **31.65** |

(c) NICO++

| Method | TODAY | ARCHIVE | AVG |
|---|---|---|---|
| Text | 5.09 | 5.30 | 5.20 |
| Image | 9.06 | 23.60 | 16.33 |
| Text + Image | 10.17 | 24.47 | 17.32 |
| Text × Image | 15.88 | 23.67 | 19.78 |
| Pic2Word | 17.86 | 24.67 | 21.27 |
| CompoDiff | 15.45 | 27.76 | 21.61 |
| WeiCom | 24.56 | 28.63 | 26.60 |
| SEARLE | 20.82 | 30.10 | 25.46 |
| MagicLens | 33.77 | 14.65 | 24.21 |
| FreeDom | 30.95 | 35.52 | 33.24 |
| BASIC | **42.76** | **40.00** | **41.38** |

(d) LTLL

| Method | FICT | LAND | MOB | HOUSE | TECH | FASH | PROD | ART |
|---|---|---|---|---|---|---|---|---|
| Text | 4.79 | 3.91 | 2.30 | 3.48 | 1.33 | 2.30 | 1.58 | 0.76 |
| Image | 1.05 | 2.43 | 1.78 | 2.64 | 2.52 | 2.18 | 1.82 | 1.77 |
| Text + Image | 2.56 | 12.85 | 8.18 | 3.96 | 9.92 | 3.48 | 4.18 | 4.86 |
| Text × Image | 21.05 | 27.88 | 25.50 | 11.25 | 22.97 | 11.87 | 18.10 | 26.81 |
| Pic2Word | 19.38 | 32.85 | 21.27 | 13.95 | 14.18 | 14.56 | 15.39 | 33.54 |
| CompoDiff | 10.51 | 15.06 | 19.27 | 3.56 | 6.18 | 7.41 | 4.54 | 13.04 |
| WeiCom | 18.15 | 30.21 | 20.27 | 8.82 | 23.30 | 12.22 | 18.67 | 34.27 |
| SEARLE | 31.10 | 31.05 | 18.52 | 11.69 | 16.97 | 14.39 | 17.23 | 22.75 |
| CIReVL | 28.66 | 29.64 | 23.39 | 14.06 | 23.75 | 11.74 | 24.89 | 19.04 |
| MagicLens | 28.79 | 34.98 | 29.31 | **29.06** | 18.71 | **25.63** | 26.69 | 34.99 |
| FreeDom | 30.73 | 17.55 | 28.87 | 15.55 | 14.51 | 12.81 | 25.49 | 23.52 |
| BASIC | **47.85** | **39.34** | **45.79** | 22.42 | **30.59** | 21.99 | **33.67** | **38.04** |

(e) *i*-CIR (visual)

| Method | PROJ | DOM | ATTR | APP | VIEW | ADD | CONT |
|---|---|---|---|---|---|---|---|
| Text | 2.09 | 3.69 | 2.77 | 4.13 | 1.87 | 4.62 | 2.75 |
| Image | 0.90 | 0.63 | 4.37 | 1.13 | 6.36 | 1.53 | 3.00 |
| Text + Image | 4.08 | 2.46 | 10.07 | 3.28 | 14.30 | 3.20 | 19.72 |
| Text × Image | 16.33 | 23.09 | 11.53 | 26.16 | 30.86 | 15.56 | 26.20 |
| Pic2Word | 20.02 | 24.41 | 11.92 | 19.02 | 32.51 | 16.79 | 34.55 |
| CompoDiff | 3.72 | 13.49 | 6.85 | 15.78 | 8.53 | 2.88 | 16.34 |
| WeiCom | 20.55 | 20.55 | 13.75 | 21.88 | 37.00 | 9.51 | 34.56 |
| SEARLE | 27.00 | 16.20 | 17.20 | 36.78 | 36.73 | 15.69 | 32.76 |
| CIReVL | 27.55 | 23.66 | 17.86 | 29.28 | 44.37 | 16.16 | 27.98 |
| MagicLens | 31.05 | 31.08 | 24.07 | 25.76 | 40.06 | **28.24** | **36.40** |
| FreeDom | 21.46 | 22.34 | 16.42 | 32.37 | 17.66 | 19.04 | 16.96 |
| BASIC | **53.10** | **39.31** | **26.29** | **48.83** | **47.81** | 24.04 | 35.59 |

(f) *i*-CIR (textual)

## E.4 Additional composed image retrieval datasets

**CIRR, FashionIQ, and CIRCO.** Table E5 presents a comparison between BASIC and prior work on CIRR, CIRCO, and FashionIQ. BASIC is outperformed by other approaches, such as CIReVL and CompoDiff, that do not perform well on *i*-CIR. The different datasets, *i.e. i*-CIR, domain conversion datasets, and those in Table E5, have different objectives and reflect different tasks. As a consequence, there is no single approach that is the best among all datasets. We argue that *i*-CIR better reflects real-world applications and use cases.

Unlike *i*-CIR and the domain conversion datasets, the text-only baseline consistently outperforms the image-only baseline. This aligns with our qualitative (Figure E10, Figure E11) and quantitative findings (Figure 6, main manuscript), confirming that in these datasets, the text query often provides sufficient information to resolve the task independently, rather than serving to refine or complement the image query. Notably, in CIRR under the $R@1$ metric, the text-only baseline even surpasses CompoDiff. Similarly, in FashionIQ, the average performance of the text-only baseline is only $2.65\%$ $R@10$ below that of FreeDom. CIRCO appears to be less affected by this issue than the other two datasets. To further investigate the imbalance between modalities, we conduct an ablation study by

Table E4: Performance comparison in terms of mAP (%) using SigLIP [58] across five datasets. (a-d): four domain conversion, where for each source domain (columns), we report the average mAP across all corresponding target domains; (e-f): *i*-CIR, where AP performance is averaged over queries grouped by their respective visual (e) or textual (f) category. AVG: average mAP over all source-target domain combinations. **Bold**: best; purple: second best.

| Method | CAR | ORI | PHO | SCU | TOY | AVG |
|---|---|---|---|---|---|---|
| Text | 0.88 | 0.80 | 0.62 | 0.95 | 0.90 | 0.83 |
| Image | 4.97 | 3.71 | 0.85 | 8.18 | 7.40 | 5.02 |
| Text + Image | 7.88 | 5.84 | 3.08 | 13.50 | 12.71 | 8.60 |
| Text × Image | 6.57 | 4.34 | 4.89 | 6.46 | 7.46 | 5.94 |
| FreeDom | 49.46 | 27.12 | 38.11 | 47.52 | 46.90 | 41.82 |
| BASIC | 53.67 | 39.54 | 36.40 | 51.39 | 53.61 | 46.92 |

(a) ImageNet-R

| Method | CLIP | PAINT | PHO | SKE | AVG |
|---|---|---|---|---|---|
| Text | 0.76 | 0.72 | 0.76 | 0.75 | 0.74 |
| Image | 5.07 | 7.53 | 3.68 | 6.15 | 5.61 |
| Text + Image | 7.79 | 11.33 | 10.80 | 9.02 | 9.74 |
| Text × Image | 3.00 | 2.60 | 4.34 | 3.18 | 3.28 |
| FreeDom | 57.14 | 45.47 | 59.71 | 52.21 | 53.63 |
| BASIC | 59.76 | 45.58 | 54.67 | 47.73 | 51.94 |

(b) MiniDN

| Method | AUT | DIM | GRA | OUT | ROC | WAT | AVG |
|---|---|---|---|---|---|---|---|
| Text | 1.08 | 1.13 | 1.04 | 1.26 | 1.10 | 1.11 | 1.12 |
| Image | 6.19 | 5.19 | 5.42 | 7.67 | 7.44 | 5.62 | 6.25 |
| Text + Image | 8.35 | 7.19 | 8.08 | 11.42 | 10.57 | 8.12 | 8.95 |
| Text × Image | 2.31 | 2.91 | 3.26 | 3.53 | 3.25 | 2.90 | 3.03 |
| FreeDom | 30.28 | 29.96 | 33.86 | 37.16 | 33.14 | 26.49 | 31.81 |
| BASIC | 28.11 | 32.39 | 31.13 | 34.07 | 29.26 | 23.12 | 29.68 |

(c) NICO++

| Method | TODAY | ARCHIVE | AVG |
|---|---|---|---|
| Text | 5.02 | 3.84 | 4.43 |
| Image | 28.14 | 10.25 | 19.20 |
| Text + Image | 26.73 | 10.16 | 18.44 |
| Text × Image | 3.49 | 4.87 | 4.18 |
| FreeDom | 47.00 | 27.45 | 37.22 |
| BASIC | 45.01 | 39.09 | 42.05 |

(d) LTLL

| Method | FICT | LAND | MOB | HOUSE | TECH | FASH | PROD | ART |
|---|---|---|---|---|---|---|---|---|
| Text | 7.61 | 5.09 | 2.18 | 11.59 | 4.40 | 8.33 | 1.82 | 1.59 |
| Image | 1.31 | 3.46 | 2.91 | 6.32 | 2.42 | 5.96 | 2.03 | 1.52 |
| Text + Image | 3.91 | 13.41 | 6.92 | 12.51 | 12.90 | 11.71 | 7.89 | 3.58 |
| Text × Image | 25.56 | 21.42 | 12.93 | 32.86 | 28.53 | 18.30 | 12.18 | 22.67 |
| FreeDom | 27.75 | 27.10 | 19.32 | 43.02 | 31.36 | 35.31 | 20.27 | 33.11 |
| BASIC | 50.65 | 53.45 | 39.56 | 48.87 | 50.84 | 52.83 | 45.11 | 44.43 |

(e) *i*-CIR (visual)

| Method | PROJ | DOM | ATTR | APP | VIEW | ADD | CONT |
|---|---|---|---|---|---|---|---|
| Text | 5.95 | 5.14 | 5.22 | 6.88 | 1.74 | 10.45 | 4.55 |
| Image | 1.24 | 1.09 | 6.31 | 1.35 | 8.98 | 4.09 | 4.63 |
| Text + Image | 7.37 | 3.52 | 11.18 | 3.29 | 17.12 | 12.66 | 20.12 |
| Text × Image | 21.61 | 17.89 | 20.19 | 26.47 | 11.15 | 28.24 | 25.13 |
| FreeDom | 22.94 | 25.25 | 20.03 | 31.52 | 38.34 | 39.17 | 25.59 |
| BASIC | 53.41 | 51.17 | 42.19 | 50.62 | 63.73 | 47.42 | 50.90 |

(f) *i*-CIR (textual)

removing the Harris regularization (denoted as BASIC *). Results show improved performance in CIRR and FashionIQ, supporting the hypothesis of modality imbalance.

**Refined CIRR and FashionIQ.** We evaluate BASIC and BASIC without Harris regularization (BASIC *) on the newly released CoLLM [18] refined [18] versions of CIRR and FashionIQ in Table E6. Both variants gain notably on the refined splits (*e.g.* up to +13.7 *R*@10 on refined–FashionIQ SHIRT). These gains are consistent with CoLLM's goal of reducing annotation ambiguity. This is orthogonal to modality dominance (as evidenced by BASIC * outperforming BASIC on text-dominant settings) and to CLIP's phrasing sensitivities (*e.g.* instructional/relational text).

## E.5 Dedicated vs generic corpora

We further explore the impact of using dataset or domain-specific (dedicated) vs. generic corpora on retrieval performance. This analysis is conducted on LTLL, which contains a narrow range of object categories—specifically, landmarks such as monuments and architectural sites.

The goal is to assess whether adding more targeted corpora can improve performance. While incorporating a landmark-only corpus into *i*-CIR would be problematic, due to the large number of irrelevant distractors that can introduce noise when projecting into a restricted semantic space, LTLL is composed entirely of landmark-related imagery, making it well-suited for such decomposition.

We experiment with two types of corpora:

- *Generic corpora*, which include general stylistic descriptors.
- *Dedicated corpora*, designed specifically for this task:

Table E5: Performance comparison using CLIP [41] across three generic composed image retrieval datasets. BASIC is evaluated for the default settings, as well as without the Harris regularization ($\star$), which found to be beneficial for these datasets.

| Method | R@1 | R@5 | R@10 | R@50 |
|---|---|---|---|---|
| Text | 20.96 | 44.89 | 56.80 | 79.16 |
| Image | 7.42 | 23.61 | 34.07 | 57.40 |
| Text + Image | 12.41 | 36.15 | 49.18 | 78.27 |
| Text × Image | 22.55 | 50.36 | 62.84 | 86.02 |
| Pic2Word | 23.90 | 51.70 | 65.30 | 87.80 |
| SEARLE | 24.20 | 52.50 | 66.30 | 88.80 |
| CompoDiff | 18.20 | 53.10 | 70.80 | 90.30 |
| FreeDom | 21.00 | 48.70 | 61.90 | 88.10 |
| CIReVL | 24.60 | 52.30 | 64.90 | 86.30 |
| MagicLens | 30.10 | 61.70 | 74.40 | 92.60 |
| BASIC | 15.83 | 40.89 | 53.90 | 82.27 |
| BASIC * | 17.98 | 44.92 | 58.80 | 86.51 |

(a) CIRR

| Method | mAP@5 | mAP@10 | mAP@25 | mAP@50 |
|---|---|---|---|---|
| Text | 3.09 | 3.25 | 3.76 | 4.01 |
| Image | 1.60 | 2.02 | 2.76 | 3.13 |
| Text + Image | 4.06 | 5.20 | 6.29 | 6.85 |
| Text × Image | 11.64 | 12.29 | 13.64 | 14.28 |
| Pic2Word | 8.70 | 9.50 | 10.70 | 11.30 |
| SEARLE | 11.70 | 12.70 | 14.30 | 15.10 |
| CompoDiff | 12.60 | 13.40 | 15.80 | 16.40 |
| FreeDom | 14.00 | 14.80 | 16.40 | 17.20 |
| CIReVL | 18.60 | 19.00 | 20.90 | 21.80 |
| MagicLens | 29.60 | 30.80 | 33.40 | 34.40 |
| BASIC | 15.95 | 16.77 | 18.19 | 18.94 |
| BASIC * | 15.95 | 16.77 | 18.21 | 19.00 |

(b) CIRCO

| Method | Dress | | Shirt | | Toptee | | Average | |
|---|---|---|---|---|---|---|---|---|
| | R@10 | R@50 | R@10 | R@50 | R@10 | R@50 | R@10 | R@50 |
| Text | 14.53 | 32.92 | 20.51 | 34.69 | 21.83 | 39.57 | 18.95 | 35.73 |
| Image | 4.36 | 12.84 | 10.55 | 19.97 | 8.21 | 16.37 | 7.71 | 16.39 |
| Text + Image | 17.40 | 35.60 | 21.84 | 36.26 | 23.30 | 39.37 | 20.85 | 37.08 |
| Text × Image | 21.52 | 41.00 | 27.63 | 42.05 | 28.71 | 47.22 | 25.95 | 43.42 |
| Pic2Word | 20.00 | 40.20 | 26.20 | 43.60 | 27.90 | 47.40 | 24.70 | 43.70 |
| SEARLE | 20.50 | 43.10 | 26.90 | 45.60 | 29.30 | 50.00 | 25.60 | 46.20 |
| CompoDiff | 32.20 | 46.30 | 37.70 | 49.10 | 38.10 | 50.60 | 36.00 | 48.60 |
| FreeDom | 16.80 | 36.30 | 23.50 | 38.50 | 24.70 | 43.70 | 21.60 | 39.50 |
| CIReVL | 24.80 | 44.80 | 29.50 | 47.40 | 31.40 | 53.70 | 28.60 | 48.60 |
| MagicLens | 25.50 | 46.10 | 32.70 | 53.80 | 34.00 | 57.70 | 30.70 | 52.50 |
| BASIC | 18.59 | 37.68 | 26.30 | 44.80 | 23.92 | 40.95 | 22.94 | 41.14 |
| BASIC * | 19.98 | 39.86 | 29.88 | 47.06 | 26.21 | 44.57 | 25.36 | 43.83 |

(c) FASHIONIQ

Table E6: *Performance on CoLLM-refined CIRR and FashionIQ. Both BASIC and BASIC * improve on the refined splits, indicating reduced annotation ambiguity.*

| Dataset | Split | Method | mAP | R@1 | R@5 | R@10 | R@50 |
|---|---|---|---|---|---|---|---|
| CIRR | Legacy | BASIC | 24.34 | 15.83 | 40.89 | 53.90 | 82.27 |
| | Refined | BASIC | 28.28 | 21.87 | 48.96 | 60.82 | 86.66 |
| | Legacy | BASIC * | 26.89 | 17.98 | 44.92 | 58.80 | 86.51 |
| | Refined | BASIC * | 31.89 | 25.30 | 53.49 | 65.94 | 90.15 |
| FashionIQ DRESS | Legacy | BASIC | 6.96 | – | – | 18.59 | 37.68 |
| | Refined | BASIC | 9.91 | – | – | 24.40 | 47.63 |
| | Legacy | BASIC * | 7.43 | – | – | 19.98 | 39.86 |
| | Refined | BASIC * | 11.27 | – | – | 28.59 | 52.43 |
| FashionIQ SHIRT | Legacy | BASIC | 10.93 | – | – | 26.30 | 44.80 |
| | Refined | BASIC | 17.15 | – | – | 36.82 | 56.09 |
| | Legacy | BASIC * | 12.93 | – | – | 29.88 | 47.06 |
| | Refined | BASIC * | 22.55 | – | – | 43.54 | 62.40 |
| FashionIQ TOPTEE | Legacy | BASIC | 10.28 | – | – | 23.92 | 40.95 |
| | Refined | BASIC | 14.57 | – | – | 32.08 | 51.41 |
| | Legacy | BASIC * | 12.08 | – | – | 26.21 | 44.57 |
| | Refined | BASIC * | 19.08 | – | – | 38.27 | 57.98 |

- Architectural terms (*e.g.* "Byzantine basilica", "Crusader fortress", "slate tile", "oxidized copper", "weathered limestone')
- Temporal/stylistic cues (*e.g.* "flash photography", "shadows of age", "digital camera photo", "restored image")

Table E7: LTLL dataset mAP (%) results using different combinations of positive (+) and negative (–) text corpora: generic (Gen) or dedicated (Ded). We consider different text queries with the same semantic meaning. Rows correspond to target domains - which are the text queries used.

| Target Domain | Only Gen+ | Gen+ & Gen– | Only Ded+ | Ded+ & Ded– | Gen+ & Ded– | Ded+ & Gen– |
|---|---|---|---|---|---|---|
| archive | 41.41 | 40.00 | 42.83 | 43.00 | 43.60 | 40.97 |
| today | 42.13 | 42.76 | 43.96 | 45.97 | 44.38 | 45.74 |
| *mean* | 41.77 | 41.38 | 43.40 | 44.48 | 43.99 | 43.36 |
| old | 50.52 | 48.86 | 50.78 | 50.77 | 50.58 | 48.89 |
| new | 38.24 | 40.00 | 41.77 | 45.68 | 41.12 | 44.66 |
| *mean* | 44.38 | 44.43 | 46.28 | 48.22 | 45.85 | 46.78 |
| vintage | 54.04 | 53.92 | 54.30 | 57.37 | 57.43 | 53.86 |
| today | 42.13 | 42.76 | 43.96 | 45.97 | 44.38 | 45.74 |
| *mean* | 48.08 | 48.34 | 49.13 | 51.67 | 50.90 | 49.80 |

We also evaluate the effect of varying textual queries that convey the same underlying meaning, such as synonyms or stylistic variants. Examples include:

- ''archive'' vs. ''old'' vs. ''vintage'' — all describing temporally distant or aged imagery.

- ''today'' vs. ''new'' — both indicating contemporary or modern representations.

Table E7 presents these performance comparisons over different combinations of corpora and text queries. The following observations can be made:

- Different textual queries, despite their close-almost identical-semantic meaning, can have a substantial impact on retrieval results. The results for the first two textual pairs (''today'' vs. ''now'' and ''archive'' vs. ''old'') reveal interesting trends. The query ''today'' is more effective at retrieving contemporary photos compared to ''now''. In contrast, the ''archive'' underperforms relative to ''old''. Semantically, this is expected—''archive'' is not commonly used to describe old photos in natural language queries. To address this, we also include the query ''vintage'', which offers a clearer and more intuitive descriptor for aged or historical imagery.

- The use of dedicated corpora consistently improves performance across all textual variants. Even a single dedicated corpus yields a notable boost.

- Adding a negative corpus generally enhances performance, with the exception of the (''today'', ''archive'') pair when combined with the generic stylistic corpus.

- The baseline mAP for the (''today'', ''archive'') setting is 41.38% using generic corpora. With dedicated corpora and well-matched textual queries the performance increases to 51.67%, representing a gain of more than 10% mAP.

We further examine the case of *i*-CIR by leveraging its predefined visual categories, as illustrated in Figure C1. Based on this taxonomy, we construct a dedicated object corpus of 300 elements tailored specifically to *i*-CIR. This setup serves as a proof-of-concept, as it assumes access to detailed visual categorization—information that is typically unavailable in realistic retrieval scenarios.

To avoid introducing bias toward any specific category, which may collapse the semantic representation towards a specific direction, we ensure that objects are selected from across all visual categories. As in previous experiments, we concatenate the dedicated corpus entries with those from the generic corpus.

Table E8: *i*-CIR performance comparison of different object corpora in terms of mAP (%). CLIP [41] backbone is used. We compare the case of the generic corpus vs the dedicated corpus across the different visual categories of *i*-CIR.

| Positive Corpus | FICT | LAND | MOB | HOUSE | TECH | FASH | PROD | ART |
|---|---|---|---|---|---|---|---|---|
| Generic | 47.85 | 39.34 | 45.79 | **22.42** | 30.59 | 21.99 | 33.67 | **38.04** |
| Dedicated | **49.48** | **39.35** | **47.05** | 21.92 | **34.45** | **22.91** | **35.00** | 37.88 |

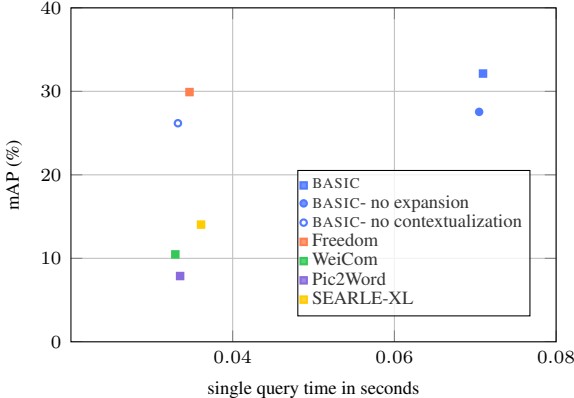

Figure E5: *Latency vs. mAP(%)* on ImageNet-R. Comparison of methods from the literature and BASIC with and without query expansion and contextualization.

Table E8 reports per-category mAP on *i*-CIR. The category-aware dedicated corpus yields improvements in six of eight categories (parity in LANDMARK, minor drops in HOUSEHOLD and ART). Note that these corpora were automatically generated with ChatGPT from a short prompt (see example in box) and were not hand-balanced to match *i*-CIR 's taxonomy; their distribution only loosely aligns with the actual category frequencies. Despite this mismatch—and the fact that we simply concatenate the dedicated with the generic corpus—we still observe consistent gains, suggesting that a weak, category-aware prior can help without tightly fitting the dataset. We view this as a proof-of-concept.

> **Dedicated word corpus prompt**
>
> ```
> Provide a .txt file with 300 distinct household objects (as if they
> are labels for a classification task).  Be broad and diverse.
> ```

### E.6 Time requirements

Figure E5 presents a per-query latency comparison between BASIC and several existing methods from the literature: WeiCom, Pic2Word, SEARLE-XL, and Freedom, evaluated on ImageNet-R. BASIC achieves the highest performance, reaching a mAP of 32.13 with a latency of 70.96ms per query. When query expansion is disabled, the performance drops by 4.59 mAP, while the latency improves slightly by 0.48ms. The primary overhead in BASIC is the contextualization step. When removed, BASIC (33.23ms) becomes faster than Freedom (34.66ms), SEARLE-XL (36.09ms), and Pic2Word (33.50ms), and only marginally slower than WeiCom by 0.03ms. Note that no further retrieval optimization has been applied for these experiments (*e.g.* FAISS [22]).

We intentionally exclude CompoDiff [13] and CIReVL [23] from this latency comparison: CompoDiff relies on a diffusion generator at inference, and CIReVL invokes a captioner (BLIP 2 [25]) and an LLM (LLaMA [49]). These pipelines are orders of magnitude heavier and are not designed for low-latency retrieval, so including them would make the plot incomparable and obscure the efficiency trade-offs among lightweight methods.

As expected, aside from the contextualization step — which introduces the overhead of computing additional CLIP embeddings (fixed to generating 100 such descriptions across all experiments) — the proposed approach remains very lightweight. The overhead introduced by contextualization is a current limitation of the method, but could be addressed by integrating a separate shallow network that contextualizes the CLIP embedding of a given text query, as discussed in Appendix D.

### E.7 *i*-CIR: Compact but hard

We use randomly selected images from LAION as negatives to assess how challenging *i*-CIR is in comparison to a large-scale database that is commonly shared across all queries and lacks explicit hard negatives. Figure E6 shows that the performance drop on *i*-CIR is comparable to the degradation

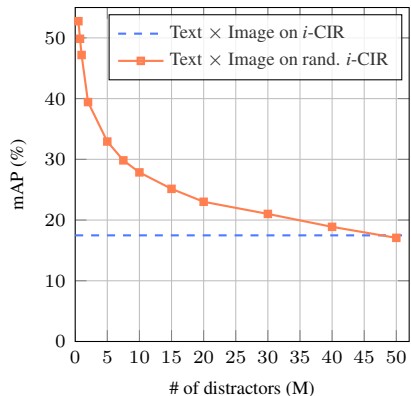

Figure E6: *Impact of curated vs. random negatives on retrieval difficulty.* We plot mAP on *i*-CIR with its 750K curated hard negatives against the same benchmark augmented with up to 50M random LAION distractors. Even tens of millions of distractors do not match the difficulty imposed by curated hard negatives.

caused by tens of millions of randomly sampled LAION images. Using the Text × Image baseline, we show that *i*-CIR, with its curated hard negatives, is substantially more challenging than a 40M-image LAION distractor set.

## E.8   Collection bias

While we take explicit steps to mitigate potential bias toward CLIP ViT-L/14-based methods during data collection such as discarding all seed images and ensuring that seed sentences do not exactly match any text query, the possibility of residual bias cannot be entirely ruled out. Since the retrieval of candidate negatives relies on CLIP ViT-L/14, one might expect this model to underperform relative to weaker models due to the increased difficulty of the curated negatives tailored to its embedding space. To test this, we evaluate the standard Text × Image baseline using a weaker model, CLIP ViT-B/32. The results show that ViT-B/32 achieves 18.66% mAP on *i*-CIR, notably lower than the 23.28% mAP of ViT-L/14. This confirms that no bias favoring alternative models has been introduced and that the performance of ViT-L/14 is not artificially deflated by the collection process.

## E.9   Effect of text query style on retrieval

A text query can be almost equivalently formulated in different ways, *e.g.* concisely (''`engraving`''), in a longer sentence (''`as a vintage stylish engraving`''), in a relational way (''`the same landmark as the one depicted in the image but as an engraving`''), and in an instructional way (''`engrave the landmark depicted in the image`''). In the context of *i*-CIR, we refer to those cases as *original*, *longer*, *relational*, and *instructional*, respectively. The queries of *i*-CIR are brief and comprehensive, while existing datasets like CIRR and CIRCO are often relational or instructional, while also having larger lengths.

To probe the robustness of BASIC against such text query styles, we automatically rewrite (via ChatGPT) all text queries of *i*-CIR $_{named}$, and we present the results in Table E9 ($\Delta$ is the relative change with respect to the original row).

Table E9: *Impact of text query style on retrieval.* We compare the original (concise) queries from *i*-CIR $_{named}$ with automatically rewritten *longer*, *relational*, and *instructional* variants and report mAP for BASIC, FreeDom, and MagicLens. $\Delta$ is the relative change vs. the original row.

| Text-query Variant | BASIC mAP | BASIC $\Delta$% | FreeDom mAP | FreeDom $\Delta$% | MagicLens mAP | MagicLens $\Delta$% |
|---|---|---|---|---|---|---|
| Original (concise) | 47.39 | – | 25.14 | – | 32.00 | – |
| Longer | 45.02 | −5 | 23.91 | −5 | 26.90 | −16 |
| Relational | 38.95 | −18 | 18.99 | −25 | 29.34 | −8 |
| Instructional | 37.47 | −21 | 18.31 | −27 | 28.85 | −10 |

**Longer.** We expand all text queries by adding 1 to 4 extra words while preserving their meaning. On average the phrases grow from 4.4 to 7.9 words, an increase of 3.5 words or $+79\%$, effectively almost doubling their length. The median rises from 4 to 8 words, the shortest query goes from 1 to 4 words, and the longest from 13 to 15. Example: ''crowded'' becomes ''`in a dense and crowded scene`''. The ChatGPT prompt used to convert text queries is presented below.

---

**Longer text query prompt**

```
Take each text query and, without altering its core meaning or
introducing new concepts, enrich it with tasteful descriptive words
or clarifying phrases so that its length is roughly doubled.  Preserve
the original order and casing and return each pair as ''original
query'':  ''augmented longer query'' with no additional commentary.
Example:  ''crowded'' → ''in a dense and crowded scene''
```

---

We observe that BASIC has a small performance drop similar to that of FreeDom, while MagicLens has a larger drop, therefore revealing *a new advantage* of training-free methods (relying on CLIP). Related to the above question, MagicLens presumably has a larger drop due to shorter sentences in its training.

**Relational.** To give every text query an explicit link to its paired reference image, we rewrite all text queries by prepending a short relational clause while keeping the original order and casing intact. Example: ''with fog'' becomes ''`the same object instance depicted in the image but with fog`''. This yields a suite of explicitly relational captions that stress compositional understanding without altering the original intent. The ChatGPT prompt used to convert text queries is presented below.

---

**Relational text query prompt**

```
Rewrite every text query so that it explicitly refers to the same
object instance of the image query, while preserving original words
in the same order.  Simply prepend a concise relational clause, avoid
introducing new concepts or proper names, and return each pair as
''original query'':  ''relational query'', with no additional commentary.
Example:  ''with fog'' becomes ''the same object instance depicted in the
image but with fog''
```

---

We observe that the drop of BASIC is noticeably smaller than that of FreeDom, but also greater than that of MagicLens. This pronounces the benefit of training for composed image retrieval, and a drawback of training-free methods, which is due to the fact that there is no or little relational information during the CLIP pre-training. Nevertheless, BASIC still performs reasonably well.

**Instructional.** We recast every text query as an imperative instruction that tells the system what to do with the reference-image object. Concretely, we prepend an action verb (*e.g.* ''`turn`'', ''`place`'', ''`render`'') and then a clause like ''`the object depicted in the image`'', then append the original words in the original order. Example: ''`in a live action scene`'' becomes ''`adapt the object depicted in the image into a live action scene`''. The ChatGPT prompt used to convert text queries is presented below.

---

**Instructional text query prompt**

```
Rewrite each text query as a concise imperative instruction that
starts with an appropriate action verb, explicitly mentions something
like ''the object depicted in the image'', preserves the original
words in the same order, introduces no new concepts, and output
each pair exactly as ''original query'':  ''instructional query'' with
no commentary.
```

---

The performance pattern across the methods in this case mirrors the relational case.

Additionally, we perform the reverse mapping for 10 manually selected relational queries of CIRR that are converted into shorter and absolute descriptions. For example, ''`Make the dog older and have two birds next to him and make everything look like a painting`'' becomes ''`as a painting of an older dog with two birds`'' and ''`Change the background to trees and remove all but one dog sitting in grass looking right`'' becomes ''`as a single dog sitting in grass with trees in the background`''.

Interestingly, evaluating using the 10 rewritten queries of CIRR (shown in Table E10) to make them non-relational (concise) demonstrates a large performance increase for BASIC from $18\%$ mAP ($10\%$ R@1) to $44.4\%$ mAP ($40\%$ R@1). This signifies that the relational description might not be the best way to describe those queries either.

Table E10: *Making relational CIRR queries concise.* Converting 10 relational CIRR queries into concise, non-relational descriptions substantially improves BASIC, shown via mAP and recall.

| Text-query Variant | mAP | R@1 | R@5 | R@10 | R@50 |
|---|---|---|---|---|---|
| Relational | 18.0 | 10.0 | 40.0 | 60.0 | 80.0 |
| Concise | 44.4 | 40.0 | 60.0 | 70.0 | 90.0 |

### E.10 Retrieval visualizations across BASIC components

Figure E7 and Figure E8 qualitatively demonstrate the effects of centering and the performance of the full BASIC compared to the Text $\times$ Image baseline on *i*-CIR. Notably, even centering alone yields substantial performance gains.

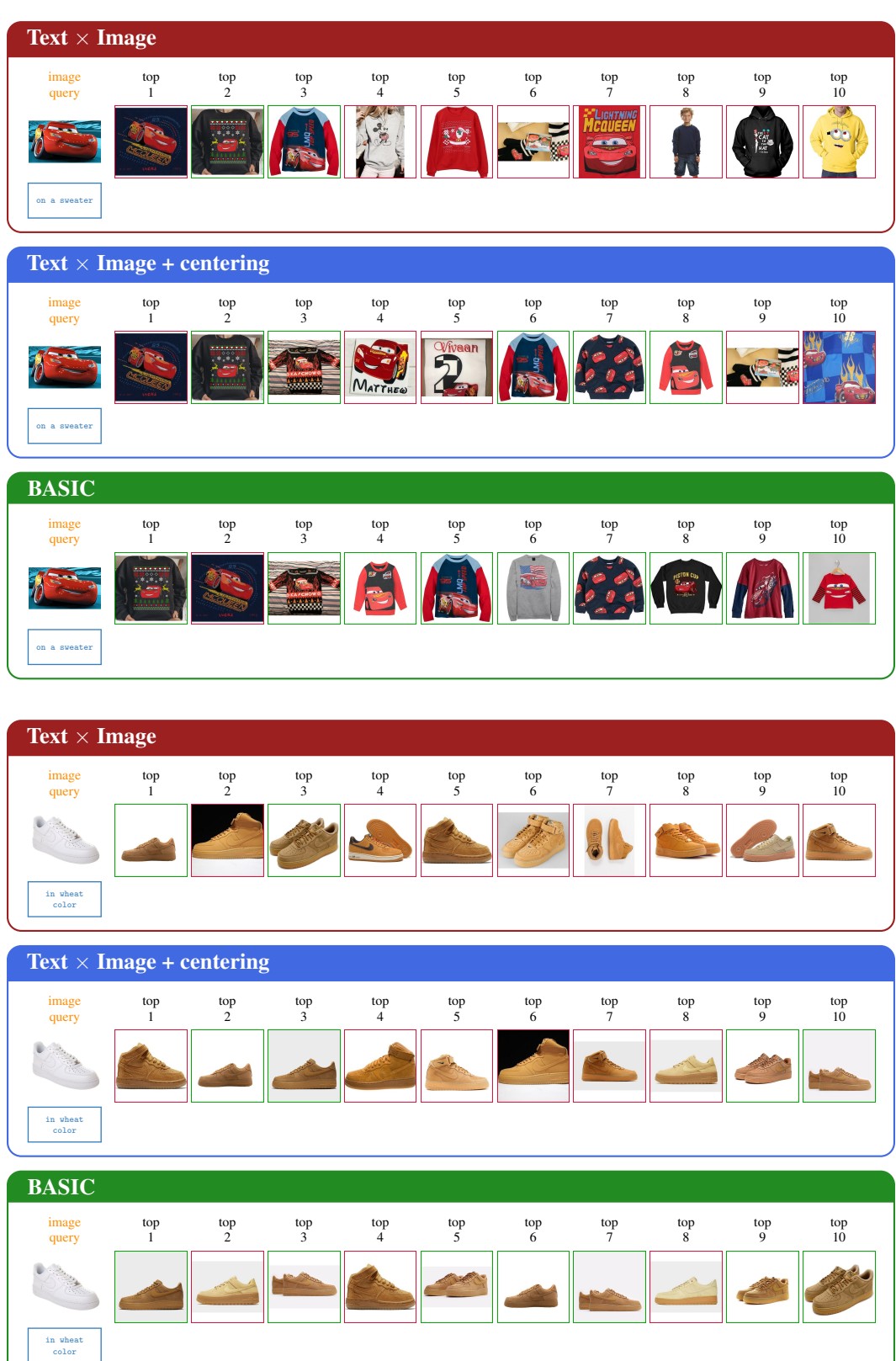

Figure E7: *Retrieval results* for Text × Image, Text × Image with centering, and BASIC on *i*-CIR.

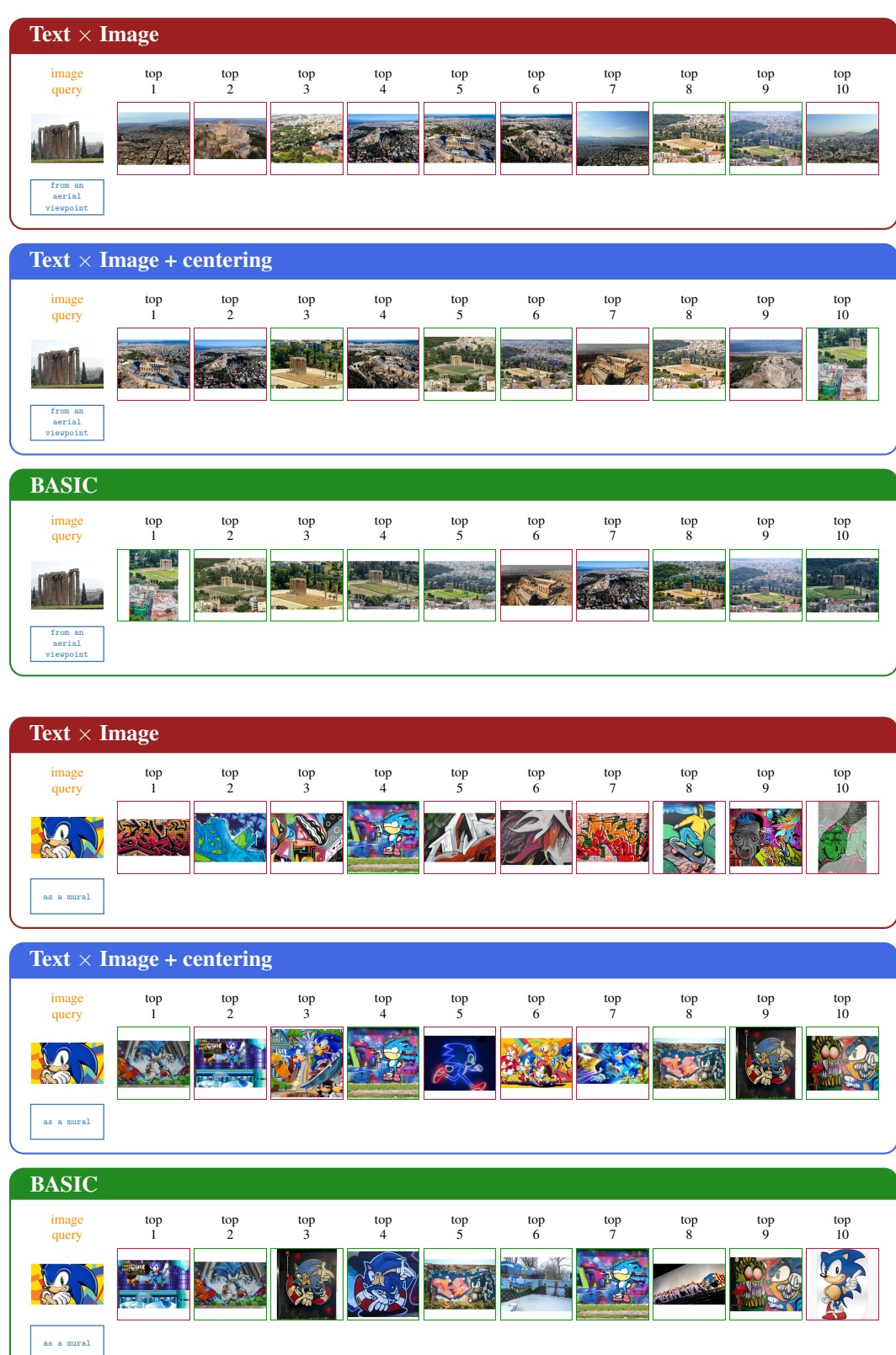

Figure E8: *Retrieval results* for Text × Image, Text × Image with centering, and BASIC on *i*-CIR.

- along with the tetris game cartridge
- arranged beside the bed
- as a balloon
- as a billboard advertisement
- as a bobblehead
- as a bobblehead toy
- as a colored pencil drawing
- as a colorful artistic illustration
- as a colorful digital illustration
- as a colorful sketch
- as a colorized artistic interpretation
- as a decorative piece in bowl
- as a decorative piece on a table
- as a detailed line drawing
- as a digital 2D illustration
- as a digital drawing
- as a digital illustration
- as a digital illustration with a Christmas tree in front
- as a digital photomosaic
- as a digital watercolor painting
- as a figurine
- as a flat vector orthographic digital illustration
- as a freehand monochromatic sketch
- as a giant balloon
- as a graffiti mural
- as a historical monochromatic drawing
- as a huge balloon
- as a huge floating balloon
- as a human size sculpture
- as a human size statue
- as a lego
- as a lego brick figure
- as a lego painting
- as a live action adaptation
- as a mascot costume
- as a miniature
- as a monochromatic sketch
- as a monochrome engraving
- as a monochrome sketch
- as a mural
- as a napkin holder
- as a painting
- as a painting including its surroundings
- as a paper folding art
- as a pen sketch
- as a pencil drawing
- as a photo during a Christmas night
- as a photo during day
- as a photo reflecting on water
- as a photorealistic 3d model wearing a hat
- as a pixel art graffiti mural
- as a plushie
- as a pop art poster
- as a professional mascot costume
- as a professional walk-around mascot costume
- as a public art installation
- as a real size statue
- as a rendered 3D model
- as a sand sculpture
- as a scale model
- as a sculptured edible figure on a cake
- as a silhouette with reflection on water
- as a sketch
- as a street art mural
- as a stylized digital illustration in a framed print
- as a stylized digital-paint illustration
- as a technical schematic design
- as a toy
- as a travel poster art
- as a vector clipart-style digital illustration
- as a vintage postcard
- as a wash drawing or an aquatint
- as a watercolor painting
- as a watercolor style illustration
- as an AI generated digital illustration
- as an action figure
- as an architectural plan
- as an archive photo
- as an emoji
- as an engraving
- as an iconic rooftop sign
- as an oil painting
- as an old archival photo
- as digital illustration
- as japanese art
- as large scale outdoor installation
- as oil painting
- as pencil drawing
- as pixel art
- as seen from a city road
- as vivid pop art
- at day
- at day with blue sky
- at night
- at night with a sculpture in front
- at night with full moon rising behind
- at night with the clock showing around 8:40
- at sunset
- at the evening reflecting on the water
- attached to keys
- being deflated
- being lit
- being wet
- being worn
- broken
- carried by a person
- clipped on a striped bag
- closed
- collapsed and leaned against a wall
- compressed into a flat form
- containing an old photograph
- covered with a black hat
- covered with a decorative embroidered cloth
- covered with a towel
- covered with soil in a garden
- covered with stones
- crowded
- decorated with flags
- disassembled
- displayed on a nightstand
- during day
- during day with blue sky
- during liftoff
- during night
- during sunset
- during the day reflecting on the water
- during the making
- filled with clothes
- filled with coffee
- filled with cosmetics
- filled with dog food
- filled with food leftovers
- filled with orange juice
- filled with sugar
- filled with water
- floating on seawater
- floating on the surface of seawater
- folded
- folded with other clothes
- from a 3D animation movie
- from a drone shot
- from a fully vertical top-down perspective
- from a low-angle perspective
- from a top-down fully vertical perspective
- from an aerial perspective
- from an aerial viewpoint
- from an aerial viewpoint during day
- hanging from a flower bouquet
- hanging from a tree branch
- hanging on a clothesline
- hanging on a clothesline outdoors
- hanging on a drying rack
- hanging on a folding beach chair
- hanging on a t-shirt
- hanging on a wall
- hanging on fireplace guard
- hanging on the drying rack
- hanging on the key holder
- hanging on the wall
- having mud and dirt marks
- held in hand
- held in hand on a beach
- held in hands
- holding a bunch of bananas
- holding a chicken
- holding bars of soap
- holding hair
- holding two espresso cups
- holding various clothes
- illuminated
- in a cabrio version
- in a closed position
- in a comic panel
- in a deep state of rest
- in a desk organizer
- in a filled state
- in a folded state
- in a fully unfolded position
- in a garden pot
- in a laundry tub
- in a manga panel
- in a person's ear
- in a vintage magazine advertisement
- in a vintage magazine page
- in a wet state
- in an old archival photo
- in an open state
- in archive photo
- in black and white color with gum outsole
- in black color
- in front of buildings
- in golden color
- in golden yellow color
- in green color
- in pink color
- in purple color
- in solid yellow color
- in the original video game
- in turquoise color on a black t-shirt
- in wheat color
- in white color
- inside a case
- inside a dishwasher
- inside a plastic bag
- inside a small bag
- inside a wallet
- inside a washing machine
- inside an open suitcase
- inside its original packaging
- inside their case
- lit up with nighttime projection mapping visuals
- lying open
- next to a dog
- next to a plant
- next to coffee beans
- next to other bottles
- next to other fishing boats
- next to other kitchen utensils
- next to other sunglasses
- next to other toys
- next to plants
- next to the real animal it represents
- on a Christmas sweater
- on a backpack
- on a billboard ad
- on a black t-shirt
- on a blue t-shirt
- on a bookshelf
- on a curtain
- on a desk next to a laptop
- on a fireplace
- on a flyer
- on a hat
- on a mug
- on a sweater
- on a t-shirt
- on a t-shirt with a cartoonish style
- on a table next to food
- on a white covered armchair
- on a white t-shirt
- on a yellow t-shirt
- on clothing
- on hanger
- on watermelon shaped dog bed
- outdoors on grass
- packed into its cardboard box
- partially flipped upside down
- placed around a table
- placed in a garden
- placed in a shelf
- placed inside a glass jar
- placed on a metal chair
- placed on a metal tray
- placed on a plastic chair
- placed on a wall-mounted holder
- placed on top of clothes
- plugged in and glowing light
- positioned in front of an office desk
- printed on a white pillow
- reflected in a mirror
- reflecting on water
- riding a white dinosaur
- rolled-up in a bag
- sitting on a chair
- smashed
- spread on the bed
- spread out on a rocky ground
- stacked on shelves
- stained with coffee
- stored in a nightstand
- stored in a wordrobe
- surrounded by chairs
- surrounded by metallic kitchenware
- the mountain depicted on the packaging
- tied around a ponytail
- unassembled
- under running water
- used as a seat cushion
- while charging
- while flying
- with a candle on top
- with a close up on the bridge
- with a crowd in front
- with a darker mask and a hoodie
- with a dog inside
- with a film around
- with a glass on top
- with a hair claw clip clipped onto it
- with a lot of people around
- with a man holding it
- with a man proposing to a woman in front
- with a more rounded handle
- with a numeric keypad
- with a patio umbrella open
- with a person pretending to hold it up
- with a person sitting on
- with a real car in front
- with a seaside background
- with a towel draped over
- with an external flash attached
- with bananas placed on it
- with big waves crashing against the shoreline
- with blond fur
- with brown or dark beige leather
- with candies on top
- with doors open
- with drawers open
- with fireworks
- with fog
- with people around
- with people in front
- with people standing on
- with seat cushion
- with snow
- with the front wheel removed
- with the tongue out
- with various ornaments displayed on top
- with waves crashing against it
- with yellow shoelaces
- without cushion
- without seat cushion and pillows
- without sticky paper notes but filled with pens and markers
- without sunset
- worn
- worn around a dog's neck
- worn around a human's neck
- worn by a child
- worn by a dog
- worn by a girl
- worn by a person
- worn by someone
- worn on a face sculpture
- worn on the ankle
- worn with colorful swimwear

Figure E9: *The unique text queries of* i-*CIR dataset*, listed in alphabetical order. These queries reflect diverse compositional transformations across appearance, context, attributes, viewpoint, and more.

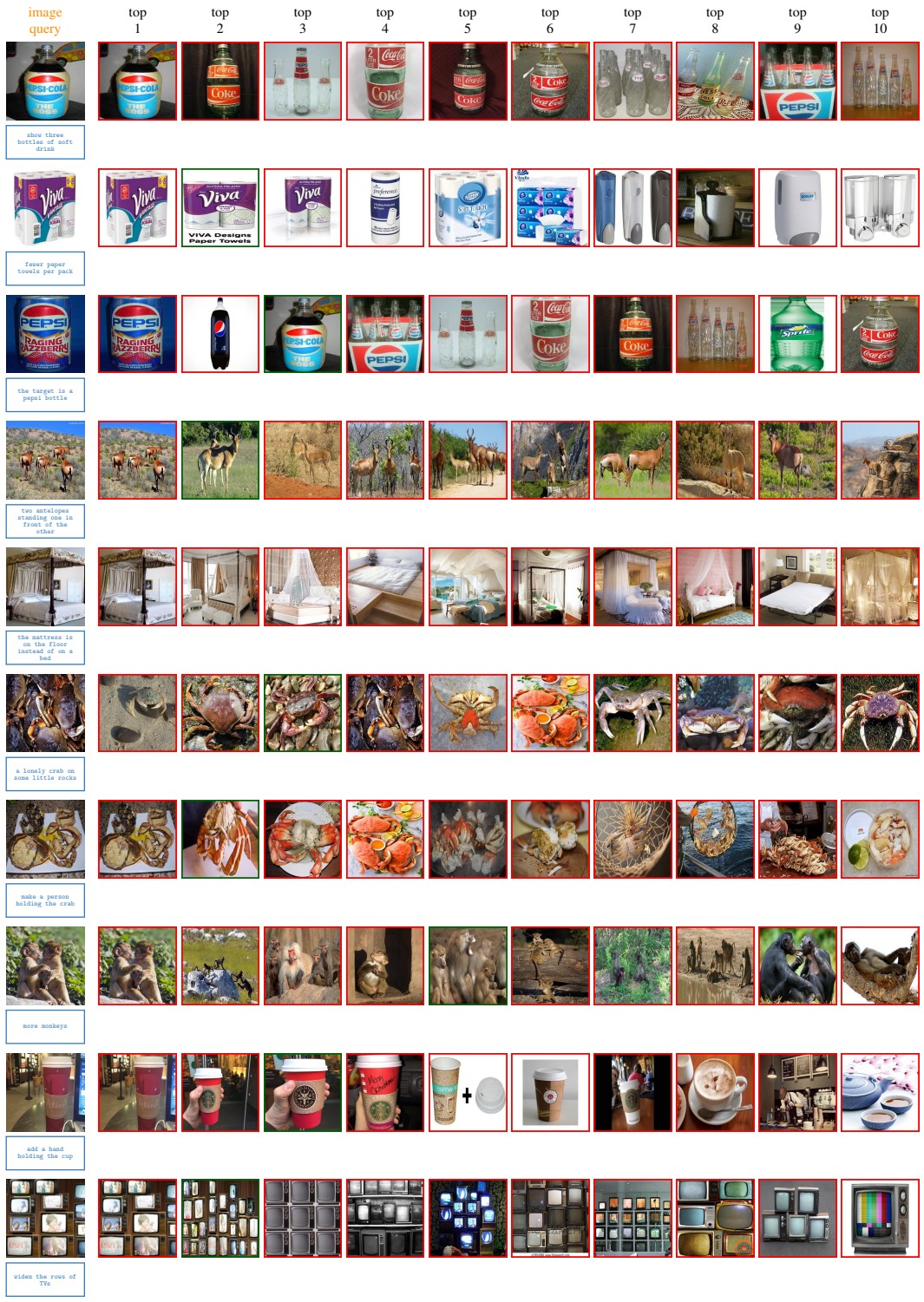

Figure E10: *CIRR retrieval results using Text × Image, illustrating dataset limitations.* Each row shows a composed query with its top-10 retrieved results. Green boxes indicate correct retrievals, red denote incorrect ones. Several cases reveal issues with ground-truth annotations and the limited role of the image query.

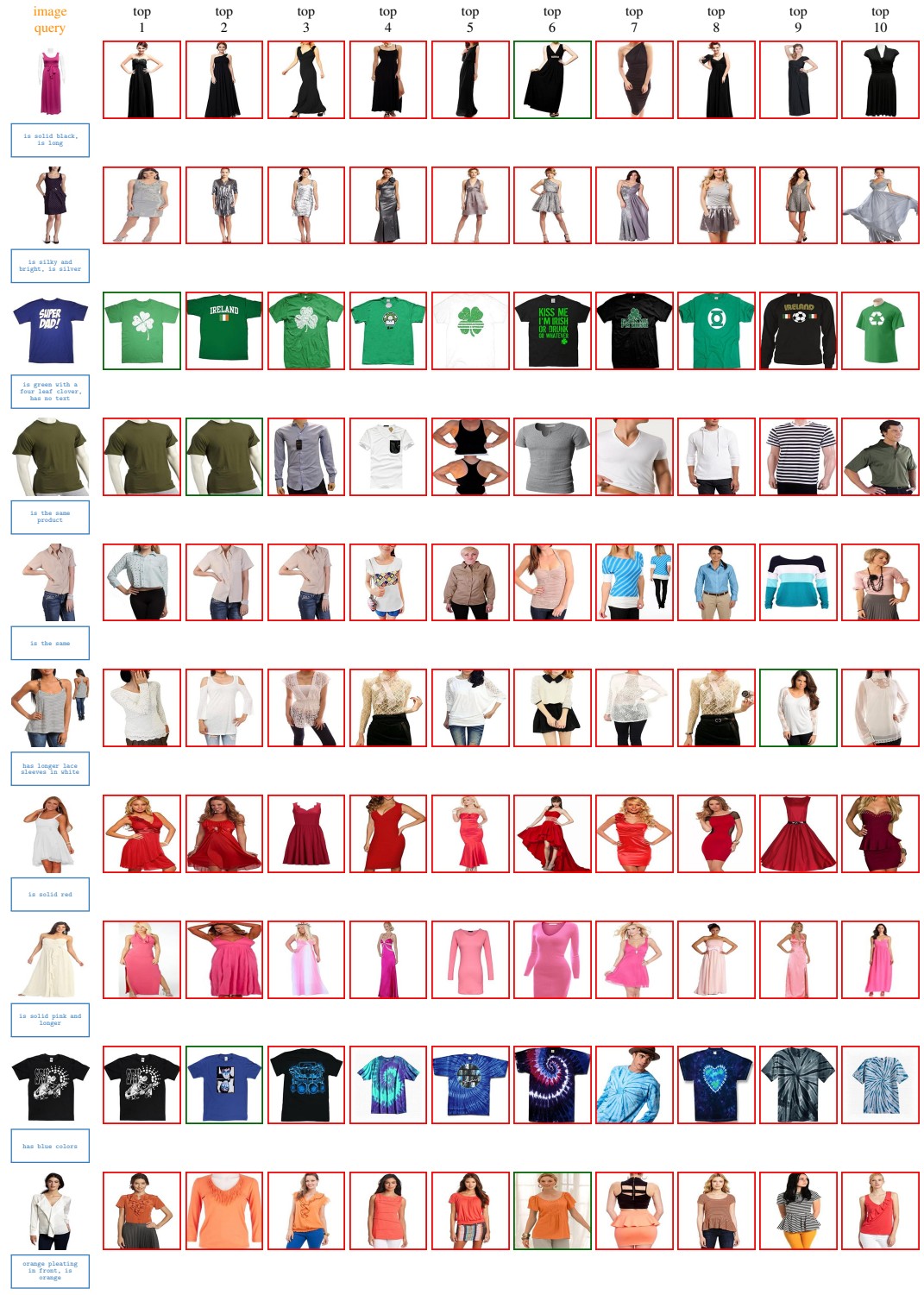

Figure E11: *FashionIQ retrieval results using Text × Image, highlighting dataset limitations.* Each row shows a composed query with image (left) and text (side). The top-10 retrieved results are shown left to right. Green boxes mark correct retrievals, red boxes mark incorrect ones. The figure highlights false negatives and limited reliance on the image query.

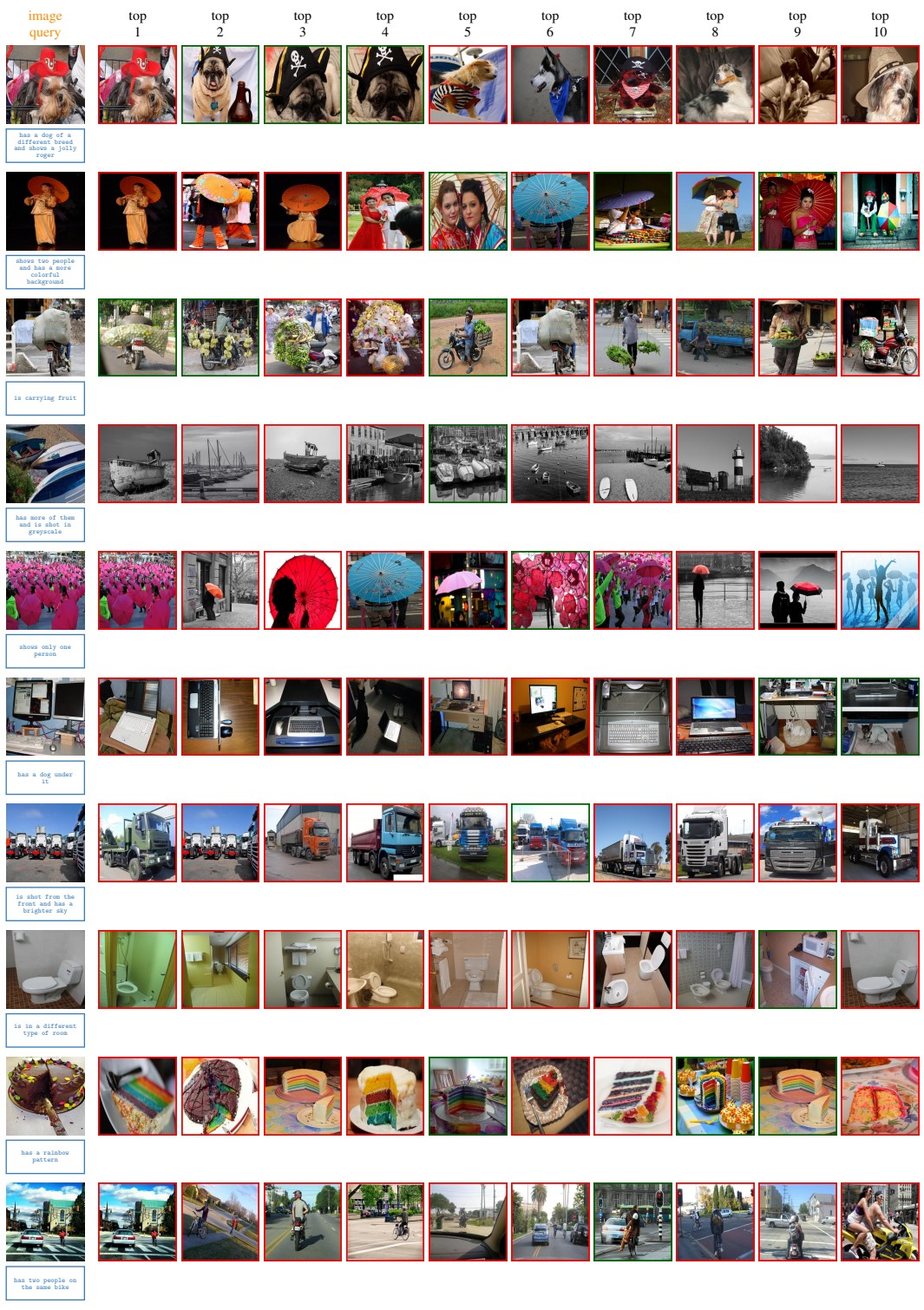

Figure E12: *CIRCO retrieval results using Text × Image, highlighting dataset limitations.* Each row shows a composed query with image (left) and text (side). The top-10 retrieved results are shown left to right. Green boxes mark correct retrievals, red boxes mark incorrect ones. The figure highlights false negatives and the instructional nature of the text query.

