# OpenReview forum: "Instance-Level Composed Image Retrieval"
_NeurIPS.cc/2025/Conference — NeurIPS 2025 poster_

### Official Review · Reviewer_FuSR · 2025-06-23

**Clarity:** 3
**Significance:** 3
**Originality:** 3
**Rating:** 5
**Confidence:** 4

**Summary:**

This paper focuses on composed image retrieval (CIR), where given an image and a textual condition, images that fulfill both have to be retrieved. Authors highlight that there are no high-quality datasets to train and evaluate CIR systems, so they propose their own evaluation dataset called i-CIR. To build the dataset, they use a semi-automatic approach, combining Vision-Language Models for image collection and manual annotation. The dataset contains various types of hard-negative images which makes the retrieval challenging. Afterwards, authors also propose their own CIR system, called BASIC, avoiding any training stage (hence they use the term of zero-shot CIR). BASIC is a pipeline systems that leverages dual-encoder VLMs such as CLIP. It has various modules, namely i) centering for bias removal, ii) projection onto semantic subspace, iii) image query expansion, iv) contextualization of text queries, and v) score normalization and fusion. BASIC is evaluated on i-CIR; but also on some other reference datasets for CIR, such as ImageNet-R, NICO++, MiniDN and LTLL. Results show the overall superior performance of BASIC compared to several baselines and SOTA approaches. Finally, authors also run an ablation study measuring the contribution of each of the components of their pipeline for BASIC.

**Questions:**

1. Suggestion: this is entirely in authors' hands, but I would recommend to change the title of the paper and write something more specific. "Instance-Level Composed Image Retrieval" is too generic in my opinion.
2. Question: in Line 52, authors say that "The image-to-image similarity is enhanced through a projection learned not in the image space, but...". Isn't this a contradiction with the claim that BASIC is a training-free approach? In Section 4, I didn't see that the projection is learned.
3. Related Work: this section is very short and lacks a critical view of the literature. Authors should position BASIC in the "Methods" paragraph, highlighting the limitations of current methods and where BASIC makes a contribution. Similarly, in the "Datasets" paragraph, authors should discuss how i-CIR is different from existing approaches. I think this is already done in Section 3.4, so I would move that information to Related Work and that way, Section 3 would automatically be motivated.
4. Question: in Line 125, authors say "The neighbors retrieved from LAION are classified as visual, textual and composed hard negatives". How do you do that? Manually? You use a classifier? Please, specify.
5. Question: in Line 128, we can read "Annotators then manually inspect..." How many annotators? How do you recruit them? Please, give details about the annotation process, which is very important.
6. Question: Line 200: "To ensure scalability and generalization, we compute $\mu^v$  using a large external dataset $X^v$, e.g. LAION." As far as I understood, the images of i-CIR are also collected from LAION. Is this fair? Are not you using the statistics of the test dataset for the benefit of your system?
7. Typo: Line 284: a) and (b) are used differently. Please, either use (a) and (b) or a) and b).
8. Question: Line 289: "The only exceptions are fashion, ... and art, ..." Any explanation for this? I think it would be interesting to analyze some examples and try to understand what's going on those categories. It may show potential ways to improve BASIC.
9. Suggestion: Table 2. During the text, authors use line numbers to refer to this table, but there are not line numbers in the table. Please, add line numbers to make the reading easier.

**Ethical Concerns:**

["NO or VERY MINOR ethics concerns only"]

**Final Justification:**

My major concern, how the hyperparameters of their system were tuned, has been suitably addressed by the authors, so I think that the paper should be accepted. The paper has many strong points in my opinion, as I listed in my review.

**Limitations:**

Yes

**Paper Formatting Concerns:**

No concerns.

**Quality:**

3

**Strengths And Weaknesses:**

**Strengths:**
1. The i-CIR dataset is an important contribution for CIR research. The semi-automatic process designed for the dataset creation can be leveraged to update and extend the dataset in the future.
2. The proposed BASIC system is also very interesting. It shows that pre-trained VLMs such as CLIP can provide very rich representations which can be enhanced and used for various scenarios without any further training. This is specially interesting for problems such as CIR, where high-quality annotated data is hard to obtain.
3. The results obtained in various CIR datasets are promising for BASIC.

**Weaknesses:**
1. My main concern is about BASIC development methodology. BASIC has many hyperparameters that have to be tuned, as explained by authors in Section 4. However, authors do not detail how they set the values of those hyperparameters. Indeed, in the supplementary materials, Section S3.1, authors state (caption of Figure S3): "Effect of key hyperparameters on mAP (%) across all considered datasets". Should we understand that the hyperparameter search is performed on the test datasets? That is my conclusion, since I cannot find any development dataset in the paper (nor in the supplementary materials). Furthermore, the ablation studies provided in Section 5.3 are also performed on the test datasets. This means that authors could test and select the components of BASIC directly on the test sets and use the components that maximize the performance. Those two considerations make me suspect that authors have developed BASIC on the test dataset directly, which is not methodologically correct, since they are "optimizing" their system for the test datasets. My score to the paper has been heavily influenced by those methodological flaws, but if authors can explain and correct those important issues, I would happily increase my score.
2. The other concerns I have are lighter. I will list them as questions and suggestions for the authors.

---

> ### Author Rebuttal · Authors · 2025-07-31
>
> We thank **Reviewer FuSR** for emphasising the key strengths of our work:
> (i) a **carefully engineered, extensible i-CIR benchmark**,
> (ii) a **genuinely training-free** BASIC pipeline that **unlocks pre-trained VLMs for CIR**, and
> (iii) **consistently promising results across five datasets**.
> Below we clarify the remaining concerns—most notably hyper-parameter (HP) selection—and address every question in turn.
>
> ---
>
> ### Hyper-parameter Tuning is *Never* Done on the Test Sets
>
> We *apologize for any confusion* and we would like to *thank the reviewer for this comment*. We have omitted some information in the original submission, which we provide herein. During the development of the BASIC method, we reserved a small slice of the i-CIR crawl as a *development* set; none of those images appear in the final test benchmark. i-CIR_{dev} consists of 15 object instances, 92 composed queries, and 45K images in total. All BASIC HPs—contrastive scaling α, number of PCA components k, and Harris regularization λ—were fixed on this split and then frozen. The purpose of Fig. S3 is to show that BASIC’s performance curve is stable over a reasonably wide neighbourhood of hyperparameters (HP). To further demonstrate this robustness, we now perform a cross-dataset check. We set each dataset in turn as the development set and evaluate on the others in a “leave-one-out” logic.
> Even though the largest performance appears on the diagonal, there is no large difference compared to tuning on the development dataset (or any other dataset). A small exception is LTLL, presumably due to its limited unique textual queries (2) and its very small size.
> We shall definitely *add this leave-one-out table* in the supplementary material of the paper. We shall also explicitly mention the development set that was used for tuning BASIC's HP.
>
> | Dev → Test (HPs) | i-CIR | ImageNet-R | NICO++ | MiniDN | LTLL | Avg. |
> |------------------|------:|-----------:|-------:|-------:|-----:|-----:|
> | i-CIR (α=0.2,k=300,λ=0.1)      | **47.41** | 31.43 | 31.06 | 38.89 | 41.90 | 38.14 |
> | ImageNet-R (α=0.4,k=200,λ=0.1) | 46.34 | **32.63** | 31.39 | 38.49 | 40.41 | 37.85 |
> | NICO++ (α=0.2,k=200,λ=0.1)     | 46.35 | 32.55 | **31.85** | 39.81 | 41.32 | 38.38 |
> | MiniDN (α=0.2,k=200,λ=0.1)     | 46.35 | 32.55 | 31.85 | **39.81** | 41.32 | 38.38 |
> | LTLL (α=0.0,k=300,λ=0.1)       | 46.22 | 28.96 | 30.16 | 37.65 | **42.44** | 37.09 |
> | i-CIR_{dev} (α=0.2,k=250,λ=0.1) | 47.39 | 32.13 | 31.65 | 39.58 | 41.38 | 38.42 |
>
> ---
>
> ### ``Learned projection'' vs Training-Free
> We use the term “learned” as commonly used for PCA which is data-dependent.
> Therefore, the projection in BASIC is not learned via supervision or optimization, but rather *computed analytically* from two fixed text corpora:  (C_+) (object-centric terms) and (C_-) (style/context terms). As described in section 4 (Eq. 2), we construct a contrastive covariance matrix and perform eigendecomposition to extract the top positive eigenvectors, which define the projection space. This process involves **no training, backpropagation, or labeled data**, and is consistent with BASIC’s training-free design. To avoid misinterpretation, we will revise the sentence.
>
> ---
>
> ### LAION Statistics Do Not Leak Test Information
>
> LAION is very large; i-CIR overlaps with it by <0.001 %. Moreover, each instance of i-CIR has its own tailored database consisting of the explicitly curated hard negatives. Using the global CLIP LAION mean therefore does not leak any test set statistics, yet gives a domain-agnostic center that transfers well across all datasets. To substantiate this, we computed the center on ImageNet-1K and simply replaced the LAION mean with this new one. BASIC on i-CIR achieves 46.67% mAP with the ImageNet mean versus 47.39% mAP with the LAION mean, a negligible drop of \<0.8 mAP. This small gap confirms that our results are not artefacts of test leakage, but are the result of the BASIC design.
>
> ---
>
> ### Details of i-CIR Construction
>
> The labelling of the neighbours retrieved from LAION is a *pseudo-labelling*. For example, if the seed sentence is a textual description of the instance like “Temple of Poseidon”, then the retrieved neighbours should depict exactly this instance OR in a worse case just “temples”. Thus, these images are pseudo-labeled/classified as visual hard negatives. This is what L126 “for cases (i \& ii), (ii), and (iv \& v), respectively” was trying to describe. We shall rephrase for more clarity. The annotators were trained workers performing a well-defined annotation task. They completed a one-week training on composed image retrieval task, as also on copyright and sensitive-content removal. We shall include these details in the supplementary material. Thanks for the suggestion.
>
> ---
>
> ### Why BASIC Trails in the Fashion Category
>
> MagicLens is pre-trained on 36.7 M triplets dominated by words such as “brand” and “style” (according to their Fig. 5 b), which explains its narrow advantage in fashion.  Outside that niche, BASIC leads by up to 34.8 % mAP (household category). We will add exemplar failure cases per visual category in the supplementary material.
>
> ---
>
> ### Other Questions & Promised Edits
>
> We will consider a more specific variant of the title. We will move some info from the i-CIR comparison (current section 3.4) into section 2 and add some comments positioning BASIC among composed image retrieval methods. We will correct any typos.

---

> > ### Comment · Reviewer_FuSR · 2025-08-04
> >
> > I would like to thank the authors for the detailed answer to my concerns. The main weakness I identified (hyperparameter tuning) is now perfectly explained. I think authors should include all this information in the paper. I would mention in the main paper the existence of a development set, and detail that development set and the process of HP tuning in the appendices. The new table with the cross-dataset experiments also adds value to the BASIC system, so I encourage authors to add it to the paper also. Regarding all the other questions, I am happy with the answers of the authors. I hope authors will make the necessary changes to the paper, as promised in the rebuttal. Accordingly, I will raise my score.

---

> > > ### Author Response · Authors · 2025-08-04
> > >
> > > We sincerely thank the reviewer for the **constructive follow-up** and are glad that the **concerns around hyperparameter tuning are now resolved**. As suggested, we will:
> > >
> > > - Clearly mention the *existence and composition of the i-CIR development set* in the main paper.
> > > - Include full details of the *tuning process* and *cross-dataset results*.
> > > - Revise and clarify the annotation process, pseudo-labeling, and rest of the points mentioned above.
> > >
> > > We truly appreciate your **helpful feedback** and **updated score**.

---

### Official Review · Reviewer_pyz4 · 2025-06-29

**Clarity:** 3
**Significance:** 2
**Originality:** 2
**Rating:** 4
**Confidence:** 4

**Summary:**

This paper introduces i-CIR, a new instance-level composed image retrieval benchmark designed to better reflect real-world retrieval scenarios where a query combines both an image and a text-based modification. Unlike existing CIR datasets that often rely on class-level semantics or automatically constructed triplets, i-CIR emphasizes retrieving images of the same object instance under textual modifications, making it both compact and challenging. To address the lack of training data, the authors propose a training-free method called BASIC, which leverages pre-trained vision-language models to separately compute image-to-image and text-to-image similarities, followed by a fusion step. The approach sets new state-of-the-art performance on i-CIR and other CIR datasets, demonstrating strong composition abilities without requiring additional training.

**Questions:**

1.The proposed BASIC method relies entirely on fixed heuristics and manually designed components (e.g., projection, score fusion), without any learnable modules. How would the method adapt to domains that differ significantly from those seen during CLIP pretraining? Have the authors considered hybrid or lightly fine-tuned variants?
2.The dataset appears to include mainly short or mid-length modification texts. Could the authors comment on how their method would handle longer, more compositional or relational queries that require fine-grained semantic understanding?
3.Most text queries are phrased at attribute or scene level. Is the model capable of handling relational or comparative queries, e.g., “the same object as in A but rotated and with a different texture”?
4.The experiments do not include comparisons against more recent CIR methods from 2024–2025. Are there any strong baselines published after 2023 that could be added to better position the contribution?
5.While the projection and fusion modules are effective, they depend on manually selected corpora and static subspaces. Could the authors clarify how robust these choices are across different instances and domains? Is there a way to make them more adaptive?

**Ethical Concerns:**

["NO or VERY MINOR ethics concerns only"]

**Final Justification:**

After carefully reading the author's response, for Q1, the authors carefully analyzed the effectiveness of the Tran-free approach and proposed a more targeted and compatible expansion direction, namely exploring hybrid text embedding strategies. I strongly agree with the author's analysis. For Q2 and Q3, the authors conducted a controlled study, automatically converting the original query into a longer, relational, and imperative form. Although BASIC is a training-free method, it demonstrates good robustness across variants compared to the baseline, validating its generalization ability in natural language transformation. For Q4, the authors provided additional experiments, strengthening the persuasiveness of their article. For Q5, the authors demonstrated the cross-domain robustness of their universal projection corpus. Overall, the author provided a clear answer, which resolved my confusion and enhanced the persuasiveness of the article.

**Limitations:**

Yes

**Paper Formatting Concerns:**

No major formatting issues identified. The paper follows NeurIPS guidelines appropriately.

**Quality:**

3

**Strengths And Weaknesses:**

Strengths:
1.The motivation of this paper is clear. Figures 1 and 3 powerfully demonstrate the deficiencies of the existing CIR datasets in instance-level definition and negative sample construction, highlighting the necessity and challenge of constructing the i-CIR dataset. Section 3 elaborates on the complete process from instance definition to negative sample screening, demonstrating a high level of annotation quality and engineering controllability.
2.Instance-level CIR is closer to real application scenarios such as product search. The BASIC method proposed by the author does not require training, and the fusion strategy is concise and efficient, especially suitable for scenarios with low resources or high deployment flexibility requirements.
3.Performance evaluations on multiple CIR benchmarks (such as i-CIR, ImageNet-R, MiniDN, LTLL) demonstrate the method's wide applicability and stable performance across different categories and tasks. The ablation experiments in Table 2 clearly verified the effectiveness of key modules such as text contextualization and semantic projection, demonstrating good design logic and module contributions.
Weakness:
1.The proposed BASIC method relies entirely on fixed heuristics and manually designed components (e.g., projection, score fusion), without any learnable modules. How would the method adapt to domains that differ significantly from those seen during CLIP pretraining? Have the authors considered hybrid or lightly fine-tuned variants?
2.The dataset appears to include mainly short or mid-length modification texts. Could the authors comment on how their method would handle longer, more compositional or relational queries that require fine-grained semantic understanding?
3.Most text queries are phrased at attribute or scene level. Is the model capable of handling relational or comparative queries, e.g., “the same object as in A but rotated and with a different texture”?
4.The experiments do not include comparisons against more recent CIR methods from 2024–2025. Are there any strong baselines published after 2023 that could be added to better position the contribution?
5.While the projection and fusion modules are effective, they depend on manually selected corpora and static subspaces. Could the authors clarify how robust these choices are across different instances and domains? Is there a way to make them more adaptive?

---

> ### Author Rebuttal · Authors · 2025-07-31
>
> We thank **Reviewer pyz4** for the review and for explicitly recognising several core strengths of our submission:
> (i) a **clear motivation** punctuated by Figs. 1 & 3 and the full annotation pipeline in section 3,
> (ii) instance-level CIR is a **real-world task**,
> (iii) BASIC is a training-free method with a **concise and efficient fusion strategy**, especially suitable for scenarios with low resources or high deployment flexibility requirements,
> (iv) **strong, broad empirical results** across i-CIR, ImageNet-R, MiniDN, and LTLL,
> (v) ablations in Table 2 **verify the logic and value of each component** of BASIC.
>
> ---
>
> ### Adaptability of a Fixed, Training-Free BASIC
>
> BASIC purposefully avoids task-specific learning so that it *inherits* CLIP’s open-domain coverage instead of narrowing it down. BASIC relies on manually designed components, each *motivated by intuitive principles*: semantic projection isolates object-level semantics, Harris fusion balances dual-modality relevance, and contextualization mitigates CLIP’s sensitivity to phrasing. These components are carefully chosen to remain training-free and interpretable, achieving SOTA performance across *diverse and shifted domains* and showing *strong robustness* without requiring adaptation. Nonetheless, as the reviewer rightly notes, BASIC also inherits CLIP’s limitations—especially in the textual domain, where phrasing, abstraction level, or relational structure can affect performance. For visual inputs, CLIP’s vast pre-training set offers strong coverage. While fine-tuning could improve in-domain performance, it risks reducing generality.
>
> A more targeted and compatible extension would be to explore **hybrid text embedding strategies**. For example, training a lightweight text encoder for relational or instructional queries, related to the following reviewer’s comment, or fine-tuning a contextualization-like embedding to reduce sensitivity and inference overhead. These directions remain promising extensions that target specific constraints (e.g., CLIP sensitivity to query wording or time overhead of contextualization).
>
> ---
>
> ### Longer, Relational, or Instructional Text Queries
>
> Thank you for the comment, which led us to the following *insightful analysis*. We assume that a text query can be almost equivalently formulated in different ways, e.g. **concisely** (“engraving”), in a **longer** sentence (“as a vintage stylish engraving”), in a **relational** way (“the same landmark as the one depicted in the image but as an engraving”), and in an **instructional** way (“engrave the landmark depicted in the image”). In the context of i-CIR, we refer to those cases as *original*, *longer*, *relational* and *instructional*. The queries of i-CIR are brief and comprehensive, while existing datasets like CIRR and CIRCO are often relational or/and instructional, while having larger lengths.
>
> To probe BASIC’s robustness against such text query styles we automatically rewrite (via ChatGPT) all text queries of i-CIR, evaluate mmAP and present the results in the Table below (Δ is the relative change with respect to the original row). This is an automated process that allows us to perform such an insightful experiment despite the possibility of noise. Additionally, we perform the reverse mapping for 10 relational queries of CIRR that are converted into shorter and absolute descriptions. Results are discussed below.
>
> | Text-query Variant | BASIC mmAP | BASIC Δ % | FreeDom mmAP | FreeDom Δ % | MagicLens mmAP | MagicLens Δ % |
> |--------------------|-----------:|----------:|-------------:|------------:|---------------:|--------------:|
> | Original (concise) | 47.39 | —  | 25.14 | —  | 32.00 | — |
> | Longer             | 45.02 | −5 | 23.91 | −5 | 26.90 | −16 |
> | Relational         | 38.95 | −18| 18.99 | −25| 29.34 | −8 |
> | Instructional      | 37.47 | −21| 18.31 | −27| 28.85 | −10 |
>
> **Longer.** We expand all text queries by adding 1–4 extra words while preserving their meaning. On average the phrases grow from 4.4 → 7.9 words—an increase of 3.5 words or +79 %, effectively almost doubling their length. The median rises from 4 to 8 words, the shortest query goes from 1 to 4 words, and the longest from 13 to 15. Example: “crowded” becomes “in a dense and crowded scene”. The ChatGPT prompt used to convert the text queries is:
>
> ```Take each text query and, without altering its core meaning or introducing new concepts, enrich it with tasteful descriptive words or clarifying phrases so that its length is roughly doubled. Preserve the original order and casing and return each pair as ‘original query’ : ‘augmented longer query’ with no additional commentary. Example: ‘crowded’ → ‘in a dense and crowded scene’```
>
> We observe that BASIC has a *small drop in performance* similar to that of FreeDom, while MagicLens has a *larger drop*, therefore revealing a new advantage of training-free (relying on CLIP) methods. Related to the above question, MagicLens presumably has a larger drop because of *shorter form sentences in its training*.
>
> **Relational.** To give every text query an explicit link to its paired reference image, we rewrite all text queries by prepending a short relational clause while keeping the original order and casing intact. Example: “with fog” becomes “the same object instance depicted in the image but with fog”. This yields a suite of explicitly relational captions that stress compositional understanding without altering the original intent. The ChatGPT prompt used to convert the text queries is:
>
> ```Rewrite every text query so that it explicitly refers to the same object instance of the image query, while preserving original words in the same order. Simply prepend a concise relational clause, avoid introducing new concepts or proper names, and return each pair as ‘original query’ : ‘relational query’, with no additional commentary. Example: ‘with fog’ becomes ‘the same object instance depicted in the image but with fog’```
>
> We observe that the drop of BASIC is noticeably smaller than that of FreeDom, but also *larger than that of MagicLens*. This pronounces the benefit of training for composed image retrieval, and a drawback of training-free methods, which is due to the fact that there is no or little relational information during the CLIP pre-training. Nevertheless, BASIC still performs reasonably well.
>
> Interestingly, evaluating using the 10 rewritten queries of CIRR (shown in the Table below) to make them non-relational (concise) demonstrates a *large performance increase* for BASIC from 18% mAP (10% R@1) to 44.4% mAP (40% R@1). This signifies that the relational description might not be the best way to describe those queries either.
>
> | Text-query Variant | mAP | R@1 | R@5 | R@10 | R@50 |
> |--------------------|----:|----:|----:|-----:|-----:|
> | Relational | 18.0 | 10.0 | 40.0 | 60.0 | 80.0 |
> | Concise    | 44.4 | 40.0 | 60.0 | 70.0 | 90.0 |
>
> | Original relational caption of CIRR | Concise, absolute rewrite of caption |
> |-----------------------------|---------------------------|
> | “Make the dog older and have two birds next to him and make everything look like a painting.” | as a painting of an older dog with two birds |
> | “Change the background to trees and remove all but one dog sitting in grass looking right.” | as a single dog sitting in grass with trees in the background |
> | “Remove the mannequin and change the necklace to a bracelet.” | as a standalone bracelet without a mannequin |
>
> **Instructional.** We recast every text query as an *imperative* instruction that tells the system what to do with the reference-image object. Concretely, we prepend an action verb (e.g. “turn”, “place”, “render”) and then a clause like “the object depicted in the image”, then append the original words in the original order. Example: “in a live action scene” becomes “adapt the object depicted in the image into a live action scene”. The ChatGPT prompt used was:
>
> ```Rewrite each text query as a concise imperative instruction that starts with an appropriate action verb, explicitly mentions something like ‘the object depicted in the image’, preserves the original words in the same order, introduces no new concepts, and output each pair exactly as ‘original query’ : ‘instructional query’ with no commentary.```
>
> The performance pattern across the methods in this case mirrors the relational case.
>
> ---
>
> ### Newer Baselines
>
> The experiments already include comparisons against methods from 2024-2025. Specifically, FreeDom is a WACV 2025 oral, MagicLens is an ICML 2024 oral, CIReVL (supplementary Table S2) is an ICLR 2024 poster publication. Following the suggestion of Reviewer rzCM, we have now included **CoVR-2** (TPAMI 2024) and **MCL** (ICML 2024). BASIC consistently outperforms both methods across 5 CIR datasets.
>
> ---
>
> ### Robustness of Projection &amp; Fusion Corpora
>
> As described in supplementary § S2, we use ChatGPT to construct a generic object-centric corpus (C_{+}) that defines the projection subspace. The corresponding stylistic/contextual corpus (C_{-}) is detailed in our response to Reviewer przL. These corpora are fixed and shared across all datasets and domains, enabling training-free and scalable retrieval without task-specific tuning. To assess robustness, Tables S5 and S6 of the supplementary report results using domain-specific corpora (e.g. mobility-, landmark-, or fashion-focused). These dedicated variants yield only a +0.4 mAP improvement on average, indicating that the generic corpus is already effective across diverse datasets. At the same time, the projection matrix can be recomputed from any text pool in *under 1 second on CPU*, making the method trivially adaptable at deployment time if domain-specific knowledge is available. A natural direction for future work is to automatically construct such corpora by parsing the dataset or its metadata using a large vision-language model (LVLM), which could eliminate the need for manual corpus design entirely.

---

> > ### Author Response · Authors · 2025-08-05
> >
> > We would greatly appreciate it if the reviewer could *review our rebuttal* and let us know *if our responses address their concerns*. We would be glad to provide additional clarification or engage in further discussion as needed.

---

> ### Comment · Reviewer_pyz4 · 2025-08-06
> **Response**
>
> I thank the authors for their detailed responses to my concerns. The main confusion I raised: how the model handles longer, more compositional or relational queries with fine-grained semantic understanding, is now perfectly explained. After re-evaluation, I will increase my rating accordingly.

---

> > ### Author Response · Authors · 2025-08-06
> >
> > We **sincerely thank you** for revisiting your evaluation and for **raising your overall score**. Your question on longer, compositional, and relational queries prompted additional experiments, which uncovered **useful insights** we will definitely include in the paper. We believe this analysis will *benefit the broader CIR community*. We thank you once more.

---

### Official Review · Reviewer_rzCM · 2025-06-30

**Clarity:** 4
**Significance:** 3
**Originality:** 4
**Rating:** 5
**Confidence:** 5

**Summary:**

This paper makes two significant contributions to composed image retrieval: a new benchmark and a training-free approach. The large-scale, high-quality benchmark effectively addresses ambiguity issues present in prior datasets, benefiting from a well-designed semi-supervised construction pipeline with human verification. The proposed training-free method, utilizing late-interaction between image and text features, achieves remarkable performance across various benchmarks, even outperforming several training-based methods. Extensive ablation studies and experiments further reinforce the paper's claims and contributions.

**Questions:**

1. The authors should include the performance of some strong fine-tuning methods on large-scale datasets, especially those leveraging better text encoders, such as CoVR-2 [a] and MCL [b]. This would provide a more comprehensive comparison with state-of-the-art training-based approaches.
2. In Figure S9, a significant portion of the top-1 retrieval results appears to be the reference image itself. This observation seems to contradict the explanation in Line 245, "text query often provides sufficient information to resolve the task independently." Could the authors elaborate on the specific challenges or limitations of their solution on popular benchmarks (CIRR, Fashion-IQ, CIRCO) that lead to this phenomenon? While acknowledging the known issues with these datasets, the extremely low performance of "BASIC" on these benchmarks raises concerns about the trustworthiness and generalizability of the proposed solution. Furthermore, given that CoLLM [c] introduces refined versions of the CIRR and Fashion-IQ benchmarks, it would be valuable to evaluate the method on these refined datasets to determine if similar issues persist.
3. What are the speed and storage requirements of “BASIC” during retrieval, especially when compared to a single dot-product operation? Providing these metrics would offer crucial insights into the practical efficiency and deployability of the proposed training-free method.


[a] Ventura, Lucas, et al. "CoVR-2: Automatic Data Construction for Composed Video Retrieval." TPAMI 2024.


[b] Li, Wei, et al. "Improving context understanding in multimodal large language models via multimodal composition learning." ICML. 2024.


[c] Huynh, Chuong, et al. "Collm: A large language model for composed image retrieval." CVPR. 2025.

**Ethical Concerns:**

["NO or VERY MINOR ethics concerns only"]

**Final Justification:**

After reading the rebuttals and other reviews, I believe the paper meets the acceptance threshold for the conference. Other reviewers' main concerns are about the reproducible and the ethical issues during reconstructing the benchmark. However, I believe the authors already explain during their rebuttals. I keep my initial score for this paper.

**Limitations:**

Yes

**Quality:**

3

**Strengths And Weaknesses:**

## Strengths
### Quality
The paper presents a comprehensive and complete body of work, supported by extensive experiments and detailed supplementary material. The authors demonstrate a thorough understanding of previous research, providing strong empirical evidence to validate their hypotheses. For the dataset construction, it provides many examples and statistics to show the strengths of the new benchmark. The construction details are also clearly discussed. The new simple but effective baseline is explained clearly and reasonable for the problem when both image and text should be considered.
### Clarity
The paper is exceptionally well-written, complemented by clear and informative figures and tables that effectively highlight its contributions. The narrative flow is easy to follow, enhancing overall readability. The pipeline figure is very easy to understand and connected to the text in Sec 4. Moreover, many examples and additional analysis on both model and dataset are included in the supplementary
### Significance
Both the proposed benchmark and solution offer significant contributions to the community. The new benchmark is particularly impactful, addressing critical ambiguity issues in prior datasets that have historically undermined the trustworthiness of existing methods. Furthermore, the simple yet effective training-free solution opens a promising new direction for research in late-interaction methods within CIR.
### Originality
This work demonstrates novel insights through both its evaluation strategies and its innovative training-free solution for very challenge topic - CIR. It clearly reflects the authors' advanced understanding of the field, pushing the boundaries of current approaches.


## Weaknesses
### Quality
While comprehensive, the work could benefit from the inclusion of additional experiments to further bolster its completeness. (See questions)
### Significance
Despite the proposed solution's efficiency and the acknowledged discussion of its limitations, the modest performance gains on key benchmarks such as CIRR, Fashion-IQ, and CIRCO represent a critical concern regarding its overall impact and practical utility.

---

> ### Author Rebuttal · Authors · 2025-07-31
>
> We thank **Reviewer rzCM** for a **very positive** assessment and for highlighting that:
> (i) the i-CIR construction pipeline is **transparent and well-motivated**,
> (ii) the work is backed by a **comprehensive experimental suite** and rich **supplementary** material that provide **strong empirical evidence for every claim**,
> (iii) the manuscript is **exceptionally clear**—with intuitive figures (particularly the section 4 pipeline) and plentiful illustrative examples—making the narrative **easy to follow**, and
> (iv) the combination of an **ambiguity-free benchmark** and a **simple yet effective training-free BASIC method** represents a **significant and original step** forward for late-interaction approaches in composed image retrieval.
> Below we address the remaining suggestions and concerns:
>
> ---
>
> ### Comparison with Additional SOTA Training-Based Methods
>
> We appreciate the reviewer’s suggestion to include *recent training-based methods*. Using the authors’ official code and checkpoints, we evaluated CoVR-2 and MCL on the five CIR benchmarks of Table 1 of the main paper. As shown in the table below, our *training-free* BASIC still achieves the **best mAP on every dataset**—often by a sizeable margin (e.g. +20.6% on ImageNet-R and +10.5% on i-CIR). These results strengthen the evidence that BASIC offers competitive accuracy without the computational cost of training or fine-tuning.
>
> | Method (venue) | ImageNet-R | NICO++ | MiniDN | LTLL | *i*-CIR |
> |---------------|-----------|--------|--------|------|---------|
> | CoVR-2 (TPAMI 2024) | 11.52 | 24.93 | 27.76 | 24.68 | 36.94 |
> | MCL (ICML 2024) | 8.13 | 19.09 | 18.41 | 16.67 | 24.41 |
> | **BASIC (ours)** | **32.13** | **31.65** | **39.58** | **41.38** | **47.39** |
>
> ---
>
> ### Why Gains Are Smaller on CIRR, FashionIQ, and CIRCO
>
> **1:** These three benchmarks suffer from modality domination: many queries are answerable from the *text alone*. To make the point quantitative, we swept a mixing weight \( λ ∈ [0,1] \) between text-only \( λ = 0 \) and image-only \( λ = 1 \) similarity for three naïve fusion methods (WeiCom, Sum, and Product). The average (over three fusion methods) peak–vs.–unimodal gap is a good proxy for how much a dataset *rewards composition*. On i-CIR the average gap is enormous: **+16.1 mAP (+391% relative)**, whereas it shrinks to +3.0 mAP (+11%) on CIRR, +5.0 mAP (+26%) on FashionIQ, and +6.8 mAP (+167%) on CIRCO. Moreover, the highest unimodal performance is always when \( λ = 0 \), i.e. text-only. Thus legacy datasets reward composition only marginally, whereas *i*-CIR demands strong cross-modal synergy; BASIC is designed for the latter scenario. Detailed results are presented on the Table below.
> **2:** Sensitivity to relational or instructional phrasing (link to **Reviewer pyz4**). As shown in our answer to **Reviewer pyz4**, rewriting just *ten* highly relational CIRR queries into concise, absolute descriptions boosts BASIC from **18.0% → 44.4% mAP** and from **10% → 40% R@1** (Table 2 in that response). This confirms that the low scores stem largely from the relational wording prevalent in those datasets rather than from a fundamental weakness in composition.
> **3.** Following the reviewer’s suggestion, we evaluate BASIC and BASIC without Harris regularization (BASIC\*) on the newly released CoLLM-Refined versions of CIRR and Fashion-IQ (second Table below) . Both variants show notable performance gains on the refined datasets (**up to 13.7% R@1** on refined-Fashion-IQ Shirt). However, it’s important to highlight that the CoLLM refinements primarily aim to *reduce annotation ambiguities* in CIRR and Fashion-IQ—this addresses a different challenge than modality dominance (evidenced by BASIC\* outperforming BASIC) or CLIP’s sensitivity to specific types of phrasing, such as instructional or relational queries.
>
> | λ | **WeiCom CIRCO** | **WeiCom CIRR** | **WeiCom FIQ** | **WeiCom i-CIR** | **Sum CIRCO** | **Sum CIRR** | **Sum FIQ** | **Sum i-CIR** | **Prod CIRCO** | **Prod CIRR** | **Prod FIQ** | **Prod i-CIR** |
> |--------------|-----------------|-----------------|---------------|------------------|--------------|--------------|-------------|---------------|---------------|---------------|--------------|---------------|
> | 0   | 4.00 | 26.09 | 18.94 | 4.11 | 4.10 | 26.65 | 18.95 | 4.10 | 4.10 | 26.66 | 18.95 | 4.10 |
> | 0.1 | 10.84 | 20.86 | 18.47 | 17.94 | 7.00 | 30.70 | 22.52 | 8.89 | 5.30 | 28.36 | 20.29 | 5.24 |
> | 0.2 | 10.84 | 19.98 | 17.80 | 20.94 | 10.30 | 31.23 | 25.69 | 18.30 | 6.49 | 30.08 | 21.39 | 7.63 |
> | 0.3 | 11.07 | 19.60 | 17.79 | 21.39 | 8.14 | 27.40 | 26.95 | 15.97 | 9.11 | 31.19 | 22.86 | 11.37 |
> | 0.4 | 11.10 | 19.51 | 17.42 | 20.72 | 6.93 | 22.48 | 24.71 | 9.61 | 10.35 | 30.73 | 24.63 | 16.83 |
> | 0.5 | 10.95 | 18.86 | 16.91 | 19.48 | 5.61 | 19.02 | 20.83 | 6.04 | 11.12 | 29.37 | 25.95 | 20.78 |
> | 0.6 | 10.99 | 18.31 | 16.28 | 17.52 | 4.57 | 16.26 | 16.22 | 4.11 | 10.50 | 26.33 | 25.94 | 19.24 |
> | 0.7 | 10.52 | 17.99 | 15.57 | 14.77 | 3.82 | 14.61 | 12.54 | 3.09 | 7.61 | 22.18 | 24.06 | 13.51 |
> | 0.8 | 10.77 | 17.22 | 14.49 | 11.70 | 3.16 | 13.46 | 10.24 | 2.56 | 5.75 | 18.05 | 19.48 | 7.99 |
> | 0.9 | 10.55 | 16.62 | 12.70 | 6.98 | 2.86 | 12.49 | 8.66 | 2.28 | 3.93 | 14.45 | 12.60 | 4.06 |
> | 1.0 | 3.26 | 11.56 | 7.69 | 2.07 | 2.61 | 11.73 | 7.69 | 2.07 | 2.61 | 11.73 | 7.72 | 2.07 |
>
> | Dataset | Split | Method Variant | mAP | R@1 | R@5 | R@10 | R@50 |
> |---------|-------|----------------|-----|-----|-----|------|------|
> | **CIRR** | Legacy | BASIC | 24.34 | 15.83 | 40.89 | 53.90 | 82.27 |
> | | Refined | BASIC | **28.28** | **21.87** | **48.96** | **60.82** | **86.66** |
> | | Legacy | BASIC\* | 26.89 | 17.98 | 44.92 | 58.80 | 86.51 |
> | | Refined | BASIC\* | **31.89** | **25.30** | **53.49** | **65.94** | **90.15** |
> | **Fashion-IQ Dress** | Legacy | BASIC | 6.96 | – | – | 18.59 | 37.68 |
> | | Refined | BASIC | **9.91** | – | – | **24.40** | **47.63** |
> | | Legacy | BASIC\* | 7.43 | – | – | 19.98 | 39.86 |
> | | Refined | BASIC\* | **11.27** | – | – | **28.59** | **52.43** |
> | **Fashion-IQ Shirt** | Legacy | BASIC | 10.93 | – | – | 26.30 | 44.80 |
> | | Refined | BASIC | **17.15** | – | – | **36.82** | **56.09** |
> | | Legacy | BASIC\* | 12.93 | – | – | 29.88 | 47.06 |
> | | Refined | BASIC\* | **22.55** | – | – | **43.54** | **62.40** |
> | **Fashion-IQ Toptee** | Legacy | BASIC | 10.28 | – | – | 23.92 | 40.95 |
> | | Refined | BASIC | **14.57** | – | – | **32.08** | **51.41** |
> | | Legacy | BASIC\* | 12.08 | – | – | 26.21 | 44.57 |
> | | Refined | BASIC\* | **19.08** | – | – | **38.27** | **57.98** |
>
> ---
>
> ### Speed and Storage Footprint
>
> We consider Text × Image (“Product”) as the minimal CIR baseline, requiring only dot-products between the CLIP-encoded queries and the precomputed image features. BASIC builds on this with lightweight components—all applied at query time only.
>
> Three components introduce **non-negligible overhead**:
>
> * **Semantic Projection:**
>   Requires storing a d × k projection matrix and applying it to the query, with both time and storage complexity of O(dk).
>
> * **Query Expansion:**
>   In our current implementation, this involves a secondary retrieval over top-Q items (O(QN)), along with a projection step of O(dk). This step can be made efficient using the FAISS library.
>
> * **Contextualization:**
>   This is the most computationally expensive component, requiring one extra (batched) forward pass through the CLIP text encoder, followed by averaging.
>
> Other components—*centering*, *min-based normalization*, and *Harris fusion*—have trivial cost (O(d) or less) and negligible storage overhead.
>
> Total runtime on an NVIDIA A100 40 GB GPU:
>
> | Method | Runtime (ms) |
> |---------|--------------|
> | BASIC | 61.9 |
> | –Contextualization | 25.9 |
> | –Contextualization –Q.Exp. | 25.8 |
> | –Contextualization –Q.Exp. –Projection | 25.6 |
> | Product | 25.5 |
>
> As shown, all components except contextualization add *negligible latency*, with total runtime increasing only from 25.5 ms to 25.9 ms. Contextualization, which involves CLIP forward passes, is the dominant factor—accounting for most of the 61.9 ms cost in the full BASIC pipeline (as shown also in Supplementary Figure S4).
> To mitigate this, a key future direction is to develop a *lightweight embedding network on top of CLIP* that approximates contextualized representations without runtime overhead.

---

> > ### Comment · Reviewer_rzCM · 2025-08-03
> >
> > Thank you for your response. In the last table, how many samples did you perform the test on? How does it change when the number of samples (candidates in the gallery) increases?

---

> > > ### Author Response · Authors · 2025-08-05
> > >
> > > We thank the reviewer for the follow-up. **ImageNet-R** was used to simulate the gallery, containing approximately **17K images**, and the same (image, text) query pair was used across all runs for consistency. The reported timings are averaged over 1,000 runs. To reflect realistic conditions, the reported numbers include both **preprocessing and feature extraction** for the input queries (text and image). This query-side processing dominates the baseline runtime at 25.1 ms, and is also the primary source of contextualization overhead.
> > >
> > > Importantly, all major overheads in BASIC, such as semantic projection or contextualization, are confined to the query side, and do not grow with dataset size. As such, BASIC remains suitable for large-scale retrieval, with scaling behavior comparable to simple dot-product retrieval. To evaluate scalability, we simulate larger galleries by repeating the ImageNet-R images by ×10 and ×100. Below are the observed runtimes:
> > >
> > > | Method | Runtime (ms) | | |
> > > |--------|--------------|--------------|--------------|
> > > | **dataset size** | **x1 (~17K)** | **x10 (~170K)** | **x100 (~1.7M)** |
> > > | BASIC | 61.9 | 63.4 | 72.8 |
> > > | -Contextualization | 25.9 | 27.1 | 36.9 |
> > > | -Contextualization -Q.Exp. | 25.8 | 26.6 | 33.0 |
> > > | -Contextualization -Q.Exp. -Projection | 25.6 | 26.4 | 32.9 |
> > > | Product | 25.5 | 26.3 | 32.8 |
> > >
> > > We observe that even when scaling the gallery by ×100, the overhead introduced by BASIC is similar to that of the baseline Product method. This is because the main computations are performed only once on the query side and are independent of the gallery size. The only module whose cost grows with the dataset is query expansion, which involves a linear scan to identify the top-K candidates (this step can be efficiently implemented in sub-linear time using libraries such as FAISS).
> > >
> > > We hope this clarifies the details regarding the runtime experiments and scalability. We thank the reviewer again for the constructive feedback—these experiments are valuable and will be included in the final manuscript.

---

### Official Review · Reviewer_przL · 2025-07-02

**Clarity:** 3
**Significance:** 4
**Originality:** 2
**Rating:** 4
**Confidence:** 3

**Summary:**

This paper provides a new benchmark dataset for composed image retrieval task called $i$-CIR, which contains each query with visual/textual/composed hard negative samples, making it as challenge as randomly selected distractor samples datasets four orders of magnitude larger. The author also come up with a new composed image retreival method called BASIC which take both query-image-to-image and query-text-to-image similarities into account. Without the necessery of any training process, BASIC beats sota models on $i$-CIR and other CIR datasets.

**Questions:**

### Questions
- Line 122 mentions "another objects of the same category", how are the objects categoried?
- In Line 127~135 the author introduces the process to generate query, positive and hard negative samples with the help of annotators. However, in Line 132 "Finally, annotators manually select images within the visual hard negatives to serve as image queries." makes me confusing. Why is the hard negatives are select as image queries, as all composed positives have already picked up from hard negative pools. What's left must mismatch the seed image/seed sentence.
- In Line 228~230 the author describes how corpus $C_+$ is generated from single-word text queries. Is this process also require the help of LLM or is done by human or fixed templates?

### Suggestions
- Besides the current information metioned in the main text, it is better to add the details on how ChatGPT is used in supplementary materials, including prompts used and several real QA samples.

**Ethical Concerns:**

["NO or VERY MINOR ethics concerns only"]

**Final Justification:**

Most of the problems come with my ignorance of the suplementary materials. And since these are not problems, I improve the clarity score from 2 to 3.
Other problems are well explained in the rebuttal and I am satisfied with the explainations. So I will keep the remaining scores unchanged.

**Limitations:**

yes

**Paper Formatting Concerns:**

The instruction block of the checklist(Line 450 ~ Line 477) should be deleted.

**Quality:**

3

**Strengths And Weaknesses:**

### Strengths:
- The paper is well orgainzed and the author provides details on the building process of the benchmark datsets.
- Through expriments are carried out on sota CIR methods and the results demostrates the effectiveness of BASIC method on CIR task.
- Compared with other CIR benchmarks, $i$-CIR is more challenging and will contributes to the further development in CIR task.
- Though not new models/training schemes/dedicated losses are designed, the paper dive into current available methods and proposes a new training-free framework which beats all sota models in CIR task.
### Weaknesses
- The code is not available until acceptance.
- Supplemental material is not provided, and we fail to check these details mentioned in main text, such as Line 164, Line 244, Line 307, Line 330 and Line 340.
- There exists some typo
  - Firuge 2(a) descriptions should be "image queries"
  - Line 129 "and composed hard negatives for cases (i & ii), (ii), and (iv & v), respectively." the second (ii) should be (iii)
  - Line 279 "a = 0.2" here $a$ maybe refer to $\alpha$ in Eq(2) ?
- Multiple-step of the generation of $i$-CIR benchmark relies on the involvement of human annotators, which may hinder the scaling up of the benchmark.

---

> ### Author Rebuttal · Authors · 2025-07-31
>
> We thank **Reviewer przL** for the review and for explicitly recognising several key strengths of our work:
> (i) the paper is **well organised** and explains the full benchmark-building pipeline,
> (ii) **thorough experiments** on strong baselines demonstrate the **effectiveness of BASIC** on CIR task,
> (iii) i-CIR is **clearly more challenging** than existing CIR datasets and will **“contribute to further development”** in CIR task, and
> (iv) a training-free framework that nevertheless **beats the SOTA** is practically valuable.
>
> Many of the points that the reviewer could not verify *are clarified in the supplementary material* (available via the *ZIP* link on the OpenReview page). We understand that its location *may not have been immediately obvious* and kindly invite the reviewer to consult it. Below, we provide point-by-point responses to all remaining comments.
>
> -----
>
> ### Code Release
> For NeurIPS main-track papers it is permitted to release code **“upon acceptance”**. About the public release policy, full data and code will be released under MIT license immediately upon acceptance. We wish to maximize the dissemination and usage of our dataset and method, thus we will also create a **project page** with engaging visualizations and demos.
>
> ### Supplementray Material is Not Provided / Wrong Statement → It Is Provided
> The full supplementary material—containing additional figures, tables, implementation details, and further discussion—is available via the *ZIP* link on the OpenReview page. We kindly invite the reviewer to consult it, as it addresses the points noted on lines 164, 244, 307, 330, 340, and others.
>
> ### Typos & Minor Edits
> We will fix in the camera-ready version of the paper.
>
> ### Human Annotation vs. Scalability
> **High-quality**, instance-level composed image retrieval ``labels'' cannot be obtained purely automatically, especially given (i) the exhaustive and explicit collection and curation of visual, textual, and **composed hard negatives**, and (ii) the **copyright-safe** and **sentitive-content-safe** approach we decided to follow. In instance-level image-to-image retrieval, datasets like GLDv2 (Google Landmarks Dataset v2 -- A Large-Scale Benchmark for Instance-Level Recognition and Retrieval, CVPR 2020) that are created automatically with crowd-sourced labels are *known to contain a high amount of noise*. Additionally, composed image retrieval has an extra complexity in creating a dataset.
> In CIR, alternative shortcuts in CIRR and FashionIQ result in **well-known weaknesses**, which are explicitly mentioned in our paper (subsection 3.4). These exact weaknesses are *known in the community* and have recently resulted (CoLLM: A Large Language Model for Composed Image Retrieval, CVPR 2025) in the introduction of a *refined version* of them (refined-CIRR, refined-FashionIQ), as mentioned by Reviewer rzCM. We **perform experiments on these refined datasets** as an answer to Reviewer rzCM and observe a *performance boost* of up to 13.7% R@10 on Fashion-IQ Shirt for our method.
> Our semi-automatic pipeline involving expert annotators yields a compact set that is as hard as adding 20M random LAION distractors (supplementary Fig. S5). As a compact benchmark, i-CIR **lowers compute barriers** and **encourages fast iteration**, but **without sacrificing difficulty**. For instance, extracting CLIP ViT-L features for the 300K-image i-CIR takes merely 1 h 25 min on a single NVIDIA V100 (32 GB), whereas processing a 20M-image distractor set would extend to almost four days. Thus, *scaling up is not hindered*. We believe that this trade-off is *preferable* for future benchmarks.
>
> ### Category Hierarchy
> Supp. Fig. S1 shows our 3-level taxonomy. “Same category” in Line 122 refers to level-3 (the most specific) categories.
>
> ### Why a Visual Hard-Negative Can Become an Image Query / All Composed Positives Have Already Picked Up from Hard Negative Pools
> The composed positives are picked from the overall **candidate image pool** (L115-116), which consists of potential queries, positives and (hard) negatives. There are three types of hard negatives: visual, textual, and composed (L130-132). Visual hard negatives include two cases: (i) the same instance under the same or different conditions; (ii) a very similar but different instance. Images of type (i) make excellent alternative image queries. Annotators therefore pick a subset of these and drop all original seed images.
>
> ### Corpus Construction & GPT Usage
> Section S2 of the supplementary material already describes the prompt we used with ChatGPT to generate the positive word corpus ($C_+$), which consists of object-centric terms used to guide semantic projection. However, *due to an oversight*, the corresponding prompt for generating the negative word corpus ($C_-$), which captures stylistic and contextual variations, was not included. We *thank the reviewer for pointing this out*. Below, we provide the exact prompt used:
>
> ```We are working on a composed image retrieval task, where the input consists of an image and a text. The goal is to retrieve the most relevant matching image from a database. Typically, the image contains a main object in a particular “state”. This state can refer to a style (e.g., retro, handwritten, futuristic) or a contextual setting (e.g., next to the sea, on the beach, beside a table). For example: a retro mug, a cat next to a table, a handwritten notebook. We would like you to generate a .txt file containing 100 possible states in which an object can be found. Each line in the file should contain a single state expressed in natural language. Wait for me to say “next” before giving the next batch of states.```
>
> Here are the first 10 outputs of such a command:
>
> ```covered in snow
> on a wooden shelf
> floating in water
> wrapped in plastic
> painted with graffiti
> next to a fireplace
> sitting on grass
> surrounded by candles
> hanging from a tree
> under a spotlight
> ```
>
> To support transparency and reproducibility, we will include this prompt in the revised supplementary material and provide the exact corpora used as part of the released codebase.

---

> > ### Author Response · Authors · 2025-08-05
> >
> > We would greatly appreciate it if the reviewer could *review our rebuttal* and let us know *if our responses address their concerns*. We would be glad to provide additional clarification or engage in further discussion as needed.

---

> > > ### Comment · Reviewer_przL · 2025-08-05
> > >
> > > Sorry for the missing suplementary matrials. I redownloaded the zip file and currently is clear with Line 164, Line 244, Line 307, Line 330 and Line 340 and the category problem.
> > > The author explains that automatic annotation for instance-level composed images will introduce inevitable noices and make it crucial for human annotators and I will accept this.

---

> > > > ### Author Response · Authors · 2025-08-05
> > > >
> > > > Thank you for re-checking the supplementary ZIP and confirming that the earlier issues are resolved. We’re *glad our explanations addressed your concerns*. We appreciate your careful consideration and the updates to your review.

---

### Official Review · Reviewer_RdrS · 2025-07-03

**Clarity:** 1
**Significance:** 2
**Originality:** 2
**Rating:** 2
**Confidence:** 3

**Summary:**

The authors propose an evaluation dataset for instance-level Composed Image Retrieval (iCIR) and propose, BASIC, and VLM-based method to support the task. The method is based on the idea of bidirectional retrieval.

**Questions:**

- Could the authors clarify, what is the main purpose of BASIC? Is it a model to use to help build CIR datasets or a model to perform CIR with? In the abstract it sounds like you want it to do both so I would love to clarify.

- The authors state "Existing CIR datasets often suffer from poor quality due to their construction process, i.e. two similar images are selected automatically and their difference is textually described." What is this statement based on? Could the authors refer to any prior work in the domain that could support this claim? Could they perhaps provide examples that illustrate their point?

- (line 9) "keep the dataset compact to facilitate future research". Could the authors elaborate on the statement? I would argue that, while dataset quality is important, having datasets that are too compact can actually hinder research.

- Beyond the modality domination assumption the authors mention in the footnote and the supplementary materials, I could not find any other limitations of their own work, neither in the paper nor in the supplementary materials; The core limitation is based on the idea of domination of one modality; how does one define such domination? And what about other limitations of this work when it comes to dataset collection process and the core assumptions the authors make there?

**Ethical Concerns:**

["Major Concern: Improper research involving human subjects"]

**Final Justification:**

I thank the authors for the rebuttal. I think the paper represents an interesting idea and would greatly benefit from another round of revision. I encourage the authors to resubmit the paper and keep my score.

**Limitations:**

- Broader impacts are not dicussed.

- Experimental setting/details: while the authors provide an overview of the models, baselines, and datasets, some important details are ommited, making it hard to understand and reproduce the results.

- Results significance testing: the authors limit all the runs to a single run and did not perform any significance tests, making it harder to interpret the results.

- Experiments compute resources: I could not find any information about the compute used in the experiements and the training time.

- Licenses for existing assets: I could not find any information about the repositories/checkpoints the authors used.
Related to this, in the checklist, the authors mention not only CLIP but also BLIP, however, I could not find any information on their usage of BLIP in the paper. Was BLIP also used?

- Crowdsourcing and research with human subjects: Although the authors employed human annotators to curate and filter the dataset, they did not provide any information on the annotation study, raising ethical concerns.
The authors did not describe to the annotators any potential risks, or obtained an IRB, which I find somewhat concerning (see checklist section 14-15 for more details)

**Paper Formatting Concerns:**

The checklist: The authors did not delet the instruction block in the checklist.

**Quality:**

2

**Strengths And Weaknesses:**

Strengths:
- The paper focuses on important problem of CIR.
- The visualizations across the paper and the supplement are very done and do a good job illustrating the approach.
- The authors thought about the safeguard when developing the dataset which is great.


Weaknesses:

- Open access to data and code: while the main contribution of the paper is the dataset, the dataset is not provided. It would have been very helpful to attach even a sample of dataset in the supplementary material. In addition, the authors promise to release the codebase, too, and make several claims about its content, but it is now available at the moment of review, making it hard to estimate the paper contribution.

- The authors open the paper with the statement "Composed image retrieval (CIR) combines image-to-image retrieval and text-to-image retrieval". Perhaps I do not fully understand what they mean but this, but I am not sure I agree with this statement; In my opinion, CIR implies retrieving image, given a multmodal query, typically image+text. The bidirectional retrieval the authors mention is one of the possible ways to implement CIR models.

- Related to this, the authors state "Existing CIR datasets often suffer from poor quality due to their construction process, i.e. two similar images are selected automatically and their difference is textually described." Similarly, I am not sure I agree with this statement. Perhaps the authors could support their claim by referring to relevant prior work or any other motivating examples?

- The authors provide an extensive ablation study, also mentioned in Table 2. While I appreciate the ablation, I do find the table a bit confusing in terms of structure. I think it would have been better to isolate each individual component to better understand its contribution. For example, authors mention that centering provides notable boost (row 2 of the table), but maybe the observed gains are due to the fact that it was the very first component that was switched on? Also, the authors mention that Harris criterion really benefits the model, but, according to the table, having it switched off still leads to very competittive results.

- No clarity on motivation for instance-level CIR. To the best of my knowledge most datasets in the field and instance-level with queries centered on manipulating object properties, e.g., 'image of Eiffel tower but sparkling'.

- While it is great that the authors thought about the dataset safeguards, the model release safeguards are not discussed.

- Baselines:
	- WeiCom, one of the baselines, is not discussed in related work, making it hard to understand how it compares to BASIC.
	- Some relevant baselines, such as TIRG, a classical method for CIR, are excluded from the experiments.

---

> ### Author Rebuttal · Authors · 2025-07-31
>
> We thank **Reviewer RdrS** for the review and for explicitly recognizing strengths of our work:
> (i) **timely problem**: CIR is an important problem,
> (ii) **clear visual presentation** of our method, and
> (iii) **safeguards** in the development of the **dataset**.
> Below, we address each of their comments in detail.
>
> ---
>
> ### Dataset Sample and Code Availability
> **Dataset.** A sample of the dataset is already provided. Supp. Figs. S2 (one image query per instance) and S8 (all text queries) cover most query pairs in i-CIR; Figs. 1 & 3 show composed queries with positives and hard negatives.
> **Code.** For NeurIPS main-track papers it is permitted to release code “upon acceptance”. Full data and code will be released under MIT license immediately upon acceptance. To maximize dissemination, we will also create a **project page** with engaging visualizations and demos.
>
> ### Definition of CIR &nbsp;“bidirectional retrieval”
> Our opening phrase—“CIR combines image-to-image and text-to-image retrieval → (image and text) to image retrieval”—only means that the **query is multimodal (image + text) while the database is visual**. No “bidirectional” retrieval is implied. We can rephrase in the camera-ready for clarity.
>
> ### Why Existing CIR Datasets Can Be Noisy
> The construction protocols of CIRR, CIRCO, and FashionIQ rely on automatic selection of visually or semantically similar image pairs, followed by the annotation of a textual description that highlights their differences. The poor quality of CIRR and FashionIQ is well known and has led to refined-CIRR/FashionIQ (CoLLM, CVPR 2025), as mentioned by Reviewer rzCM. We refer explicitly to the construction details provided in the original papers:
>
> * **CIRR** (§4.1): ``We first form image pairs then collect related annotations by crowd-sourcing. The pairs are drawn from subsets of images, as described below... We use the popular NLVR2 dataset... Prior to forming reference-target image pairs, we construct multiple subsets of six images that are semantically and visually similar, denoted as S... Here, to construct a subset, we randomly pick one image from the large corpus. We then sort the remaining images in dataset by their cosine similarity to this image using ResNet152 image feature vectors pre-trained on ImageNet... Within each constructed image subset S, we draw nine pairs of images... We collect a modification sentence for each pair of reference-target images using Amazon Mechanical Turk (AMT).''.
> * **CIRCO** (§4): ``...We introduce an open-domain benchmarking dataset named Composed Image Retrieval on Common Objects in context (CIRCO). It is based on real-world images from COCO 2017 unlabeled set and is the first dataset for CIR with multiple ground truths. Contrary to CIRR, we start from a \emph{single pair of visually similar images and write a relative caption}... Then, we propose to employ our approach to retrieve the top 100 images according to the query and combine them with the top 50 images most visually similar to the target one. Finally, we select the images that are valid matches for the query.''.
> * **FashionIQ** (§3): "The images of fashion products that comprise Fashion IQ were originally sourced from a product review dataset... For each image, we followed the link to the product website available in the dataset, in order to extract corresponding product information, when available... To ensure that the relative captions can describe the fine-grained visual differences between the reference and target image, we leveraged product title information to select similar images for annotation with relative captions.".
>
> This methodology often yields:
> (i) pairs lacking meaningful fine-grained differences (supp. Fig. S10, rows 4–5);
> (ii) text-only queries sufficing to find the target (supp. Fig. S9, row 4; quantitative in rebuttal Table in the answer to Reviewer rzCM);
> (iii) negative pools containing false negatives (supp. Fig. S9, row 6).
>
> Our semi-automatic pipeline with expert annotators yields a **compact 300 K-image benchmark that is as hard as adding 20M random LAION distractors** (supp. Fig. S5).
>
> ### Ablation Table Structure & Harris Regularization
>
> Table 2 is cumulative by design: we switch BASIC’s components on to illustrate how performance builds up through a **logical sequence** of design steps. **Ordering is not arbitrary**: Some components *depend* on the presence of others to be *effective* (e.g. projection assumes centred features, Harris requires min-normalised scores). At the reviewer's request, we also report a leave-one-out ablation (Table below), but note the expected caveats: removing min-norm or centering while leaving dependent steps active naturally hurts results. We hope this expanded analysis provides a clearer understanding of each component’s role and justifies the original cumulative layout of Table 2. Finally, Harris is a novel contribution to CIR that brings consistent improvements over all cases, even if its gains may appear more modest compared to other components. Its strength is aligned with the goal of CIR, as it reinforces retrievals that satisfy both the image and the text query, rather than favouring one modality over the other.
>
> | Centering | Min Norm. | Harris | Context. | Proj. | Q. Exp. | ImageNet-R | NICO | MiniDN | LTLL | i-CIR |
> |-----------|-----------|--------|----------|-------|---------|-----------:|-----:|-------:|-----:|------:|
> | ✓ | ✓ | ✓ | ✓ | ✓ | ✓ | **32.13** | **31.65** | **39.58** | **41.38** | 47.39 |
> | ✗ | ✓ | ✓ | ✓ | ✓ | ✓ | 23.29 | 19.93 | 24.87 | 10.16 | 37.85 |
> | ✓ | ✗ | ✓ | ✓ | ✓ | ✓ | 18.83 | 19.93 | 24.67 | 4.36 | 42.14 |
> | ✓ | ✓ | ✗ | ✓ | ✓ | ✓ | 30.75 | 29.82 | 38.85 | 40.65 | 46.11 |
> | ✓ | ✓ | ✓ | ✗ | ✓ | ✓ | 26.18 | 30.61 | 33.64 | 34.50 | 39.74 |
> | ✓ | ✓ | ✓ | ✓ | ✗ | ✓ | 17.31 | 13.96 | 21.22 | 22.42 | 27.21 |
> | ✓ | ✓ | ✓ | ✓ | ✓ | ✗ | 27.54 | 28.90 | 35.75 | 38.22 | **47.73** |
>
> ### Motivation for Instance-level CIR
> We kindly *disagree* with the reviewer. Even though the example that the reviewer provides, mentions the Eiffel tower → is an instance-level one, **existing datasets follow a semantic-level definition** (we cannot guarantee that a tiny fraction of instance-level queries does not exist, since we have not manually inspected the whole datasets). Examples from original papers:
> (i) CIRCO [2] Fig. 7 retrieves *different* lunch box, clock, horse, etc.;
> (ii) CIRR [23] Figs. 1&4 retrieve *different* dog or carriage;
> (iii) FashionIQ [35] Fig. 3 retrieves a *different* T-shirt/blouse;
> (iv) ImageNet-R (Pic2Word) [30] Fig. 3 retrieves *any* fish/tiger/shark matching the style cue.
> Instance-level CIR therefore **fills a genuine gap**.
>
> ### “Model-release safeguards”
> We release **no new model weights**; BASIC is training-free (regarding composed retrieval) and it uses public OpenCLIP/SigLIP checkpoints. Only the original licences apply.
>
> ### Baselines
> * **WeiCom** = convex combination of normalised image-to-image and text-to-image similarities (λ ∈ [0, 1]). We shall add.
> * **TIRG** requires supervised triplet training, so it is not a zero-shot baseline.
>
> ### Purpose of BASIC
> BASIC is **purely a retrieval method**; it is **not** used in dataset construction.
>
> ### “Compact Yet Hard” Dataset
> Thanks to curated visual, textual, and composed hard negatives, a 300K-image i-CIR database matches the difficulty of 20M LAION distractors. Extracting CLIP ViT-L features: 1 h 25 m on a single V100 vs. ≈ 4 days for 20 M images. Therefore experimentation and benchmarking is **significantly sped up** while **keeping the difficulty of the dataset high**.
>
> ### Limitations
> **Methodology-wise**, we already mention two limitations: modality-domination and cases where text query takes the form of an instructional or relational phrasing (in which training-free methods can underperform). Both are further explored in this rebuttal; see responses to Reviewer rzCM and Reviewer pyz4, respectively. Throughout the manuscript we also mention other limitations, such as the computational overhead of contextualization and the construction process of corpora via ChatGPT. **Dataset-wise**, our semi-automatic pipeline trades some speed for fidelity: every composed query is manually vetted (e.g. for hard negatives, licence/safety issues, etc.), so building i-CIR is slower and costlier than fully automatic datasets but yields an **ambiguity-free** benchmark the field currently lacks (Reviewer rzCM). Moreover, the design mostly assumes a single salient object per image query and uses English-only phrasing with public-domain imagery. We shall add.
>
> ### Experimental Details
> * Determinism: BASIC has **no trainable parameters**; single runs suffice.
> * Compute: see supp. Fig. S4 for exact timings.
> * Checkpoints/licences: OpenCLIP ViT-L/14 and SigLIP ViT-L/16 (Apache-2.0). BLIP not used—checklist typo.
>
> ### Crowdsourcing & IRB
> We appreciate the reviewer’s concern regarding ethical practices involving human annotators. To clarify, the human annotators in our study were **employed** (salaried institution employees) **solely for the task of labelling and curating images** for a new instance-level composed image retrieval dataset. The annotators were trained workers performing a well-defined annotation task. **Our research is not performed on human subjects**. No personal or sensitive data, such as images of people, faces, license plates, or other identifiable features, were included in the dataset. As such, our use of annotators falls under standard data labelling practices and *does not meet the criteria for human subjects research that would require IRB oversight*. We will update the paper and the checklist to clarify these points and avoid any misunderstanding regarding ethical compliance.

---

> > ### Author Response · Authors · 2025-08-05
> >
> > We would greatly appreciate it if the reviewer could *review our rebuttal* and let us know *if our responses address their concerns*. We would be glad to provide additional clarification or engage in further discussion as needed.

---

> > > ### Comment · Reviewer_RdrS · 2025-08-07
> > >
> > > I appreciate the authors’ rebuttal and discussion. These responses clarified some of my earlier concerns. However, many issues remain insufficiently resolved: the dataset and code were not available for proper evaluation at review time; broader impacts and limitations are not clearly and cohesively presented; some baseline choices and comparisons remain underdeveloped; and the experimental rigor is weakened by the absence of significance testing and limited reproducibility details in the main paper. As several aspects of clarity, transparency, and evaluation remain unclear, I am keeping my original score.

---

> > > > ### Author Response · Authors · 2025-08-08
> > > > **Appreciation and Request for Further Clarification**
> > > >
> > > > We sincerely thank the reviewer for engaging with our rebuttal and for *acknowledging that some concerns have been clarified*. We’d greatly appreciate the opportunity to **better understand the remaining points**, as we are committed to addressing any gaps in clarity, transparency, or reproducibility.
> > > >
> > > > If the reviewer is willing, we would be grateful for *further elaboration* on a few aspects mentioned in the final comment:
> > > >
> > > > - *“Broader impacts and limitations are not clearly and cohesively presented”*: We’d like to kindly refer the reviewer to the sections we added in direct response to the Ethics Reviewers: **Broader-Impact & Dual-Use Discussion**, **Residual Dual-Use Vectors & Mitigations**, and **Transparency & Governance Commitments**. If possible, we would appreciate more *specific feedback* on which aspects felt unclear or incomplete.
> > > >
> > > >
> > > > - *“Some baseline choices and comparisons remain underdeveloped”*: Questions related to TIRG and WeiCom are addressed in our rebuttal. Also note that in the response to Reviewer rzCM a new comparison with **two additional methods** (CoVR-2, MCL) is provided. We would be glad if the reviewer can *be more specific* about what is still missing.
> > > >
> > > >
> > > > - *“Absence of significance testing”*: Our method is **fully deterministic and training-free**, so conventional significance testing (across repeated training runs) *does not apply*. If the reviewer had a different type of analysis in mind (e.g., bootstrapping), we’d be happy if the reviewer can *be more specific*.
> > > >
> > > >
> > > > - *“Limited reproducibility details in the main paper”*: The original review merely states “some important details are ommited”. Note that our responses to reviewers **include several clarifications** with respect to experimental settings and the method (e.g. Reviewer przL: “Corpus Construction & GPT Usage”, Reviewer rzCM: “Speed and Storage Footprint”). Therefore, we would appreciate it if the reviewer can *be more specific*.
> > > >
> > > > **Thank you once again for your feedback and time**. We remain fully open to refining our work and clarifying any remaining concerns.

---

### Decision · Program_Chairs · 2025-09-17

**Decision:**

Accept (poster)

**Comment:**

Key contributions:

i-CIR benchmark: curated visual/text/composed hard negatives; substantially harder than legacy sets.
BASIC method: strong zero-shot results across i-CIR, ImageNet-R, NICO++, MiniDN, LTLL.

2) Main weaknesses: Whether mitigated

Availability/reproducibility at review time: Dataset/code not public; missing significance tests/compute details. Somewhat mitigated during discussion.

Some discussion about human-subjects compliance for annotations.

3) Most important reasons for accept recommendation: Reviewers (nearly) universally positive.  Most negative reviewer was imposing an implicit requirement that dataset needs to be public at time of submission.  This is not the NeurIPS policy and I believe that the authors fulfilled community expectations as they exist today.